# COMPUTATIONAL BOTTLENECKS FOR DENOISING DIFFUSIONS

**Andrea Montanari & Viet Vu**
Department of Statistics, Stanford University
Stanford, CA 94305, USA
{montanar, vietvu01}@stanford.edu

## ABSTRACT

Denoising diffusions sample from a probability distribution $\mu$ in $\mathbb{R}^d$ by constructing a stochastic process $(\hat{\boldsymbol{x}}_t : t \geq 0)$ in $\mathbb{R}^d$ such that $\hat{\boldsymbol{x}}_0$ is easy to sample, but the distribution of $\hat{\boldsymbol{x}}_T$ at large $T$ approximates $\mu$. The drift $\boldsymbol{m} : \mathbb{R}^d \times \mathbb{R} \to \mathbb{R}^d$ of this diffusion process is learned by minimizing a score-matching objective.

Is every probability distribution $\mu$, for which sampling is tractable, also amenable to sampling via diffusions? We address this question by studying its relation to information-computation gaps in statistical estimation. Earlier work in this area constructs broad families of distributions $\mu$ for which sampling is easy, but approximating the drift $\boldsymbol{m}(\boldsymbol{y}, t)$ is conjectured to be intractable, and provides rigorous evidence for intractability.

We prove that this implies a failure of sampling via diffusions. First, there exist drifts whose score matching objective is superpolynomially close to the optimum value among polynomial time drifts and yet produce samples with distribution that is very far from the target $\mu$. Second, any polynomial-time drift that is also Lipschitz continuous results in equally incorrect sampling.

We instantiate our results on the toy problem of sampling a sparse low-rank matrix, and further demonstrate empirically the failure of diffusion-based sampling. Our work implies that caution should be used in adopting diffusion sampling when other approaches are available.

## 1 INTRODUCTION

### 1.1 BACKGROUND

Diffusion sampling (DS) (Song & Ermon, 2019; Ho et al., 2020) has emerged as a central paradigm in generative artificial intelligence (AI). Given a target distribution $\mu$ on $\mathbb{R}^d$, we want to sample $\boldsymbol{x} \sim \mu$. Diffusions achieve this goal by generating trajectories of a stochastic process $(\hat{\boldsymbol{x}}_t)$ whose state $\hat{\boldsymbol{x}}_T$ at large $T$ is approximately distributed according to $\mu$. This suggests a natural question:

**Q:** Are there distributions $\mu$ for which sampling via diffusions fails even if sampling from $\mu$ is easy?

In order to explain how DS might fail, it is useful to recall the setup and introduce some notations[1]. The basic DS approach implements an approximation of the following stochastic differential equation (SDE), with initialization $\boldsymbol{y}_0 = \boldsymbol{0}$:

$$\mathrm{d}\boldsymbol{y}_t = \boldsymbol{m}(\boldsymbol{y}_t; t)\mathrm{d}t + \mathrm{d}\boldsymbol{B}_t, \tag{1}$$

$$\boldsymbol{m}(\boldsymbol{y}, t) := \mathbb{E}\{\boldsymbol{x} | t\boldsymbol{x} + \sqrt{t}\boldsymbol{g} = \boldsymbol{y}\}, \tag{2}$$

where $(\boldsymbol{B}_t)_{t \geq 0}$ is Brownian motion (BM) and in Eq. (2) $\boldsymbol{x} \sim \mu$ is independent of $\boldsymbol{g} \sim \mathsf{N}(\boldsymbol{0}, \boldsymbol{I}_d)$.

It is not hard to show that, if $\boldsymbol{y}_t$ is generated according to the above SDE, then there exists $\boldsymbol{x} \sim \mu$ and an independent standard BM $(\boldsymbol{W}_t)_{t \geq 0}$ (different from $(\boldsymbol{B}_t)_{t \geq 0}$) such that

$$\boldsymbol{y}_t = t\,\boldsymbol{x} + \boldsymbol{W}_t \,. \tag{3}$$

---

[1] We follow the formulation of Montanari (2023), which does not require time reversal (c.f. Appendix B)

Therefore, running the diffusion (1) until some large time $T$, and returning $\boldsymbol{y}_T/T$ or $\boldsymbol{m}(\boldsymbol{y}_T, T)$ yields a sample approximately distributed according to $\mu$.

In practice, the function $\boldsymbol{m}$ is generally not accessible (cf. discussion below (6)), and is replaced by an approximation $\hat{\boldsymbol{m}}(\boldsymbol{y}, t)$. We can implement an Euler discretization of the SDE (1):

$$\hat{\boldsymbol{y}}_{t+\Delta} = \hat{\boldsymbol{y}}_t + \hat{\boldsymbol{m}}(\hat{\boldsymbol{y}}_t, t)\Delta + \sqrt{\Delta}\,\hat{\boldsymbol{z}}_t\,, \tag{4}$$

with $\Delta$ a small stepsize, and $(\hat{\boldsymbol{z}}_t)_{t\in\mathbb{N}\Delta} \sim_{iid} \mathsf{N}(\boldsymbol{0}, \boldsymbol{I}_d)$. After iterating (4) up to a large time $T$, we output $\hat{\boldsymbol{x}}_T = \hat{\boldsymbol{m}}(\hat{\boldsymbol{y}}_T, T)$. We refer to $\hat{\boldsymbol{x}}_T$ as a *diffusion sample*.

Diffusions reduce the problem of sampling from a distribution $\mu$ to that of approximating the conditional expectation $\boldsymbol{m}$ (Eq. (2)) by $\hat{\boldsymbol{m}}$. The mapping $\boldsymbol{y} \mapsto \boldsymbol{m}(\boldsymbol{y}, t)$ is the Bayes-optimal estimator of $\boldsymbol{x}$ in Gaussian noise:

$$\boldsymbol{m}(\,\cdot\,, t) = \underset{\boldsymbol{\varphi}:\mathbb{R}^n\to\mathbb{R}^n}{\arg\min}\; \mathbb{E}\big\{\|\boldsymbol{\varphi}(\boldsymbol{y}_t) - \boldsymbol{x}\|^2\big\}\,. \tag{5}$$

In words, we are given a Gaussian observation $\boldsymbol{y}_t \sim \mathsf{N}(t\boldsymbol{x}, t\boldsymbol{I}_d)$ (for a single $t$) and want to estimate $\boldsymbol{x}$ as to minimize mean square error (MSE). This is also known as the 'score-matching objective'.

The minimization in Eq. (5) has to be modified for two reasons: *First,* in general we do not know the distribution of $\boldsymbol{x}$ over which the expectation in (5) is taken; we only have a sample $(\boldsymbol{x}_i)_{i\leq N} \sim_{iid} \mu$. We thus replace the MSE by its sample version:

$$\text{minimize}\quad \frac{1}{N}\sum_{i=1}^{N}\big\|\boldsymbol{\varphi}(\boldsymbol{y}_{i,t}) - \boldsymbol{x}_i\big\|^2\,,\quad \text{subj. to}\;\; \boldsymbol{\varphi}\in\mathscr{N}\,, \tag{6}$$

where $\boldsymbol{y}_{i,t} = t\boldsymbol{x}_i + \sqrt{t}\boldsymbol{g}_i$ for $(\boldsymbol{g}_i)_{i\leq N}\sim_{iid}\mathsf{N}(0, \boldsymbol{I}_d)$. The minimization in (5) must be restricted to a function class $\mathscr{N}$ (e.g. neural nets). A (near-)optimal solution to (6) will be $\hat{\boldsymbol{m}}$.

*Second,* to efficiently implement the generative process (4), $\hat{\boldsymbol{m}}$ should be computable in polynomial time. For this reason, $\mathscr{N}$ must be a set of such functions. This is a purely computational constraint, and is present even if we have access to $\mu$ (i.e., for $N = \infty$).

Most of the literature on diffusion sampling studies how samples quality deteriorates because of finite sample size $N$ or non-vanishing step size $\Delta$. Here we focus on a more fundamental limitation that arises because $\hat{\boldsymbol{m}}$ must be computable in polynomial time (the second remark above).

A key remark here is that the ideal drift $\boldsymbol{m}(\boldsymbol{y}, t)$ is the Bayes-optimal denoiser, see (5). Namely it is the optimal function to estimate $\boldsymbol{x}$ with prior distribution $\mu$ from noisy observations $\boldsymbol{y}_t \sim \mathsf{N}(t\boldsymbol{x}, t\boldsymbol{I}_d)$: $t$ can be interpreted as the signal-to-noise ratio (SNR) of this denoising problem. We will say that an *information-computation gap* arises for this problem (at SNR $t$) there exists a constant $\mathsf{gap}(t) > 0$ such that, for all polynomial-time algorithms $\hat{\boldsymbol{m}}$, if $d$ is large enough

$$\mathbb{E}\big\{\|\hat{\boldsymbol{m}}(\boldsymbol{y}_t) - \boldsymbol{x}\|^2\big\} \geq \inf_{\boldsymbol{\varphi}} \mathbb{E}\big\{\|\boldsymbol{\varphi}(\boldsymbol{y}_t) - \boldsymbol{x}\|^2\big\} + \mathsf{gap}(t)\,. \tag{7}$$

Recent literature provides many instances of statistical estimation problems for which an information-computation gap is shown to exist (Brennan et al., 2018; Bandeira et al., 2022; Celentano & Montanari, 2022; Schramm & Wein, 2022) conditional on certain widely accepted conjectures. We stress that the conditional/conjectural nature of these results is, so far, unavoidable, a situation analogous to classical complexity theory that relies on P$\neq$NP. Several of the problems for which a gap arises take the form of estimating $\boldsymbol{x} \sim \mu$ from observations $\boldsymbol{y}_t = t\boldsymbol{x} + \sqrt{t}\,\boldsymbol{g}$.

Koehler & Vuong (2024) already pointed out informally that denoising problems presenting an information-computation gap can result into a failure of DS. As a concrete example, they suggested the spiked Wigner model (c.f. next section). While this informal remark is natural, making it mathematically precise is far from obvious. In fact –strictly speaking– the remark is **false**. If sampling from $\mu$ is easy, then the drift $\boldsymbol{m}(\boldsymbol{y}, t)$ can be constructed to return (for all $t \geq t_0$) a fixed random sample $\boldsymbol{x} \sim \mu$. Then the diffusion will sample correctly. However such $\boldsymbol{m}$ will be very far from an optimal denoiser. (See Proposition 2.1 for formal version of this counter-example.)

We also note that several earlier papers provided examples of probability distributions $\mu$ from physics and Bayesian statistics for which Gibbs sampling is expected to succeed, but DS appears

to fail (Montanari et al., 2007; Ricci-Tersenghi & Semerjian, 2009; Ghio et al., 2024; Huang et al., 2024). None of these papers presented a formal claim either.

The present paper fills this gap in the literature. We prove two general results that hold for any distribution $\mu$ that presents a certain version of information-computation gap (see formal statements below). *First,* we prove that there exists drifts that are approximate optimizers of the score matching objective (6) among polynomial time algorithms (up to an sub-polynomially small error) and yet lead to completely incorrect sampling. *Second,* we show that *every* polynomial-time computable drift that is a near optimum of score matching and is also Lipschitz continuous leads to incorrect sampling. *Finally,* we ilustrate the applicability of our theorems by studying a toy example, namely sampling a sparse low-rank matrix.

We emphasize that this failure of DS is of computational of nature and purely related to the requirement to approximate the Bayes optimal denoiser $\boldsymbol{m}(\boldsymbol{y}, t)$ by a polytime computable function.

## 1.2 SUMMARY OF RESULTS

Recall that the Wasserstein-1 distance between two measures $\mu_1, \mu_2$ on $\mathbb{R}^d$ is defined as

$$W_1(\mu_1, \mu_2) := \inf_{\gamma \in \mathcal{C}(\mu_1, \mu_2)} \int \|\boldsymbol{x}_1 - \boldsymbol{x}_2\|_2 \, \gamma(\mathrm{d}\boldsymbol{x}_1, \mathrm{d}\boldsymbol{x}_2) \,,$$

with the infimum taken over couplings on $\mu_1$ and $\mu_2$. Given random vectors $\boldsymbol{X}_1, \boldsymbol{X}_2$ we denote by $W_1(\boldsymbol{X}_1, \boldsymbol{X}_2)$ the $W_1$-distance of their distributions. We prove lower bounds on the $W_1$ to show incorrect sampling. Since we only consider distributions $\mu$ supported on vectors with bounded norm, a lower bound on $W_1$ implies lower bounds on TV distance and KL divergence. Hence our impossibility results are stated in a strong form.

As a running example/application, we will let $\mu$ to be the following distribution over $n \times n$ sparse low-rank matrices. Let $B_{n,k} := \{\boldsymbol{u} \in \{0, \pm 1/\sqrt{k}\}^n \,|\, \|\boldsymbol{u}\|_0 = k\}$ be the set of $0/ \pm (1/\sqrt{k})$ unit vectors with $k$ nonzero entries ($\|\boldsymbol{u}\|_0$ denotes the number of nonzeros in $\boldsymbol{u}$). We define the target distribution $\mu = \mu_{n,k}$ to be the distribution of $\boldsymbol{x} = \boldsymbol{u}\boldsymbol{u}^\mathsf{T}$ when $\boldsymbol{u} \sim \mathrm{Unif}(B_{n,k})$. Note that $\boldsymbol{x} \in \mathbb{R}^{n \times n}$ is a matrix that we identify with a vector in $\mathbb{R}^d$ for $d = n^2$. Sampling from $\mu$ is trivial: just sample a vector with entries in $\{0, 1/\sqrt{k}, -1/\sqrt{k}\}$ and exactly $k$ non-zero entries, and let $\boldsymbol{x} = \boldsymbol{u}\boldsymbol{u}^\mathsf{T}$. However, rigorous evidence supports the claim that —for certain scalings of $k, t$ with $n$— polynomial-time algorithms cannot approach the Bayes-optimal error (Butucea et al., 2015; Ma & Wu, 2015; Cai et al., 2017; Brennan et al., 2018; Schramm & Wein, 2022).

We will prove two sets of main results that hold for distributions $\mu$ such that the denoising problem presents an information-computation gap:

**1. (Theorem 1, Corollaries 3.2, D.1)** Near optimizers of score-matching can sample incorrectly. We prove that there exists $\hat{\boldsymbol{m}} : \mathbb{R}^{n \times n} \times \mathbb{R} \to \mathbb{R}^{n \times n}$ such that:

M1. $\hat{\boldsymbol{m}}(\,\cdot\,)$ can be evaluated in polynomial time.
M2. The estimation error achieved by $\hat{\boldsymbol{m}}$ (namely, $\mathbb{E}\{\|\hat{\boldsymbol{m}}(\boldsymbol{y}_t, t) - \boldsymbol{x}\|^2\}$) is close to the optimal estimation error achieved by polynomial-time algorithms. Hence $\hat{\boldsymbol{m}}(\,\cdot\,, t)$ will be a near minimizer of the score-matching objective (5) (integrated over $t$).
M3. Samples $\hat{\boldsymbol{x}}_T$ generated by the discretized diffusions (4) with drift $\hat{\boldsymbol{m}}(\,\cdot\,, t)$ at some large time $T$ have distribution that is very far from the target $\mu$ ('as far as it can be' in $W_1$ distance.)

**2. (Theorem 3, Corollary 5.1)** *All* (sufficiently) Lipschitz score-matching optimizers sample incorrectly. More precisely, we prove that any denoiser that near optimizes the score matching among polytime algorithms, acts optimally on pure noise data, and is $C/t$-Lipschitz for $t > t_1$ (for any constant $C$ a suitable $t_1$), samples incorrectly.

Additionally, (Theorems 2, 5), we prove a reduction from estimation to DS. Namely, if accurate, polytime DS is possible, then near Bayes optimal estimation of $\boldsymbol{x}$ from $\boldsymbol{y}_t = t\boldsymbol{x} + \sqrt{t}\boldsymbol{g}$ must also be possible in polynomial time for all $t$. The contrapositive of this statement implies that if an information-computation gap exists, then (near)-correct DS is impossible in polynomial time.

**Roadmap.** The rest of the paper is as follows. In Section 2 we motivate our setting and assumptions, and discuss some limitations of our results. In Section 3 we state formally our results (for technical

reasons we state two separate results depending on the growth of $k$ with $n$.) Section 4 presents the general reduction from estimation to diffusion sampling. Section 5 proves that all Lipschitz score matching optimizers fail. Section 6 provides a numerical experiment of a neural network $\hat{\boldsymbol{m}}$ that outperforms (conjectured) asymptotically optimal denoisers for finite $n$, yet still samples poorly.

**Notation.** Throughout, $a_n \ll b_n$ means $a_n/b_n \to 0$. We refer to Appendix A for notations.

## 2   DISCUSSION

**Setting.** Our results indicate that a standard application of denoising diffusions methodology will fail to sample from $\mu$ when the associated denoising problem presents an information-computation gap. The example $\mu_{n,k}$ of sparse low-rank matrices shows that DS can fail in cases in which sampling from $\mu$ is trivial.

Our example also shows that the latent structure of the distribution can be exploited to construct a better algorithm. Namely, one can use diffusions to sample $\boldsymbol{u} \sim \text{Unif}(B_{n,k})$ (the posterior expectation $\boldsymbol{m}(\boldsymbol{y}, t)$ is polytime-computable) and then generate $\boldsymbol{x} = \boldsymbol{u}\boldsymbol{u}^\mathsf{T}$. On the other hand, identifying such latent structures from data can be hard in general, both statistically and computationally.

**Limitations.** We prove that there exists drifts $\hat{\boldsymbol{m}}(\,\cdot\,, t)$ that lead to poor sampling, despite being nearly optimal (among poly-time algorithms) in terms of the score matching objective (5). In particular, these bad drifts will be near optimal solutions of the problem of (6), as long as $\mathcal{N}$ only contains polytime methods and is rich enough to approximate them. We further exclude the existence of Lipschitz drifts $\hat{\boldsymbol{m}}(\,\cdot\,, t)$ that also satisfy conditions M1 and M2 but yield good generative sampling.

In principle there could still be non-Lipschitz polytime drifts that are near score matching optimizers and sample well. However if such drifts exist, our results suggest that minimizing the score matching objective is not the right approach to find them (since the difference in value with bad drifts will be superpolynomially small).

**Correct samplers violating M2.** If we drop condition M2, i.e. we accept drifts that are bad for the score-matching objective, then it is possible to construct drifts that can be evaluated in polynomial time and yield good sampling. This is stated formally below and proven in Appendix J.

**Proposition 2.1.** *Suppose that a discretized SDE $(\hat{\boldsymbol{y}}_{\ell\Delta})_{\ell \geq 0}$ per (4) is generated, with step size $\Delta > 0$ and noise stream $\hat{\boldsymbol{z}}_t \overset{i.i.d.}{\sim} \mathsf{N}(0, \boldsymbol{I}_{n \times n})$. Then for every $n, k$, there exists a function $\hat{\boldsymbol{m}}(\boldsymbol{y}, t) = \hat{\boldsymbol{m}}(\boldsymbol{y}, t; \hat{\boldsymbol{z}}_1)$ parametrized by $\hat{\boldsymbol{z}}_1$ (with no additional randomness) such that: (i) $\mathbb{E}[\|\hat{\boldsymbol{m}}(\boldsymbol{y}_t, t) - \boldsymbol{x}\|^2] = 2(1 - o(1))$ uniformly for every $t \geq 0$ (sub-optimal score-matching); (ii) $W_1(\hat{\boldsymbol{m}}(\hat{\boldsymbol{y}}_{\ell\Delta}, \ell\Delta), \boldsymbol{x}) = 0$ for all $\ell \geq 0$ ($\hat{\boldsymbol{m}}(\hat{\boldsymbol{y}}_t, t)$ is an approximate sample of $\boldsymbol{x}$); (iii) $\lim_{\ell \to \infty} W_1(\hat{\boldsymbol{y}}_{\ell\Delta}/(\ell\Delta), \boldsymbol{x}) = 0$ ($\hat{\boldsymbol{y}}_t/t$ is an approximate sample of $\boldsymbol{x}$ at large time).*

The drift constructed in this proposition has very poor value of the score-matching objective.

**Further related work.** A number of groups proved positive results on diffusion sampling. Alaoui et al. (2022); Chen et al. (2023b); Montanari & Wu (2023); Lee et al. (2023); Benton et al. (2023) provide reductions from diffusion sampling to score estimation. Chen et al. (2023a); Shah et al. (2023); Mei & Wu (2025); Li et al. (2024) give end-to-end guarantees for classes of distributions $\mu$.

The computational bottleneck that we study here has been observed before in the context of certain Gibbs measures and Bayes posterior distributions Ghio et al. (2024); Alaoui et al. (2023); Huang et al. (2024), and random constraint satisfaction problems Montanari et al. (2007); Ricci-Tersenghi & Semerjian (2009) (the later papers use sequential sampling rather than diffusion sampling).

Our work provides an approach to rigorize the latter line of work.

## 3   NEAR-OPTIMAL POLYTIME DRIFTS WITH INCORRECT DIFFUSION SAMPLING

Given an arbitrary polytime computable drift $\hat{\boldsymbol{m}}_0$, we will construct a different polytime drift $\hat{\boldsymbol{m}}$, with nearly equal score matching objective and yet incorrect sampling. In Subsection 3.1, we state

our assumptions and general result. In Subsection 3.2, we apply the general theorem to the example of sampling sparse low-rank matrices. We also indicate several other similar examples.

In what follows $(\boldsymbol{x}, \boldsymbol{y}_t)$ will always be distributed according to the ideal diffusion process of (3), which also satisfies (2). In particular $\boldsymbol{x} \sim \mu$, $\boldsymbol{y}_t = t\boldsymbol{x} + \boldsymbol{W}_t$, for $(\boldsymbol{W}_t)_{t \geq 0}$ a BM. On the other hand, $(\hat{\boldsymbol{y}}_t)$ will denote the process generated with the implemented procedure (4).

## 3.1 GENERAL RESULT

Throughout, we will consider distributions $\mu$ that are supported on $\mathsf{B}^d(1) := \{\boldsymbol{x} \in \mathbb{R}^d : \|\boldsymbol{x}\|_2 \leq 1\}$. We will state our assumptions and results having in mind the case of measures that are roughly centered: $\mathbb{E}_{\boldsymbol{x} \sim \mu}[\boldsymbol{x}] = \int \boldsymbol{x}\, \mu(\mathrm{d}\boldsymbol{x}) \approx \boldsymbol{0}$, although this condition is not formally needed.

Our first main assumption is that any polynomial-time algorithm to estimate $\boldsymbol{x}$ from $\boldsymbol{y}_t \sim \mathsf{N}(t\boldsymbol{x}, t\boldsymbol{I}_d)$ fails when $t$ is below a certain threshold $t_{\mathrm{alg}}$. When $t/t_{\mathrm{alg}} < 1$, we expect that polytime algorithms will not perform better (in score-matching, c.f. (5)) than the best constant estimator of $\boldsymbol{x}$, namely $\mathbb{E}_{\boldsymbol{x} \sim \mu}[\boldsymbol{x}]$. In the case $\|\mathbb{E}_{\boldsymbol{x} \sim \mu}[\boldsymbol{x}]\| \approx 0$, it follows that polytime algorithms $\hat{\boldsymbol{m}}_0$ with good score-matching will have small norm $\|\hat{\boldsymbol{m}}_0(\boldsymbol{y}_t, t)\|$. This small-norm property is captured by our assumption. More details are discussed at the beginning of Subsection 3.2.1, and Proposition C.1.

**Assumption 1** (Small norm below threshold). *Let $\boldsymbol{y}_t = t\boldsymbol{x} + \boldsymbol{W}_t$, for $(\boldsymbol{x}, (\boldsymbol{W}_t)_{t \geq 0}) \sim \mu \otimes \mathrm{BM}$. Then, there exists a function $\eta_1 : \mathbb{N} \to \mathbb{R}$ (which we refer to as 'rate') such that $\eta_1(d) = o_d(1)$ and, for any $\varepsilon, \gamma > 0$,*

$$\int_0^{(1-\gamma)t_{\mathrm{alg}}} \mathbb{P}\big(\|\hat{\boldsymbol{m}}_0(\boldsymbol{y}_t, t)\| \geq \varepsilon\big)\, \mathrm{d}t = O(\eta_1(d))\,.$$

Our second assumption is that polytime *detection* is reliable for $t$ above $t_{\mathrm{alg}}$. By detection, we consider the following hypothesis testing problem. Given $\boldsymbol{y} \in \mathbb{R}^d$, we test if $\boldsymbol{y}$ is distributed as $t\boldsymbol{x} + \sqrt{t}\mathsf{N}(\boldsymbol{0}, \boldsymbol{I}_d)$ or as $\mathsf{N}(\boldsymbol{a}, t\boldsymbol{I}_d)$ for $\|\boldsymbol{a}\|$ small, where $\boldsymbol{a}$ might depend on the Gaussian noise.

**Assumption 2** (Hypothesis testing succeeds above threshold). *For $c \in (0, 1)$, define $\mathcal{A}_d(c) = \{\boldsymbol{a} \in \mathbb{R}^d : \|\boldsymbol{a}\| \leq c\, t_{\mathrm{alg}}\}$. We assume there exists $\delta, \eta_2 : \mathbb{N} \to \mathbb{R}$ (which we refer to as rates), and a polytime binary test function $\phi : \mathbb{R}^d \times \mathbb{R}_{\geq 0} \to \{0, 1\}$ such that:*

1. *(Sharp detection threshold) $\delta(d) = o_d(1)$.*

2. *For the process $(\boldsymbol{y}_t = t\boldsymbol{x} + \boldsymbol{W}_t)$, $\phi$ rejects with high probability:*

$$\int_{t_{\mathrm{alg}}(1+\delta)}^{\infty} \mathbb{P}(\phi(\boldsymbol{y}_t, t) = 0)\mathrm{d}t = O(\eta_2(d))\,.$$

3. *Uniformly over the set $\mathcal{A}_d(c)$, $\phi$ fails to reject with high probability. Namely:*

$$\mathbb{P}\left(\exists t \geq t_{\mathrm{alg}}(1 + \delta) \text{ such that } \sup_{\boldsymbol{a} \in \mathcal{A}_d(c)} \phi(\boldsymbol{a} + \boldsymbol{W}_t) = 1\right) = o(1)\,.$$

**Remark.** Since we try to state our theorem in the strongest form, Assumptions 1 and 2 do not take the same form as the information-computation gap (7). Nevertheless, it can be proven that (for a broad class of problems) these assumptions cannot hold unless an information-computation gap is present. We leave this point for future work.

**Discussion of the assumptions.** Before stating our main results, we address the validity of the assumptions.

- Assumption 1 is expected to hold for 'reasonable' polytime computable drifts in distributions $\mu$ with an information-computation gap (c.f. Subsection 3.2.1 and Proposition B.1). More precisely, in such problems no polytime computable estimator $\hat{\boldsymbol{m}}_0(\boldsymbol{y}_t, t)$ achieves correlation with the target $\boldsymbol{x}$ bounded away from zero, and therefore its loss is decreased by shrinking it to near zero.

- Assumption 2 concerns the existence of a certain efficient hypothesis test $\phi$. We can leverage the literature on information-computation gaps to determine the (conjectured) optimal efficient algorithm $\hat{\boldsymbol{m}}_\star(\boldsymbol{y}, t)$, and construct $\phi$ to test for large values of $\langle \hat{\boldsymbol{m}}_\star(\boldsymbol{y}, t), \boldsymbol{y} \rangle$. Specifically, $\phi(\boldsymbol{y}, t) = \mathbf{1}_{\langle \hat{\boldsymbol{m}}_\star(\boldsymbol{y}, t), \boldsymbol{y} \rangle \geq c_\star t}$, for some $c_\star \in (0, 1)$. The rationale is as follows: the maximum likelihood problem for the model $\boldsymbol{y}_t = t\boldsymbol{x} + \boldsymbol{B}_t$ is to find $\hat{\boldsymbol{x}}$ to maximize $\langle \hat{\boldsymbol{x}}, \boldsymbol{y} \rangle$. For $t$ above the algorithmic threshold, efficient estimators $\hat{\boldsymbol{m}}(\boldsymbol{y}_t, t)$ can approximate the (inefficient) MLE very well for the alternative model $\boldsymbol{y}_t$, leading to large values of $\langle \hat{\boldsymbol{m}}_\star(\boldsymbol{y}_t, t), \boldsymbol{y}_t \rangle \approx \langle \boldsymbol{x}, \boldsymbol{y}_t \rangle$ (in particular, it is $t - o(t)$). Note that the MLE is in turn very close to the true signal $\boldsymbol{x}$ in this regime.

  The worst-case Type I error guarantees can be checked directly with this $\phi$, and Condition 3 holds for the examples we listed in Subsection 3.2.2. Below, we employ a simple observation to show this: for the null model $\boldsymbol{a} + \boldsymbol{B}_t$, we have

$$\langle \hat{\boldsymbol{m}}_\star(\boldsymbol{a} + \boldsymbol{B}_t, t), \boldsymbol{a} + \boldsymbol{B}_t \rangle \leq \sup_{\boldsymbol{x}' \in \text{supp}(\mu)} \langle \boldsymbol{x}', \boldsymbol{a} + \boldsymbol{B}_t \rangle \leq \|\boldsymbol{a}\|_{\text{op}} + \sup_{\boldsymbol{x}' \in \text{supp}(\mu)} \langle \boldsymbol{x}', \boldsymbol{B}_t \rangle$$

  where $\text{supp}(\mu)$ is the support of $\mu$. For distributions with an information-computation gap, $\boldsymbol{x}'$ is often a "structured" vector (c.f. sparsity and examples in points (i) and (ii) of Subsection 3.2.2); as pure noise $\boldsymbol{B}_t$ does not favor any structure, we have

$$\sup_{\boldsymbol{x}' \in \text{supp}(\mu)} \langle \boldsymbol{x}', \boldsymbol{B}_t \rangle \ll t \Rightarrow \langle \hat{\boldsymbol{m}}_\star(\boldsymbol{a} + \boldsymbol{B}_t, t), \boldsymbol{a} + \boldsymbol{B}_t \rangle \leq ct_{\text{alg}} + o(t)$$

  for $c$ of Condition 3. Choosing $c_\star > c$, we have succeeded in constructing $\phi$.

We state our first main result. It stipulates that we can construct a polytime algorithm which has 'essentially' the same score-matching objective as $\hat{\boldsymbol{m}}_0$ yet yields bad samples.

**Theorem 1.** *Let $\mu$ be a probability measure supported on $\mathsf{B}^d(1)$ such that $\liminf_{d \to \infty} \int \|\boldsymbol{x}\| \, \mu(\mathrm{d}\boldsymbol{x}) = \alpha > 0$. Assume that there exist $t_{\text{alg}} = t_{\text{alg}}(d) > 0$, a drift $\hat{\boldsymbol{m}}_0 : \mathbb{R}^d \times \mathbb{R} \to \mathbb{R}$, and functions $\eta_1(d), \delta(d), \varepsilon_2(d) = o_d(1)$, such that following conditions hold: (i) $\sup_{\boldsymbol{y}, t} \|\hat{\boldsymbol{m}}_0(\boldsymbol{y}, t)\| \leq 1$. (ii) Assumption 1 holds with rate $\eta_1(d)$. (iii) Assumption 2 holds with rates $\delta(d), \eta_2(d)$.*

*Then there exists a modified drift $\hat{\boldsymbol{m}}$ such that*

**M1.** *$\hat{\boldsymbol{m}}(\cdot)$ can be evaluated in polynomial time.*

**M2.** *If $\boldsymbol{y}_t = t\boldsymbol{x} + \boldsymbol{B}_t$ is the true diffusion (equivalently given by (1)), then*

$$\int_0^\infty \mathbb{E}[\|\hat{\boldsymbol{m}}(\boldsymbol{y}_t, t) - \hat{\boldsymbol{m}}_0(\boldsymbol{y}_t, t)\|^2] \, \mathrm{d}t = O(\eta_1(d) \vee \eta_2(d)) \, .$$

**M3.** *For any step size $\Delta = \Delta_n > 0$, we have incorrect sampling:*

$$\inf_{t \in \mathbb{N} \cdot \Delta, t \geq (1+\delta)t_{\text{alg}}} W_1(\hat{\boldsymbol{m}}(\hat{\boldsymbol{y}}_t, t), \boldsymbol{x}) \geq \alpha - o_d(1) \, . \tag{8}$$

The proof is presented in Appendix G. The main idea is to let $\hat{\boldsymbol{m}}(\boldsymbol{y}, t)$ be $\hat{\boldsymbol{m}}_0(\boldsymbol{y}, t)\mathbf{1}_{\|\hat{\boldsymbol{m}}_0(\boldsymbol{y}, t)\| \leq \varepsilon}$ for $t \leq (1 - \gamma)t_{\text{alg}}$, and $\hat{\boldsymbol{m}}_0(\boldsymbol{y}, t)\phi(\boldsymbol{y}, t)$ for $t \geq (1 + \delta)t_{\text{alg}}$, with small constants $(\gamma, \varepsilon)$ and test $\phi$.

**Remark.** It makes sense to assume that $\|\hat{\boldsymbol{m}}_0(\cdot, \cdot)\| \leq 1$. Since $\text{supp}(\mu) \subseteq \mathsf{B}^d(1)$ and the latter is a convex set, projecting any $\hat{\boldsymbol{m}}_0$ onto this set yields a smaller MSE.

## 3.2 Example: Sampling low-rank matrices

We state two separate results for the probability distribution $\mu = \mu_{n,k}$ described in the introduction, depending on the scaling of $k$ with $n$: in Section 3.2.1 we assume $\sqrt{n} \ll k \ll n$; while in Appendix D we assume $k \ll \sqrt{n}$. Indeed, the nature of the problem changes at the threshold $k \asymp \sqrt{n}$.

A crucial role will be played by the following threshold

$$t_{\text{alg}}(n, k) := \begin{cases} k^2 \log\left(\dfrac{n}{k^2}\right) & \text{if } k \ll \sqrt{n} \\ \dfrac{n}{2} & \text{if } \sqrt{n} \ll k \ll n \end{cases} \tag{9}$$

It is expected that for $t \leq (1 - \delta)t_{\mathrm{alg}}(n, k)$ and $\delta$ any fixed constant, no polytime algorithm can estimate $\boldsymbol{x}$ significantly better than the estimator $\hat{\boldsymbol{m}}_{\mathrm{null}} = \mathbb{E}[\boldsymbol{x}] \approx \boldsymbol{0}$ for $k \ll n$ (see Conjecture 3.1).

Since $\|\boldsymbol{x}\|_F = 1$ for $\boldsymbol{x} \sim \mu$, the Bayes denoiser $\boldsymbol{m}(\boldsymbol{y}, t) = \boldsymbol{m}(\boldsymbol{y})$ does not depend on $t$ (this can be seen by Bayes rule). From now on, we refer to $\|\boldsymbol{x}\| = \|\boldsymbol{x}\|_F$ as the Frobenius norm.

### 3.2.1 MODERATELY SPARSE REGIME: $\sqrt{n} \ll k \ll n$

Assumption 1 states that, for $\boldsymbol{y}_t = t\boldsymbol{x} + \boldsymbol{W}_t$, the estimated drift $\hat{\boldsymbol{m}}_0(\boldsymbol{y}_t, t)$ should have small norm with high probability. This condition holds under the well-accepted Conjecture 3.1 below on information-computation gaps. In fact, a simple consequence of this conjecture is that any polytime $\hat{\boldsymbol{m}}$ *matching* this error must satisfy $\mathbb{E}\{\|\hat{\boldsymbol{m}}(\boldsymbol{y}_t, t)\|^2\} = o_n(1)$ (see Proposition C.1).

**Conjecture 3.1.** *For $\sqrt{n} \ll k \ll n$, there exists $\underline{k}_n \ll n$ such that the following holds for any $k = k_n$, with $\underline{k}_n \leq k_n \ll n$. Let $\{\hat{\boldsymbol{m}}_n\}_{n \geq 1}$, $\hat{\boldsymbol{m}}_n : \mathbb{R}^{n \times n} \times \mathbb{R} \to \mathbb{R}^{n \times n}$ be any sequence of polytime algorithms (polynomial time in $n$). Then for any $\delta > 0$, we have*

$$\inf_{t \leq (1-\delta)t_{\mathrm{alg}}} \mathbb{E}\{\|\hat{\boldsymbol{m}}_n(\boldsymbol{y}_t, t) - \boldsymbol{x}\|^2\} \geq 1 - o_n(1). \tag{10}$$

We refer to Ma & Wu (2015); Cai et al. (2017); Hopkins et al. (2017); Brennan et al. (2018); Schramm & Wein (2022); Kunisky et al. (2019) for evidence towards this conjecture. Next, we provide the following implication of Theorem 1, whose proof is in Appendix H.

**Corollary 3.2.** *Assume $\sqrt{n} \ll k \ll n$, so that $t_{\mathrm{alg}}(n, k) := n/2$ per (9). Let $\hat{\boldsymbol{m}}_0$ be an arbitrary poly-time algorithm such that $\sup_{\boldsymbol{y}, t} \|\hat{\boldsymbol{m}}_0(\boldsymbol{y}, t)\|_F \leq 1$ and Assumption 1 holds with rate $\eta_1$ such that $\eta_1 \ll n^{-D} \forall D > 0$. Then there exists an estimator $\hat{\boldsymbol{m}}$ such that:*

**M1.** *$\hat{\boldsymbol{m}}(\cdot)$ can be evaluated in polynomial time.*

**M2.** *If $\boldsymbol{y}_t = t\boldsymbol{x} + \boldsymbol{B}_t$ is the true diffusion (equivalently given by (1)), then, for every $D > 0$,*

$$\int_0^\infty \mathbb{E}[\|\hat{\boldsymbol{m}}(\boldsymbol{y}_t, t) - \hat{\boldsymbol{m}}_0(\boldsymbol{y}_t, t)\|^2] \, \mathrm{d}t = O(n^{-D}).$$

**M3.** *There exists $\delta = o_n(1)$ such that, for any step size $\Delta = \Delta_n > 0$, we have incorrect sampling:*

$$\inf_{t \in \mathbb{N} \cdot \Delta, t \geq (1+\delta)t_{\mathrm{alg}}} W_1(\hat{\boldsymbol{m}}(\hat{\boldsymbol{y}}_t, t), \boldsymbol{x}) \geq 1 - o_n(1). \tag{11}$$

To connect the last corollary with the introduction, we recall two facts from the literature on submatrix estimation: $(i)$ The Bayes estimator $\boldsymbol{m}(\boldsymbol{y}_t)$ achieves small MSE in a large interval above $t_{\mathrm{alg}}$ (Proposition 3.3); $(ii)$ No polytime estimator is expected to perform better than the null estimator below $t_{\mathrm{alg}}$ (Conjecture 3.1). Regarding $(i)$, we state a characterization of the Bayes optimal error. The proof is analogous to the main result in Butucea et al. (2015), which considers the case of asymmetric matrices. (For $k \leq n^a$, $a < 5/6$, see also Barbier et al. (2020).)

**Proposition 3.3** (Modification of Butucea et al. (2015)). *Let $\boldsymbol{m}(\boldsymbol{y})$ be the posterior mean estimator in Eq. (2). Assume $1 \ll k \ll n$, and define $t_{\mathrm{Bayes}}(n, k) := 2k \log(n/k)$. Then, for any $\delta > 0$, we have $\inf_{t \leq (1-\delta)t_{\mathrm{Bayes}}} \mathbb{E}\{\|\boldsymbol{m}(\boldsymbol{y}_t) - \boldsymbol{x}\|^2\} = 1 - o_n(1)$, $\sup_{t \geq (1+\delta)t_{\mathrm{Bayes}}} \mathbb{E}\{\|\boldsymbol{m}(\boldsymbol{y}_t) - \boldsymbol{x}\|^2\} = o_n(1)$.*

In other words, for $2k \log(n/k) \ll t \ll n$, the optimal estimator can estimate the signal $\boldsymbol{x}$ accurately, but we expect that no polytime algorithm can achieve the same.

### 3.2.2 VERY SPARSE REGIME: $k \ll \sqrt{n}$, AND OTHER EXAMPLES

In the very sparse regime $k \ll \sqrt{n}$, we prove a result similar to Corollary 3.2 (Corollary D.1).

**Other examples.** We mention a few examples where it is relatively straightforward to apply Theorem 1, following the blueprint in Corollary 3.2. $(i)$ Sampling low rank tensors, e.g. $\boldsymbol{x} = \boldsymbol{u}^{\otimes q} \in (\mathbb{R}^n)^{\otimes q}$, $q \geq 3$ when $\boldsymbol{u} \sim \mathrm{Unif}(\{+1/\sqrt{n}, -1/\sqrt{n}\}^n)$ or $\boldsymbol{u}$ is uniform on the unit sphere; the corresponding denoising problem is known as tensor PCA (Montanari & Richard, 2014) (in this case $d = n^q$). $(ii)$ Sampling elements of random linear subspaces of $\{0, 1\}^d$: $\boldsymbol{x} = \boldsymbol{G}\boldsymbol{u} \mod 2$,

where $\boldsymbol{G} \in \{0,1\}^{d \times \ell}$ is a fixed (known) uniformly random matrix and $\boldsymbol{u} \sim \mathrm{Unif}(\{0,1\}^{\ell})$, $\ell = rn$ for $r \in (0,1)$ a constant; the corresponding denoising problem amounts to decoding random linear codes (Richardson & Urbanke, 2008; Ghazi & Lee, 2017) (this example fits our framework after centering). We give two classes of examples for which applying Theorem 1 requires additional technical work (defer to future publications): $(iii)$ Sampling from Bayesian posteriors, e.g. posterior of a low-rank plus noise estimation problem that presents an information-computation gap (Lelarge & Miolane, 2017; Montanari & Wu, 2023; Ghio et al., 2024); $(iv)$ Sampling solutions of random constraint satisfaction problems (Montanari et al., 2007; Ghio et al., 2024).

## 4 REDUCTION OF ESTIMATION TO DIFFUSION-BASED SAMPLING

To complement previous results, we prove a general reduction: if diffusion sampling can be performed in polynomial time with sufficient accuracy, then we can perform also denoising. The contrapositive of this statement aligns with results in previous sections.

To avoid unessential complications, in this section we assume $\mu$ to be supported on the unit sphere $\mathbb{S}^{d-1} = \{\boldsymbol{x} : \|\boldsymbol{x}\| = 1\}$. We denote by $\mathrm{P}_{\boldsymbol{y}}^T$ the law of $(\boldsymbol{y}_t)_{0 \le t \le T}$ where $\boldsymbol{y}_t$ given by Eq. (1) and by $\mathrm{P}_{\hat{\boldsymbol{y}}}^{T,\Delta}$ the law of $(\hat{\boldsymbol{y}}_t)_{0 \le t \le T}$, which is the discretized diffusion trajectory defined in (4) (interpolated linearly outside $\mathbb{N} \cdot \Delta$).

It is further useful to define $\overline{\mathrm{P}}_{\hat{\boldsymbol{y}}}^{T,\Delta}$ to be the law of the SDE interpolating that of (4):

$$\mathrm{d}\hat{\boldsymbol{y}}_t = \hat{\boldsymbol{m}}(\hat{\boldsymbol{y}}_{\lfloor t \rfloor_\Delta}, \lfloor t \rfloor_\Delta)\,\mathrm{d}t + \mathrm{d}\boldsymbol{B}_t\,, \tag{12}$$

where $\lfloor t \rfloor_\Delta := \max\{s \in \mathbb{N} \cdot \Delta : s \le t\}$.

**Theorem 2.** *Assume that $\hat{\boldsymbol{m}}(\,\cdot\,,\,\cdot\,)$ has complexity $\chi$ and that for any $T \le \theta d$, $D_{\mathrm{KL}}(\overline{\mathrm{P}}_{\hat{\boldsymbol{y}}}^{T,\Delta} \| \mathrm{P}_{\boldsymbol{y}}^T) \le \varepsilon$*

*Then for any $\sigma > 0$ there exists an algorithm a randomized algorithm $\hat{\boldsymbol{m}}_+$ with complexity $(N\chi \cdot T/\Delta)$ that approximates the posterior expectation:*

$$\mathbb{E}\big\{\|\hat{\boldsymbol{m}}_+(\boldsymbol{y}) - \boldsymbol{m}(\boldsymbol{y})\|^2\big\} \le 2\overline{\varepsilon} + 2N^{-1}\,. \tag{13}$$

*Here $\overline{\varepsilon} := \sqrt{2\varepsilon} + \varepsilon_0(\theta)$ and $\varepsilon_0(\theta) := \mathbb{E}\|\mathrm{P}_{\boldsymbol{x}|\boldsymbol{y}} - \mathsf{N}(\boldsymbol{0}, (\theta d)^{-1}\boldsymbol{I}_d) * \mathrm{P}_{\boldsymbol{x}|\boldsymbol{y}}\|_{\mathrm{TV}}$ is the expected TV distance between $\mathrm{P}_{\boldsymbol{x}|\boldsymbol{y}}$ and the convolution of $\mathrm{P}_{\boldsymbol{x}|\boldsymbol{y}}$.*

The proof of this result is presented in Appendix T, along with a modification.

## 5 ALL LIPSCHITZ POLYTIME ALGORITHMS FAIL

In Section 3 (Theorem 1 and Corollary 3.2) we proved that there exist near-optimizers of the score matching objective that perform poorly. However, we did not rule out the possibility that the optimal (in the sense of score-matching) polytime drift $\hat{\boldsymbol{m}}$ will perform well. We next show that this is not the case, under an additional assumption, namely that the drift $\hat{\boldsymbol{m}}(\,\cdot\,;t)$ is Lipschitz continuous for $t \ge (1+\delta)t_{\mathrm{alg}}$. Proof is given in Appendix W. (We assume the Lipschitz constant to be $C/t$, because the input of the denoiser is $\boldsymbol{y}_t = t\boldsymbol{x} + \boldsymbol{W}_t$, and hence the two $t$-dependent factors cancel.)

**Theorem 3.** *Let $\mu$ be supported on $\mathsf{B}^d(1) = \{\boldsymbol{x} : \|\boldsymbol{x}\| \le 1\}$, $\int \boldsymbol{x}\,\mu(\mathrm{d}\boldsymbol{x}) = \boldsymbol{0}$, and $\liminf_{d \to \infty} \int \|\boldsymbol{x}\|\mu(\mathrm{d}\boldsymbol{x}) = \alpha > 0$. Let $\hat{\boldsymbol{m}} : \mathbb{R}^d \times \mathbb{R}_{\ge 0} \to \mathbb{R}^d$ be a polytime denoiser such that $\sup_{\boldsymbol{y},t} \|\hat{\boldsymbol{m}}(\boldsymbol{y},t)\| \le 1$ (below $\boldsymbol{W}_t$ is a standard BM):*

1. *$\hat{\boldsymbol{m}}$ is nearly optimal, namely for $\boldsymbol{y}_t = t\boldsymbol{x} + \boldsymbol{W}_t$, and every $\gamma > 0$*

$$\sup_{t \le (1-\gamma)t_{\mathrm{alg}}} \Big|\mathbb{E}\big\{\|\hat{\boldsymbol{m}}(\boldsymbol{y}_t, t) - \boldsymbol{x}\|^2\big\} - \mathbb{E}[\|\boldsymbol{x}\|^2]\Big| = o(t_{\mathrm{alg}}^{-1})\,, \tag{14}$$

$$\sup_{t \ge (1+\gamma)t_{\mathrm{alg}}} \mathbb{E}\big\{\|\hat{\boldsymbol{m}}(\boldsymbol{y}_t, t) - \boldsymbol{x}\|^2\big\} = o(1)\,, \tag{15}$$

*and that for every $t \ge 0, c \in [-1,1]$, $\mathbb{E}[\|\hat{\boldsymbol{m}}(\boldsymbol{y}_t, t) - \boldsymbol{x}\|^2] \le \mathbb{E}[\|c\hat{\boldsymbol{m}}(\boldsymbol{y}_t, t) - \boldsymbol{x}\|^2] + o(t_{\mathrm{alg}}^{-1})$.*

2. *($\hat{m}$ is small on pure noise.) For some $\delta = o(1)$, and every $\Delta = O(1)$, we have*

$$\Delta \cdot \sum_{t \in \mathbb{N} \cdot \Delta \cap [t_{\mathrm{alg}}(1+\delta), \infty]} \mathbb{E}[\|\hat{m}(W_t, t)\|^2] = o(1)$$

3. *$\hat{m}(\,\cdot\,, t)$ is $C/t$-Lipschitz for some constant $C$ and all $t \geq (1+\delta)t_{\mathrm{alg}}$.*

*Then, for every constant $C_0 > 0$ and step size $\Delta = O(1)$:*

$$\inf_{t \in \mathbb{N}\Delta \cap [0, C_0 t_{\mathrm{alg}}]} W_1(\hat{m}(\hat{y}_t, t), x) \geq \alpha - o(1)$$

**Remark.** The sum in Condition 2 of Theorem 3 is a discretized integral; when $\Delta$ is small enough, this is essentially equivalent to stating that

$$\int_{t_{\mathrm{alg}}(1+\delta)}^{\infty} \mathbb{E}[\|\hat{m}(W_t, t)\|^2] \mathrm{d}t = o(1)$$

For applications of this theorem (c.f. Corollary 5.1, we have an upper bound $\mathbb{E}[\|\hat{m}(W_t, t)\|^2] \leq c_n(t)$ with $c_n(t)$ decreasing (for $t \geq t_{\mathrm{alg}}(1+\delta)$), so that for every $\Delta$,

$$\Delta \cdot \sum_{t \in \mathbb{N} \cdot \Delta \cap [t_{\mathrm{alg}}(1+\delta), \infty]} \mathbb{E}[\|\hat{m}(B_t, t)\|^2] \leq \Delta \cdot c_n(t_{\mathrm{alg}}(1+\delta)) + \int_{t_{\mathrm{alg}}(1+\delta)}^{\infty} c_n(t)\mathrm{d}t = o_n(1)$$

so that the specific value of $\Delta$ does not matter, as long as $\Delta = O(1)$.

We apply the above theorem to our running example of sampling sparse low-rank matrices. In order to make sure that condition 2 in the theorem is verified, we introduce a variant $\overline{\mu}_{n,k}$ of $\mu_{n,k}$ (all conclusions stated for $\mu$, e.g., Theorem 1, Corollaries 3.2, D.1 hold for $\overline{\mu}_{n,k}$ as well.) Letting $\mu_{n,k}^0$ be the centered version of $\mu_{n,k}$; we define $\overline{\mu}_{n,k} = \frac{1}{2}\delta_0 + \frac{1}{2}\mu_{n,k}^0$. In words, with probability $1/2$ we let $x = 0$ and with probability $1/2$ we draw $x = \tilde{x} - \mathbb{E}[\tilde{x}]$, $\tilde{x} \sim \mu_{n,k}$, a sparse rank-one matrix, as in previous sections. As mentioned, this mixture distribution $\overline{\mu}_{n,k}$ is mainly to satisfy condition 2 of Theorem 3. Indeed, we have the following decomposition

$$\mathbb{E}_{x \sim \overline{\mu}_{n,k}}[\|\hat{m}(y_t, t) - x\|^2] = \frac{1}{2}\mathbb{E}_{x \sim \mu_{n,k}^0}[\|\hat{m}(y_t, t) - x\|^2] + \frac{1}{2}\mathbb{E}[\|\hat{m}(W_t, t)\|^2],$$

which shows that, to get $\hat{m}(y_t, t) \approx x$ under the mixture distribution, we also need $\hat{m}(W_t, t) \approx 0$. More concretely, we can get explicit rates on $\mathbb{E}[\|\hat{m}(W_t, t)\|^2]$ for $t$ above $t_{\mathrm{alg}}$ by enforcing that $\hat{m}$ *cannot* be improved by multiplying by certain hypothesis tests. The full result is as follows.

**Corollary 5.1.** *Assume $\underline{k}_n$ exists as in Conjecture 3.1. Let $k = k_n$ be such that $\underline{k}_n \vee \sqrt{n} \leq k_n \ll n$ (moderately sparse regime). Let $\hat{m}_n$ be a polytime denoiser such that for some $\delta = o_n(1)$, and every fixed constant $\gamma > 0$:*

1. *$\hat{m}_n$ is nearly optimal, namely (for $y_t = tx + W_t$, $W_t$ standard BM)*

$$\sup_{t \leq (1-\gamma)t_{\mathrm{alg}}} \left| \mathbb{E}\{\|\hat{m}(y_t, t) - x\|^2\} - \mathbb{E}[\|x\|^2] \right| = o_n(n^{-1}), \tag{16}$$

$$\sup_{t \geq (1+\gamma)t_{\mathrm{alg}}} \mathbb{E}\{\|\hat{m}(y_t, t) - x\|^2\} = o_n(1), \tag{17}$$

*and further, for any $t \geq (1+\delta)t_{\mathrm{alg}}$, the MSE of $\hat{m}_n$ smaller or equal than the MSE of $c(\lambda_1(y_t))\hat{m}_n(y_t, t)$ for any polytime function $c(\,\cdot\,)$ of the maximum eigenvalue of $(y_t + y_t^{\mathsf{T}})/\sqrt{2}$, and than the MSE of $P_{\mathsf{B}}\hat{m}_n$, for $P_{\mathsf{B}}$ the projection onto the unit ball.*

2. *$\hat{m}_n(\,\cdot\,, t) : \mathbb{R}^{n \times n} \to \mathbb{R}^{n \times n}$ is $C/t$-Lipschitz for some constant $C$ and all $t \geq (1+\delta)t_{\mathrm{alg}}$.*

*Then, for every constant $C_0 > 0$, and step size $\Delta = O(1)$:*

$$\inf_{t \in \mathbb{N}\cdot\Delta \cap [0, C_0 n]} W_1(\hat{m}(\hat{y}_t, t), x) \geq \frac{1}{2} - o_n(1). \tag{18}$$

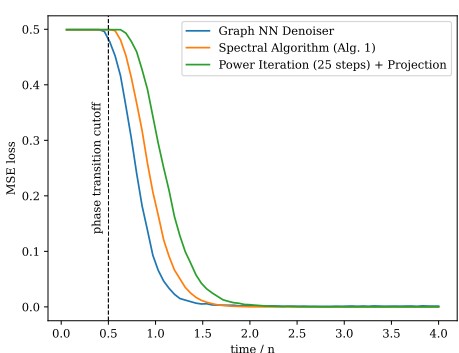 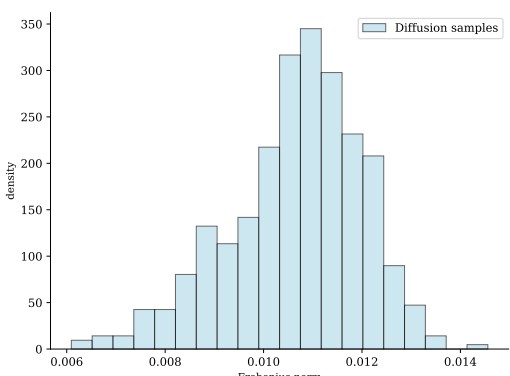

Figure 1: Generating sparse rank-one matrices $\boldsymbol{x} \sim \tilde{\mu}_{n,k}$ using denoising diffusions, for $n = 350$, $k = 20$. Left: MSE of various denoisers (vertical line corresponds to the algorithmic threshold $t_{\text{alg}}$.) Right: Frobenius norms of generated samples.

The proof is given in Appendix X. We note that the error of polytime denoisers in (16) (and sampling error of Eq. 18) is $1/2$ instead of $1$ because the best constant denoiser achieves error $1/2$.

Corollary 5.1 does not rule out the possibility that there exists a near-optimizer of score matching that violates the Lipschitz condition and samples well. However, for $t \geq (1 + \delta)t_{\text{alg}}$ accurate estimation is possible with Lipschitz algorithms, and indeed many natural methods are in this class (e.g. neural nets with bounded number of layers and suitable operator norm bounds on the weights.)

## 6 NUMERICAL ILLUSTRATION

The theory developed in the previous section yields a concrete prediction of the failure mode of DS when applied to the distribution $\tilde{\mu}_{n,k} = (1/2)\delta_{\mathbf{0}} + (1/2)\mu_{n,k}$ (with $\mu_{n,k}$ the law of $\boldsymbol{x} = \boldsymbol{u}\boldsymbol{u}^{\mathsf{T}}$, $\boldsymbol{u} \sim \text{Unif}(B_{n,k})$). Namely (for large $n$, and $\sqrt{n} \ll k \ll n$):

**1.** Given sufficient model complexity and training samples, we expect the learnt denoiser $\hat{\boldsymbol{m}}_n(t, \cdot)$ to achieve MSE close to $1/2$ for $t < (1 - \delta)t_{\text{alg}}$, and close to $0$ for $t > (1 + \delta)t_{\text{alg}}$.

**2.** We expect DS based on such a denoiser to generate samples concentrated around $\mathbf{0}$.

We tested these predictions in a numerical experiment. We considered three polytime denoisers:

$(a)$ The spectral-plus-projection denoiser of Algorithm 2;
$(b)$ A modification of the latter whereby the eigenvector calculation is replaced by 25 iterations of power method;
$(c)$ A learned graph neural network (GNN) (Scarselli et al., 2008; Kipf & Welling, 2016).

We carry out experiments with denoiser $(b)$ because $\ell$ iterations of power method can be approximated by an $\ell$-layers GNN. Hence, method $(b)$ provides a baseline for GNN denoisers.

Figure 1, left frame, reports the MSE achieved by the three denoisers $(a)$, $(b)$, $(c)$ as a function of $t/n$, for $n = 350$, $k = 20$. As GNNs are permutation-equivariant, we are training on $\approx 3\%$ of all possible outcomes, for $n = 350$ and $k = 20$. We observe that the GNN denoiser outperforms both the spectral algorithm and its approximation via power iteration. However, none of the three approaches can overcome the barrier at $t_{\text{alg}} = n/2$, while they perform reasonably well above that threshold. This confirms the prediction at point **1** above.

On the right, we plot the histogram of Frobenius norms of samples generated with the GNN denoiser. These values are close to $0$, which confirms the prediction at point **2** above. By using $\|\cdot\|_F$ as a 1-Lipschitz test function, we obtain that the Wasserstein distance between diffusion samples and the target distribution is at least $0.48$ (the asymptotic prediction from theory is $0.50$).

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

## A  NOTATIONS

Throughout the paper it will be understood that we are considering sequences of problems indexed by $n$, where $\boldsymbol{x} \in \mathbb{R}^{n \times n}$ and the sparsity index $k = k_n$ diverges as well. We write $f(n) \ll g(n)$ or $f(n) = o(g(n))$ if $f(n)/g(n) \to 0$ and $f(n) \lesssim g(n)$ or $f(n) = O(g(n))$ if $f(n)/g(n) \leq C$ for a constant $C$. Finally $f(n) = \Theta(g(n))$ or $f(n) \asymp g(n)$ if $1/C \leq f(n)/g(n) \leq C$.

We write $\boldsymbol{W} \sim \mathsf{GOE}(n)$ if $\boldsymbol{W} = \boldsymbol{W}^{\mathsf{T}}$ is a random symmetric matrix with $(W_{ij})_{i \leq j \leq n}$ independent entries $W_{ii} \sim \mathsf{N}(0,2)$, and $W_{ij} \sim \mathsf{N}(0,1)$ for $i < j$. We say that $(\boldsymbol{W}_t : t \geq 0)$ is a $\mathsf{GOE}(n)$ process if $\boldsymbol{W}_t \in \mathbb{R}^{n \times n}$ is a symmetric matrix with entries above and on the diagonal $(W_t(i,j) : i < j \leq n; W_t(i,i)/\sqrt{2} : i \leq n; t \geq 0)$ forming a collection of $n(n+1)/2$ independent BMs.

We use $C, C_i, c_i, \ldots$ to denote absolute constants, whose value can change from line to line.

## B  EQUIVALENCE TO THE TIME-REVERSAL FORMULATION

In this section, we explain the relationship between the (time-forward) formulation of Eqs. (1), (2) and the time-reversal setup of Song et al. (2021); Song & Ermon (2019). Regarding the latter, we recall the Ornstein-Uhlenbeck process:

$$\mathrm{d}\boldsymbol{Z}_s = -\boldsymbol{Z}_s \mathrm{d}s + \sqrt{2}\mathrm{d}\boldsymbol{B}_s \tag{19}$$

with $\boldsymbol{Z}_0 = \boldsymbol{x} \sim \mu$, the target distribution and $(\boldsymbol{B}_s)$ standard Brownian motion. Marginally, we have $\boldsymbol{Z}_s \overset{d}{=} e^{-s}\boldsymbol{x} + \sqrt{1 - e^{-2s}}\boldsymbol{g}$, with $\boldsymbol{g} \sim \mathcal{N}(\boldsymbol{0}, \boldsymbol{I})$ independent of $\boldsymbol{x}$. At large time-horizon $S \leq \infty$, $\boldsymbol{Z}_S$ approximately follows $\mathcal{N}(\boldsymbol{0}, \boldsymbol{I})$, which is easy to sample from.

We obtain a sampling process by time-reversing the SDE of Eq. (19). Specifically, let $T \leq \infty$ be another large time-horizon, and consider a time-change function $\mathfrak{t} : [0, S] \to [0, T]$ strictly decreasing, continuous, such that $\mathfrak{t}(0) = T, \mathfrak{t}(S) = 0$. Let $\mathfrak{s}$ be its inverse. Then, by time-reversing, we mean the process $(\boldsymbol{Z}_{\mathfrak{s}(t)})_{t \in [0,T]}$, starting with the initial condition $\boldsymbol{Z}_S$.

We can write a time-forward process $\overline{\boldsymbol{Y}}_t = \boldsymbol{Z}_{\mathfrak{s}(t)}$. It is known from Haussmann & Pardoux (1986) and Tweedie's formula that $(\overline{\boldsymbol{Y}}_t)$ follows the SDE:

$$\mathrm{d}\overline{\boldsymbol{Y}}_t = \overline{\mathbf{F}}(t, \overline{\boldsymbol{Y}}_t)\mathrm{d}t + \sqrt{2|\mathfrak{s}'(t)|}\mathrm{d}\overline{\mathbf{B}}_t \tag{20}$$

where the drift is given by

$$\overline{\mathbf{F}}(t, \boldsymbol{y}) = \left(\boldsymbol{y} + \frac{2}{1 - e^{-2\mathfrak{s}(t)}}\left\{\mathbb{E}[e^{-\mathfrak{s}(t)}\boldsymbol{x}|\boldsymbol{Z}_{\mathfrak{s}(t)} = \boldsymbol{y}] - \boldsymbol{y}\right\}\right)|\mathfrak{s}'(t)|$$

Note the resemblance between the conditional expectation $\mathbb{E}[e^{-\mathfrak{s}(t)}\boldsymbol{x}|\boldsymbol{Z}_{\mathfrak{s}(t)} = \boldsymbol{y}]$ of the previous display and the definition of $\boldsymbol{m}$ in Eq. (2). Now, we take $S = T = \infty$, and a specific time-change $\mathfrak{t}(s) = 1/(e^{2s} - 1)$. The resulting process $\overline{\boldsymbol{Y}}_t$ has initial observation $\overline{\boldsymbol{Y}}_0 = \boldsymbol{Z}_\infty \sim \mathcal{N}(\boldsymbol{0}, \boldsymbol{I})$, and Eq. (20) becomes

$$\mathrm{d}\overline{\boldsymbol{Y}}_t = -\frac{1}{t}\overline{\boldsymbol{Y}}_t + \frac{1}{\sqrt{t(1+t)}}\boldsymbol{m}(\sqrt{t(1+t)}\overline{\boldsymbol{Y}}_t, t)\mathrm{d}t + \frac{1}{\sqrt{t(1+t)}}\mathrm{d}\overline{\mathbf{B}}_t$$

Now, letting $\boldsymbol{y}_t = \sqrt{t(1+t)}\overline{\boldsymbol{Y}}_t$, employing Ito's lemma on $\overline{\boldsymbol{Y}}_t$ and the function $f(\boldsymbol{y}, t) = \sqrt{t(1+t)}\boldsymbol{y}$, we obtain that $\boldsymbol{y}_t$ follows the SDE of Eq. (1). The fact that we can write $\boldsymbol{y}_t = t\boldsymbol{x} + \boldsymbol{B}_t$ as a process comes from the fact that $\boldsymbol{y}_t = \sqrt{t(1+t)}\boldsymbol{Z}_{\mathfrak{s}(t)}$. This connection between denoising diffusions and stochastic localization stems from the specific time-change formula of $\mathfrak{t}(\cdot)$.

The treatment in Song et al. (2021) uses a finite time horizon $S$ and consider the linear time-change formula $\mathfrak{t}(s) = S - s$. Then, $(\overline{\boldsymbol{Y}}_s)$ follows the SDE of Eq. (20) with $|\mathfrak{s}'(\cdot)| = 1$ and drift

$$\overline{\mathbf{F}}(s, \boldsymbol{y}) = \boldsymbol{y} + 2\nabla_x p_{S-s}(\boldsymbol{y}) = \boldsymbol{y} + \frac{2}{1 + e^{-2(S-s)}}\left\{\mathbb{E}[e^{-(S-s)}\boldsymbol{x}|\boldsymbol{Z}_{S-s} = \boldsymbol{y}] - \boldsymbol{y}\right\}$$

It is important to note that other time-change functions $\mathfrak{t}(\cdot)$ (and thus $\mathfrak{s}(\cdot)$) will not change our conclusions. The computational bottleneck of recovering $\boldsymbol{x}$ from noisy observations $\alpha(t)\boldsymbol{x} + \beta(t)\boldsymbol{g}$ for some functions $\alpha(\cdot), \beta(\cdot)$ depends only on $\alpha(t)/\beta(t)$, which is continuous and increasing with $t$ from $0$ to $\infty$; moreover, our proof technique purely relies on controlling the discretized/generated SDE up to the computational threshold, which then should apply to other time-change functions. We chose the formula $\mathfrak{t}(s) = 1/(e^{2s} - 1)$ for notational convenience.

## C  A SIMPLE CONSEQUENCE OF CONJECTURE 3.1

We state and prove the following proposition.

**Proposition C.1.** *Suppose that Conjecture 3.1 holds for a distribution $\mu$ with $\mathbb{E}_{\boldsymbol{x}\sim\mu}[\|\boldsymbol{x}\|^2] = 1$. Then for any sequence of times $t = t_n \leq (1 - \delta)t_{\mathrm{alg}}$,*

$$\mathbb{E}[\|\hat{\boldsymbol{m}}(\boldsymbol{y}_t, t) - \boldsymbol{x}\|^2] = 1 - o(1) \Rightarrow \mathbb{E}[\|\hat{\boldsymbol{m}}(\boldsymbol{y}_t, t)\|^2] = o(1)$$

*In words, if $\hat{\boldsymbol{m}}$ is (near)-optimal in score matching for $t \leq (1 - \delta)t_{\mathrm{alg}}$, then $\|\hat{\boldsymbol{m}}(\boldsymbol{y}_t, t)\|$ is small.*

Before giving the proof, we remark that the full Conjecture 3.1 is not needed. It suffices for $\hat{\boldsymbol{m}}$ to have a weaker property; namely, that for any fixed constants $c \in [-1, 1]$ and $\delta \in (0, 1)$,

$$\inf_{t \leq (1-\delta)t_{\mathrm{alg}}} \mathbb{E}[\|c\hat{\boldsymbol{m}}(\boldsymbol{y}_t, t) - \boldsymbol{x}\|^2] \geq 1 - o(1)$$

*Proof.* Fix $c \in [-1, 1]$ to be a constant chosen later. From the property of $\hat{\boldsymbol{m}}$, we get from Cauchy-Schwarz that

$$\frac{1}{2}\mathbb{E}[\|\hat{\boldsymbol{m}}(\boldsymbol{y}_t, t)\|^2] - \mathbb{E}[\|\boldsymbol{x}\|^2] \leq 1 - o(1) \Rightarrow \mathbb{E}[\|\hat{\boldsymbol{m}}(\boldsymbol{y}_t, t)\|^2] \leq 4 - o(1)$$

We use Conjecture 3.1 for the sequence of estimators $c\hat{\boldsymbol{m}}$, which states that uniformly over $t \leq (1 - \delta)t_{\mathrm{alg}}$:

$$\mathbb{E}[\|c\hat{\boldsymbol{m}}(\boldsymbol{y}_t, t) - \boldsymbol{x}\|^2] \geq 1 - o(1) \Rightarrow c^2\mathbb{E}[\|\hat{\boldsymbol{m}}(\boldsymbol{y}_t, t)\|^2] - 2c\mathbb{E}[\langle\hat{\boldsymbol{m}}(\boldsymbol{y}_t, t), \boldsymbol{x}\rangle] \geq -o(1)$$

Suppose for sake of contradiction, that $\limsup_{n\to\infty} |\mathbb{E}[\langle\hat{\boldsymbol{m}}(\boldsymbol{y}_t, t), \boldsymbol{x}\rangle]| \geq \beta > 0$. Without loss of generality, we consider the subsequence $(n_k)$ such that $\mathbb{E}[\langle\hat{\boldsymbol{m}}(\boldsymbol{y}_t, t), \boldsymbol{x}\rangle] \geq \beta/2$. Along this subsequence, we have

$$4c^2 - c\beta \geq -o(1)$$

for all $c \in [-1, 1]$. However, we know that this is not true for $c > 0$ small enough; specifically, take $c < \beta/8$ so that we have $-c\beta/2 \geq -o(1)$, contradiction. Hence $\mathbb{E}[\langle\hat{\boldsymbol{m}}(\boldsymbol{y}_t, t), \boldsymbol{x}\rangle] = o(1)$. From the property of $\hat{\boldsymbol{m}}$, we obtain the conclusion. $\qquad\square$

## D  APPLYING THEOREM 1 TO VERY SPARSE MATRICES

As mentioned in Section E.3, we state and prove an analogous version of Corollary 3.2 in the very sparse case. One different aspect from the moderate case is that $k$ can be smaller asymptotically: in particular, $k$ can be sub-polynomial in $n$. Therefore, we first give a modification of Assumption 1.

**Assumption 3.** *Consider $\underline{k}_n \ll k \ll n$ for $\underline{k}_n$ in Conjecture 3.1. Let $\boldsymbol{y}_t = t\boldsymbol{x} + \boldsymbol{W}_t$ for $(\boldsymbol{W}_t)$ sBM independent of $\boldsymbol{x}$. Then a near-optimal estimator $\hat{\boldsymbol{m}}_0(\boldsymbol{y}, t)$ in score-matching satisfies: for every pair $(\gamma, \varepsilon) \in (0, 1)$,*

$$\int_0^{(1-\gamma)t_{\mathrm{alg}}} \mathbb{P}\left(\|\hat{\boldsymbol{m}}_0(\boldsymbol{y}_t, t)\| \geq \varepsilon\right) \mathrm{d}t = O(k^{-D})$$

*for every fixed $D > 0$.*

**Corollary D.1.** *Assume $(\log n)^2 \ll k \ll n$, so that $t_{\mathrm{alg}}(n, k) := k^2\log(n/k^2)$. Let $\hat{\boldsymbol{m}}_0$ be an arbitrary poly-time algorithm such that $\sup_{\boldsymbol{y},t} \|\hat{\boldsymbol{m}}_0(\boldsymbol{y}, t)\|_F \leq 1$ and Assumption 3 holds. Then there exists an estimator $\hat{\boldsymbol{m}}$ such that*

M1. *$\hat{\boldsymbol{m}}(\cdot)$ can be evaluated in polynomial time.*

**M2.** *If $\boldsymbol{y}_t = t\boldsymbol{x} + \boldsymbol{B}_t$ is the true diffusion (equivalently given by (1)), then, for every $D > 0$,*

$$\int_0^\infty \mathbb{E}[\|\hat{\boldsymbol{m}}(\boldsymbol{y}_t, t) - \hat{\boldsymbol{m}}_0(\boldsymbol{y}_t, t)\|^2]\,\mathrm{d}t = O(k^{-D})\,.$$

**M3.** *There exists $\delta = o_n(1)$ such that, for any step size $\Delta = \Delta_n > 0$, we have incorrect sampling:*

$$\inf_{t \in \mathbb{N}\cdot\Delta,\, t \geq (1+\delta)t_{\mathrm{alg}}} W_1(\hat{\boldsymbol{m}}(\hat{\boldsymbol{y}}_t, t), \boldsymbol{x}) \geq 1 - o_n(1)\,. \tag{21}$$

*Proof.* By the blueprint Theorem 1, we find (a sequence of) hypothesis tests $\phi(\boldsymbol{y}, t)$ indexed by $t$ such that Assumption 2 holds. We choose a rate $\delta_n = o_n(1)$ slow enough, and $\varepsilon_n$ be the resulting sequence, such that Proposition I.1 holds. We now describe $\phi(\boldsymbol{y}, t)$, based on Algorithm 1, from time $t = (1 + \delta)t_{\mathrm{alg}} = (1 + \delta)k^2 \log(n/k^2)$ upto $t = n$:

- Let $s = \sqrt{(1 + \varepsilon_n)\log(n/k^2)}$. Compute $\boldsymbol{y}_+ = \boldsymbol{y} + \sqrt{\varepsilon_n t}\boldsymbol{g}$ and $\boldsymbol{y}_- = \boldsymbol{y} - \sqrt{t/\varepsilon_n}\boldsymbol{g}$, with $\boldsymbol{g} \sim \mathsf{N}(\boldsymbol{0}, \boldsymbol{I})$. Then, compute $\boldsymbol{A}_+ = (\boldsymbol{y}_+ + \boldsymbol{y}_+^{\mathsf{T}})/(2\sqrt{t})$, and $\boldsymbol{A}_- = (\boldsymbol{y}_- + \boldsymbol{y}_-)/(2\sqrt{t})$.

- Let $\boldsymbol{v}$ be the leading eigenvector of $\eta_s(\boldsymbol{A}_+)$. Then, let $\hat{\boldsymbol{v}} = \boldsymbol{A}_- \boldsymbol{v}$. Let $\hat{S}$ be the set of $k$ indices of $\hat{\boldsymbol{v}}$ with largest magnitude, and compute $\boldsymbol{w}$ such that $w_i = (1/\sqrt{k})\operatorname{sign}(\hat{v}_i)\boldsymbol{1}_{i \in \hat{S}}$.

- Finally, reject iff $\langle \boldsymbol{w}, \boldsymbol{y}\boldsymbol{w} \rangle \geq \beta t$, for some $1 > \beta > c$.

From Proposition I.1, we know that

$$\sup_{t \geq (1+\delta)t_{\mathrm{alg}}} \mathbb{P}(\boldsymbol{w}(\boldsymbol{y}_t, t) \neq \boldsymbol{x}) \ll n^{-D}$$

for every $D > 0$. On the event that $\boldsymbol{w}(\boldsymbol{y}_t, t) = \boldsymbol{x}$, we get that

$$\langle \boldsymbol{w}(\boldsymbol{y}_t, t), \boldsymbol{y}_t\boldsymbol{w}(\boldsymbol{y}_t, t) \rangle = t + \langle \boldsymbol{w}(\boldsymbol{y}_t, t), \boldsymbol{W}_t\boldsymbol{w}(\boldsymbol{y}_t, t) \rangle \geq t - \sup_{\boldsymbol{v}: \|\boldsymbol{v}\|_0 = k, v_i \in \{0, \pm 1/\sqrt{k}\}} \langle \boldsymbol{v}, \boldsymbol{W}_t\boldsymbol{v} \rangle$$

for $(\boldsymbol{W}_t)$ standard Brownian motion. From Lemma H.1, we get that with error probability at most $\binom{n}{k}^{-D}$ for some $D$, we get that

$$\langle \boldsymbol{w}(\boldsymbol{y}_t, t), \boldsymbol{y}_t\boldsymbol{w}(\boldsymbol{y}_t, t) \rangle = t + \langle \boldsymbol{w}(\boldsymbol{y}_t, t), \boldsymbol{W}_t\boldsymbol{w}(\boldsymbol{y}_t, t) \rangle \geq t - C\sqrt{t \log \binom{n}{k}} = t(1 - o(1))$$

Therefore, we obtain that for $\beta < 1$,

$$\sup_{t \geq (1+\delta)t_{\mathrm{alg}}} \mathbb{P}(\phi(\boldsymbol{y}_t, t) = 0) \ll n^{-D}$$

for any $D > 0$. After time $t = n$, we use the same tests $\phi_1, \phi_2$ as documented in the proof of Corollary 3.2, as $t = n > (1 + \delta)(n/2)$, where $n/2$ is the algorithmic threshold of the moderately sparse case. The reason we can do this is that the spectral method, as in Algorithm 2, works even when $k \ll \sqrt{n}$ (although the threshold for this algorithm is asymptotically worse than $k^2 \log(n/k^2)$). Furthermore, the size of the perturbation $\boldsymbol{a} \in \mathcal{A}_d(c)$ is at most $\|\boldsymbol{a}\| \leq ct_{\mathrm{alg}} = ck^2 \log(n/k^2) \ll c(n/2)$.

Consequently, the first condition of Assumption 2 holds with rate $n^{-D}$ for every $D > 0$. To deal with the second condition, note simply that $\boldsymbol{w}$ is a $k$-sparse vector. A close inspection of the proof of Corollary 3.2 shows that it does not really matter how $\boldsymbol{w}$ is computed; the main idea is simply that for all $\boldsymbol{a} \in \mathcal{A}_d(c)$,

$$\langle \boldsymbol{w}, (\boldsymbol{a} + \boldsymbol{B}_t)\boldsymbol{w} \rangle \leq \|\boldsymbol{a}\| + \langle \boldsymbol{w}, \boldsymbol{B}_t\boldsymbol{w} \rangle \leq ct_{\mathrm{alg}} + \langle \boldsymbol{w}, \boldsymbol{B}_t\boldsymbol{w} \rangle \leq ct_{\mathrm{alg}} + C\sqrt{t \log \binom{n}{k}}$$

for each $t$. Of course, we have to bound this simultaneously for all $t$, and this is done in the proof of Corollary 3.2; c.f. Appendix H. □

# E CONCRETE EXAMPLES: DENOISERS FOR SPARSE LOW-RANK MATRICES

## E.1 ALGORITHMS

In this section, we provide the detailed pseudocode for Algorithms 1 and 2. In Algorithm 1 we use the following soft-thresholding function, with a parameter $s$:

$$\eta_s(y) = \operatorname{sign}(y) \max(|y| - t, 0) = \operatorname{sign}(y)(|y| - t)_+$$

---

**Algorithm 1** Submatrix Estimation Algorithm (very sparse regime)

---

1: **Input:** Data $\boldsymbol{y}_t$; time $t$; parameters $s, \varepsilon$
2: **Output:** Estimate of $\boldsymbol{x}$: $\hat{\boldsymbol{m}}(\boldsymbol{y}_t, t)$
3: Let $\boldsymbol{g}_t \sim \mathsf{N}(0, t\boldsymbol{I}_{n \times n})$ and compute $\boldsymbol{y}_{t,+} := \boldsymbol{y}_t + \sqrt{\varepsilon}\boldsymbol{g}_t$, $\boldsymbol{y}_{t,-} := \boldsymbol{y}_t - \boldsymbol{g}_t/\sqrt{\varepsilon}$
4: Symmetrize: $\boldsymbol{A}_{t,+} = (\boldsymbol{y}_{t,+} + \boldsymbol{y}_{t,+}^\mathsf{T})/(2\sqrt{t})$, $\boldsymbol{A}_{t,-} = (\boldsymbol{y}_{t,-} + \boldsymbol{y}_{t,-}^\mathsf{T})/(2\sqrt{t})$
5: Compute top eigenvector of $\eta_s(\boldsymbol{A}_{t,+})$, denoted if by $\boldsymbol{v}_t$
6: If $t \geq t_{\mathrm{alg}} \vee 1$ and $\lambda_1(\eta_s(\boldsymbol{A}_{t,+})) > k + \dfrac{\sqrt{t}}{s}$, continue; otherwise return $\hat{\boldsymbol{m}}(\boldsymbol{y}, t) := \boldsymbol{0}$
7: Compute the vector $\hat{\boldsymbol{v}}_t := \boldsymbol{A}_{t,-}\boldsymbol{v}_t$
8: Let $\hat{S}$ be the set of $k$ indices $i$ of largest values of $|\hat{v}_{t,i}|$, and compute vector $\boldsymbol{w}$ such that $w_i = \operatorname{sign}(\hat{v}_{t,i})\mathbf{1}_{i \in \hat{S}}$
9: **return** $\hat{\boldsymbol{m}}(\boldsymbol{y}_t, t) := \mathbf{1}_{\hat{S}}\mathbf{1}_{\hat{S}}^\mathsf{T}/k$

---

**Algorithm 2** Submatrix Estimation Algorithm (moderately sparse regime)

---

1: **Input:** Data $\boldsymbol{y}_t$; time $t$; parameter $\varepsilon$
2: **Output:** Estimate of $\boldsymbol{x}$: $\hat{\boldsymbol{m}}(\boldsymbol{y}_t, t)$
3: If $t \geq t_{\mathrm{alg}}$, continue; otherwise return $\hat{\boldsymbol{m}}(\boldsymbol{y}_t, t) = \boldsymbol{0}$
4: Symmetrize: $\boldsymbol{A}_t = (\boldsymbol{y}_t + \boldsymbol{y}_t^\mathsf{T})/(2\sqrt{t})$
5: If $t \geq n^2$ and $\lambda_1(\boldsymbol{A}_t) \leq \sqrt{t}/2$, return $\boldsymbol{0}$; otherwise continue
6: Compute top eigenvector of $\boldsymbol{A}_t$, denoted if by $\boldsymbol{v}_t$
7: Compute $\hat{S}$ by $\hat{S} := \left\{ i \in [n] : |v_{t,i}| \geq \frac{\varepsilon}{\sqrt{k}} \right\}$
8: Compute vector $\boldsymbol{w}$ such that $w_i = \operatorname{sign}(v_{t,i})\mathbf{1}_{i \in \hat{S}}$
9: **return** $\hat{\boldsymbol{m}}(\boldsymbol{y}_t, t) := \boldsymbol{w}\boldsymbol{w}^\mathsf{T}/|\hat{S}|$ if $|\hat{S}| \geq k/2$; otherwise return $\boldsymbol{0}$

---

## E.2 MODERATELY SPARSE REGIME: $\sqrt{n} \ll k \ll n$

Since Theorem 3.2 is somewhat abstract, we complement it with an explicit example of $\hat{\boldsymbol{m}}$: namely, it is a modification of a standard spectral estimator. While achieving near optimal estimation error (among polytime algorithms), $\hat{\boldsymbol{m}}$ fails to generate samples from the correct distribution.

**Proposition E.1.** *Assume $\sqrt{n} \ll k \ll n$, so that $t_{\mathrm{alg}}(n, k) := n/2$ per (9). Then the estimator $\hat{\boldsymbol{m}}$ defined in Algorithm 2 satisfies the following:*

M1. $\hat{\boldsymbol{m}}(\cdot)$ *can be evaluated in polynomial time.*

M2. *For any $\delta > 0$, there exists $c = c(\delta)$, $C = C(\delta)$ such that*

$$\inf_{t \leq (1-\delta)t_{\mathrm{alg}}} \mathbb{E}\big\{\|\hat{\boldsymbol{m}}(\boldsymbol{y}_t, t) - \boldsymbol{x}\|^2\big\} = 1 - o_n(1), \qquad \sup_{t \geq (1+\delta)t_{\mathrm{alg}}} \mathbb{E}\big\{\|\hat{\boldsymbol{m}}(\boldsymbol{y}_t, t) - \boldsymbol{x}\|^2\big\} \leq C\,e^{-cn/k}.$$

M3. *For any $\Delta > 0$, we have incorrect sampling:* $\inf_{t \in \mathbb{N} \cdot \Delta} W_1(\hat{\boldsymbol{m}}(\hat{\boldsymbol{y}}_t, t), \boldsymbol{x}) = 1 - o_n(1)$.

Therefore, we enforce that $\hat{\boldsymbol{m}} \equiv \boldsymbol{0}$ for $t < t_{\mathrm{alg}}$. Recall that this implies Assumption 1 holds trivially. Regarding the specific design of $\hat{\boldsymbol{m}}$, Algorithm 2 uses a thresholded spectral approach. We compute the leading eigenvector of (the symmetrized version of) $\boldsymbol{y}_t$, call it $\boldsymbol{v}_t \in \mathbb{R}^n$. We then estimate the support $S$ of the latent rank-one matrix $\boldsymbol{x}$ using the entries of $\boldsymbol{v}_t$ with largest magnitude.

### E.3 VERY SPARSE REGIME: $k \ll \sqrt{n}$

We have an analogous result for the very sparse regime, where the sparsity level $k \ll \sqrt{n}$.

**Proposition E.2.** *Assume $(\log n)^{5/2} \lesssim k \ll \sqrt{n}$, and note that here $t_{\mathrm{alg}}(n, k) = k^2 \log(n/k^2)$, per (9). Then the randomized estimator $\hat{\boldsymbol{m}} : \mathbb{R}^{n \times n} \times \mathbb{R} \to \mathbb{R}^{n \times n}$ of Algorithm 1 satisfies the following:*

**M1.** *$\hat{\boldsymbol{m}}(\cdot)$ can be evaluated in polynomial time.*

**M2.** *For any $\delta > 0$ and $D > 0$:*
$$\inf_{t \leq (1-\delta)t_{\mathrm{alg}}} \mathbb{E}\big\{\|\hat{\boldsymbol{m}}(\boldsymbol{y}_t, t) - \boldsymbol{x}\|^2\big\} = 1 - o_n(1)\,, \quad \sup_{t \geq (1+\delta)t_{\mathrm{alg}}} \mathbb{E}\big\{\|\hat{\boldsymbol{m}}(\boldsymbol{y}_t, t) - \boldsymbol{x}\|^2\big\} \ll n^{-D}\,.$$

**M3.** *For any $\Delta > 0$, we have incorrect sampling: $\inf_{t \in \mathbb{N} \cdot \Delta} W_1(\hat{\boldsymbol{m}}(\hat{\boldsymbol{y}}_t, t), \boldsymbol{x}) = 1 - o_n(1)$.*

The pseudocode for the estimator $\hat{\boldsymbol{m}}(\cdot)$ that is constructed in the above is given as Algorithm 1. This is based on a standard approach in the literature Deshpande & Montanari (2016); Cai et al. (2017), with some modifications to allow for its analysis in the diffusion setting. The main steps are as follows: (1) Perform Gaussian data splitting of $\boldsymbol{y}_t$ into $\boldsymbol{y}_{t,+}$, $\boldsymbol{y}_{t,-}$, see Line 3 of Algorithm 1, with most of the information preserved in $\boldsymbol{y}_{t,+}$. (2) Use entrywise soft thresholding $\eta_s(x) = (|x| - s)_+ \operatorname{sign}(x)$ to reduce the noise in the symmetrized version of $\boldsymbol{y}_{t,+}$. (3) Compute a first estimate of the latent vector $\boldsymbol{1}_S$ by the principal eigenvector of the above matrix. (4) Refine this estimate using the remaining information $\boldsymbol{y}_{t,-}$.

We point out that Proposition 3.3 remains true in the regime $\sqrt{n} \ll k \ll n$, and hence we observe a gap between $t_{\mathrm{alg}}(n, k)$ and $t_{\mathrm{Bayes}}(n, k)$ in this regime as well.

## F PROOF OF PROPOSITION E.1

### F.1 PROPERTIES OF THE ESTIMATOR $\hat{\boldsymbol{m}}(\cdot)$

**Proposition F.1.** *Assume $\sqrt{n} \ll k \ll n$, and note that in this case $t_{\mathrm{alg}}(n, k) = n/2$. Let $\hat{\boldsymbol{m}}(\cdot)$ be the estimator of Algorithm 2 with input parameter $\varepsilon$. For every $\delta > 0$, there exists $\varepsilon > 0$ such that*
$$\sup_{t \geq (1+\delta)t_{\mathrm{alg}}} \mathbb{E}\left[\|\hat{\boldsymbol{m}}(\boldsymbol{y}_t, t) - \boldsymbol{x}\|^2\right] \leq C \, e^{-n\varepsilon^2/64k}\,. \tag{22}$$

The proof of this proposition is standard, and will be presented in Appendix O. We note that the rate in Equation (22) gets slower the closer $k$ is to $n$; it is super-polynomial if $n \gg k \log n$.

By definition, when $\sqrt{n} \ll k \ll n$ and $t < t_{\mathrm{alg}}(n, k)$, the Algorithm 2 returns $\hat{\boldsymbol{m}}(\boldsymbol{y}, t) = \boldsymbol{0}$, so we automatically have the following result.

**Proposition F.2.** *For any fixed $\delta > 0$, and $t \leq (1 - \delta)t_{\mathrm{alg}}$, we have $\|\hat{\boldsymbol{m}}(\boldsymbol{y}_t, t) - \boldsymbol{x}\| = 1$.*

### F.2 AUXILIARY LEMMAS

The following lemmas are needed for the analysis of the generated diffusion. Their proofs are deferred to Appendices P, Q, R, S.

**Lemma F.3.** *Let $\boldsymbol{W}_t$ be a GOE process. Then for each time $t_0 \geq 0$,*
$$\mathbb{P}\left(\max_{0 \leq t \leq t_0} \|\boldsymbol{W}_t\|_{\mathrm{op}} \geq 16\sqrt{t_0 n}\right) \leq 2\exp\left(-32n\right).$$

**Lemma F.4.** *Let $\boldsymbol{W}_t$ be a GOE process, and let $\boldsymbol{v}_t$ be any eigenvector of $\boldsymbol{W}_t$ for every $t \geq 0$. Define the set*
$$A(\boldsymbol{v}_t; C) = \left\{i : 1 \leq i \leq n, |v_{ti}| \geq \frac{C\sqrt{\log(n/k)}}{\sqrt{n}}\right\}$$
*Then for any $C > 4$, we have*
$$\mathbb{P}\left(|A(\boldsymbol{v}_t; C)| \geq \max\{\sqrt{k}, k^2/n\}\right) = O\left(\exp(-(1/3)n^{1/4})\right)$$
*As a consequence, using this eigenvector, $\hat{\boldsymbol{m}}$ will evaluate to $\boldsymbol{0}$ per line 8 of Algorithm 2.*

**Lemma F.5.** *Let $W_t$ be a* GOE *process, and for each $t$, let $v_t$ be a top eigenvector of $W_t$. Then for any times $t_0 \leq t_1$, with probability at least $1 - 2\exp(-32n)$,*

$$\sup_{t_0 \leq t \leq t_1} |\langle v_t, W_{t_0} v_t \rangle - \lambda_1(W_{t_0})| \leq 32\sqrt{n(t_1 - t_0)}.$$

**Lemma F.6** (Concentration for deformed GOE model). *Consider the model $Y = \theta v v^\mathsf{T} + W$ for $W \sim$ GOE$(n)/\sqrt{n}$ and $\theta > 1$ a constant, $v$ a unit vector. Let $v_1(Y)$ be the top eigenvector of $Y$. Define $(x^\star, u^\star) = (\theta + 1/\theta, 1 - 1/\theta^2)$. For any closed set $F$ such that $d((x^\star, u^\star), F) > 0$, there exists a constant $c > 0$ such that*

$$\mathbb{P}\left((\lambda_1(Y), \langle v_1(Y), v \rangle^2) \in F\right) \leq \exp(-cn)$$

*for all $n$ large enough.*

We only use Lemma F.6 for the alignment $\langle v_1(Y), v \rangle^2$.

## F.3 ANALYSIS OF THE DIFFUSION PROCESS: PROOF OF PROPOSITION E.1

We will prove Theorem E.1 for $1 \gg \varepsilon \geq C\sqrt{\log(n/k)/(n/k)}$ for some sufficiently large constant $C$.

Suppose that we generate the following diffusion, with $(z_t)_{t \geq 0}$ a standard $n^2$-dimensional BM, and $\hat{y}_0 = 0$:

$$\hat{y}_{\ell\Delta} = \hat{y}_{(\ell-1)\Delta} + \Delta \cdot \hat{m}\left(\hat{y}_{(\ell-1)\Delta}, (\ell-1)\Delta\right) + \left(z_{\ell\Delta} - z_{(\ell-1)\Delta}\right).$$

We will prove that the generated diffusion never passes the termination conditions (c.f. Algorithm 2, lines 3, 5, 8).

### F.3.1 ANALYSIS UP TO AN INTERMEDIATE TIME

Define $t_{\text{between}} = n^2$. Following the same strategy with Section I.2, we will first show that $\hat{m} = 0$ up to $t_{\text{between}}$ with high probability by analyzing only the noise process (in short, if $\hat{m} = 0$ always, our generated diffusion coincides with the noise process). Our strategy is of the same nature as that of Section I.2. Indeed, we will attempt to prove that $\hat{m} = 0$ simultaneously for all $t$, with high probability. In this phase ($0 \leq t \leq t_{\text{between}}$), we will show that $|\hat{S}| < k/2$ (c.f. definition in Algorithm 2, lines 7, 8) for $0 \leq t \leq t_{\text{between}}$, with high probability ($v_t$ is the top eigenvector of $A_t$, c.f. Algorithm 2). Note that line 5 of Algorithm 2 is not relevant in this phase. We first show this for a sequence of time points $\{t_\ell\}_{\ell \geq 1}$, then control the in-between fluctuations. We can set $t_1$ to be any value in $[0, n/2)$, as the algorithm returns 0 if $t < t_{\text{alg}} = n/2$ anyway. We denote the GOE process

$$W_t = \frac{B_t + B_t^\mathsf{T}}{2} = \sqrt{t} A_t.$$

It is clear that the eigenvectors of $W_t$ and $A_t$ coincide.

We choose the following time points:

$$t_\ell = \frac{n}{2} - 1 + \frac{\ell}{n^4}.$$

To exceed $t_{\text{between}} = n^2$, we will need $n^6$ values of $\ell$. By union bound from Lemma F.4 (recall also the definition of the set $A(v; C)$ from this Lemma),

$$\mathbb{P}\left(\exists 1 \leq \ell \leq n^6 : |A(v_{t_\ell}; C)| \geq \max\{\sqrt{k}, k^2/n\}\right) \leq O\left(\exp(-(1/3)n^{1/4} + 6\log n)\right) \quad (23)$$

Next, we will control the in-between fluctuations; specifically, we would like to show that $\max_{t_\ell \leq t \leq t_{\ell+1}} |A(v_t; C)| \leq C_0 \max\{\sqrt{k}, k^2/n\}$ simultaneously for many values of $\ell$ (with high probability), for some constant $C_0 > 0$. Our approach is as follows.

(i) Let $v_t$ be a top eigenvector of $W_t$. If $t$ is close to $t_\ell$, then $v_t$ is an approximate solution to the equation (in $v$):

$$v^\mathsf{T} W_{t_\ell} v = \lambda_1(W_{t_\ell})$$

(ii) $v_t$ can be written in the coordinate system of the orthonormal eigenvectors $U_{t_\ell} = [u_1|\cdots|u_n]$ of $W_{t_\ell}$, corresponding to decreasing eigenvalues $\lambda_1(W_{t_\ell}) \geq \cdots \geq \lambda_n(W_{t_\ell})$. Namely, $v_t = U_{t_\ell} U_{t_\ell}^\mathsf{T} v_t = U_{t_\ell} w$ with $\|w\| = 1$.

(iii) Let $m$ be a (sufficiently large) constant integer with $1 \leq m \leq n$. The first $m$ components of $w$ take up $1 - o(1)$ in $L_2$-norm by (i) and (ii) with overwhelming probability, from which we can simply use triangle inequality to upper bound $|A(v_t; C)|$ according to Lemma F.4 for $u_1, \cdots, u_m$, which incurs only a constant factor of error probability, by union bound.

Define $p_n = P\left(|\lambda_1(W_1) - \lambda_7(W_1)| \leq n^{-C'-1/2}\right)$ for any $C' > 0$ (here we take $m = 7$). We use the following result (we have accounted for the scaling).

**Lemma F.7** (Corollary 2.5, Nguyen et al. (2017)). *Let $W_1 \sim (1/\sqrt{2})\mathsf{GOE}(n)$. For any fixed $l \geq 1, C' > 0$, there exists a constant $c_0 = c_0(l, C')$ such that*

$$\mathbb{P}\left(\lambda_1(W_1) - \lambda_{1+l}(W_1) \leq \frac{1}{2}n^{-C'-1/2}\right) \leq c_0 n^{-C' \cdot \frac{l^2+2l}{3}}.$$

We materialize our approach above. We can write, with $v_t = U_{t_\ell} w$:

$$v_t^\mathsf{T} W_{t_\ell} v_t = w^\mathsf{T} D_{t_\ell} w = \sum_{i=1}^n (D_{t_\ell})_{ii} w_i^2.$$

We then obtain that p

$$v_t^\mathsf{T} W_{t_\ell} v_t - \lambda_1(W_{t_\ell}) \leq \sum_{i=8}^n (\lambda_i(W_{t_\ell}) - \lambda_1(W_{t_\ell})) w_i^2 < -\frac{1}{2}\sqrt{t_\ell} n^{-3/2} \sum_{i=8}^n w_i^2$$

with probability at least $1 - p_n \geq 1 - c_0 n^{-8}$, from Lemma F.7 and $W_{t_\ell} \sim \sqrt{t_\ell} W_1$. Now from Lemma F.5, we know that with probability at least $1 - 2\exp(-32n)$,

$$v_t^\mathsf{T} W_{t_\ell} v_t - \lambda_1(W_{t_\ell}) \geq -32\sqrt{n(t_{\ell+1} - t_\ell)}.$$

With probability at least $1 - c_0 n^{-8} - 2\exp(-32n)$, both of these statements are true, uniformly over $t_\ell \leq t \leq t_{\ell+1}$, leading to

$$64\sqrt{\frac{t_{\ell+1} - t_\ell}{t_\ell}} \geq n^{-3/2} \sum_{i=8}^n w_i^2.$$

A simple bit of algebra shows that

$$\sqrt{\frac{t_{\ell+1} - t_\ell}{t_\ell}} \leq 2n^{-5/2} \Rightarrow \sum_{i=8}^n w_i^2 \leq 128 n^{-1}.$$

Consider the first 7 eigenvectors $\{u_{t_\ell,i}\}_{i=1}^7$ of $W_{t_\ell}$. Let

$$A = \bigcup_{i=1}^7 A(u_{t_\ell,i}; C).$$

From Lemma F.4 and a union bound, that $|A| \leq 7\max\{\sqrt{k}, k^2/n\}$ with probability at least $1 - O(\exp(-(1/3)n^{1/4}))$. For every $j \in A^c$, we have

$$|v_{tj}| \leq \sum_{i=1}^n |w_i| \cdot |u_{t_\ell,i,j}| < \sum_{i=1}^7 |w_i| \cdot \frac{C\sqrt{\log(n/k)}}{\sqrt{n}} + \sum_{i=8}^n |w_i| \leq \frac{C'\sqrt{\log(n/k)}}{\sqrt{n}} + \frac{128}{\sqrt{n}} < C''\sqrt{\frac{\log(n/k)}{n}}$$

for a large enough constant $C'' > 0$. This means that with probability at least $1 - O(n^{-8})$,

$$\sup_{t_\ell \leq t \leq t_{\ell+1}} |A(v_t; C'')| \leq 7\max\{\sqrt{k}, k^2/n\} < k/2$$

From Equation (23) and a union bound over $\ell \geq 1$, we know that with high probability,

$$\sup_{n/2-1 \leq t \leq n^2} |A(v_t; C)| \leq 7\max\{\sqrt{k}, k^2/n\}$$

for some absolute constant $C > 0$, meaning that $\hat{m} = 0$ up to $t_{\text{between}}$, as long as

$$\varepsilon > \frac{C\sqrt{\log(n/k)}}{\sqrt{n/k}}$$

### F.3.2 ANALYSIS TO THE INFINITE HORIZON

We will prove that simultaneously for all $t \geq n^2$, Algorithm 2 always terminates at line 5, or that $\lambda_1(\boldsymbol{W}_t) \leq t/2$. Similar to Subsection F.3.1, we choose the following sequence of time points for all $\ell \geq 1$:

$$t_\ell^{(2)} = n^2 + \ell - 1$$

By standard Gaussian concentration and the Bai-Yin theorem, we get, for instance, the following tail bound (constants are loose) for all $x \geq 0$:

$$\mathbb{P}\left(\lambda_1\left(\frac{\boldsymbol{W}_t}{\sqrt{t}}\right) \geq 4\sqrt{n} + x\right) \leq 2\exp\left(-\frac{x^2}{2}\right)$$

Set $x = \frac{\sqrt{t}}{8}$. Since $t \geq n^2$, we have $x \geq n/8$, and so $x + 4\sqrt{n} \leq 2x$ for $n$ large enough. Consequently,

$$\mathbb{P}\left(\lambda_1\left(\frac{\boldsymbol{W}_t}{\sqrt{t}}\right) \geq \frac{\sqrt{t}}{4}\right) \leq 2\exp\left(-c_1 t\right)$$

for some universal constant $c_1 > 0$. A union bound for the chosen points gives:

$$\mathbb{P}\left(\exists \ell \geq 1 : \lambda_1\left(\boldsymbol{W}_{t_\ell^{(2)}}\right) \geq t_\ell^{(2)}/4\right) \leq 2\sum_{\ell=1}^{\infty} \exp\left(-c_1 t_\ell^{(2)}\right) = 2\exp(-c_1 n^2)\sum_{\ell=1}^{\infty} \exp(-c_1(\ell-1)) \lesssim \exp(-c_1 n^2) \tag{24}$$

Next we control the in-between fluctuations. From a simple modification of Lemma F.3, we have

$$\mathbb{P}\left(\sup_{t_\ell^{(2)} \leq t \leq t_{\ell+1}^{(2)}} \left|\lambda_1\left(\boldsymbol{W}_{t_\ell^{(2)}}\right) - \lambda_1\left(\boldsymbol{W}_t\right)\right| \geq 16\sqrt{(t_{\ell+1}^{(2)} - t_\ell^{(2)}) \cdot t_\ell^{(2)} n}\right) \leq 2\exp\left(-32n t_\ell^{(2)}\right)$$

so that by union bound

$$\mathbb{P}\left(\exists \ell \geq 1 : \sup_{t_\ell^{(2)} \leq t \leq t_{\ell+1}^{(2)}} \left|\lambda_1\left(\boldsymbol{W}_{t_\ell^{(2)}}\right) - \lambda_1\left(\boldsymbol{W}_t\right)\right| \geq 16\sqrt{(t_{\ell+1}^{(2)} - t_\ell^{(2)}) \cdot t_\ell^{(2)} n}\right) \lesssim \exp(-32n^3) \tag{25}$$

Consider the intersection of events described in Equations (24) and (25):

$$A = \left\{\exists \ell \geq 1 : \lambda_1\left(\boldsymbol{W}_{t_\ell^{(2)}}\right) \geq t_\ell^{(2)}/4\right\} \cup \left\{\exists \ell \geq 1 : \sup_{t_\ell^{(2)} \leq t \leq t_{\ell+1}^{(2)}} \left|\lambda_1\left(\boldsymbol{W}_{t_\ell^{(2)}}\right) - \lambda_1\left(\boldsymbol{W}_t\right)\right| \geq 16\sqrt{(t_{\ell+1}^{(2)} - t_\ell^{(2)}) \cdot t_\ell^{(2)} n}\right\}$$

For each $t \geq n^2$, let $t_\ell$ be largest such that $t_\ell \leq t < t_{\ell+1}$. On $A$, we have

$$\lambda_1(\boldsymbol{W}_t) \leq \frac{t_\ell^{(2)}}{4} + 16\sqrt{(t_{\ell+1}^{(2)} - t_\ell^{(2)}) \cdot t_\ell^{(2)} n} = \frac{t_\ell^{(2)}}{4} + 16\sqrt{t_\ell^{(2)} n} \leq \frac{t_\ell^{(2)}}{2} \leq \frac{t}{2}$$

for $n$ large enough, since $t_\ell^{(2)} \geq n^2 \gg n$. Hence the algorithm always returns $\boldsymbol{0}$ with high probability, and we are done.

## G   PROOF OF THEOREM 1

We define $\hat{m}$ as follows:

$$\hat{\boldsymbol{m}}(\boldsymbol{y}, t) = \begin{cases} \hat{\boldsymbol{m}}_0(\boldsymbol{y}, t)\mathbf{1}_{\|\hat{\boldsymbol{m}}_0(\boldsymbol{y}, t)\| \leq \varepsilon} & \text{if } t \leq (1-\gamma)t_{\text{alg}}, \\ \hat{\boldsymbol{m}}_0(\boldsymbol{y}, t) & \text{if } (1-\gamma)t_{\text{alg}} < t < (1+\delta)t_{\text{alg}}, \\ \hat{\boldsymbol{m}}_0(\boldsymbol{y}, t)\phi(\boldsymbol{y}, t) & \text{if } t \geq (1+\delta)t_{\text{alg}}, \end{cases}$$

First, we check that Condition M2 holds.

$$\int_0^\infty \mathbb{E}[\|\hat{\boldsymbol{m}}(\boldsymbol{y}_t, t) - \hat{\boldsymbol{m}}_0(\boldsymbol{y}_t, t)\|^2]dt$$

$$= \int_0^{(1-\gamma)t_{\text{alg}}} \mathbb{E}[\|\hat{\boldsymbol{m}}(\boldsymbol{y}_t, t) - \hat{\boldsymbol{m}}_0(\boldsymbol{y}_t, t)\|^2]dt + \int_{(1+\delta)t_{\text{alg}}}^\infty \mathbb{E}[\|\hat{\boldsymbol{m}}(\boldsymbol{y}_t, t) - \hat{\boldsymbol{m}}_0(\boldsymbol{y}_t, t)\|^2]dt$$

$$\overset{(a)}{\leq} \int_0^{(1-\gamma)t_{\text{alg}}} \mathbb{P}(\|\hat{\boldsymbol{m}}_0(\boldsymbol{y}_t, t)\| > \varepsilon)dt + \int_{(1+\delta)t_{\text{alg}}}^\infty \mathbb{P}(\phi(\boldsymbol{y}, t) = 0)\, dt$$

$$= O(\eta_1(d) + \eta_2(d))$$

where in $(a)$ we use the fact that $\phi$ is binary and Conditions $i)$, $ii)$ and $iii.1)$. Secondly, we check that Condition M3 holds. To reduce notational clutter, we assume that $t_1/\Delta = \ell_0$ is an integer. Then, we can write

$$\hat{\boldsymbol{y}}_{t_1} = \boldsymbol{B}_{t_1} + \Delta \sum_{i=1}^{\ell_0} \hat{\boldsymbol{m}}(\hat{\boldsymbol{y}}_{i\Delta}, i\Delta) \tag{26}$$

where the drift accumulation term is bounded by, from Condition i):

$$\left\| \Delta \sum_{i=1}^{\ell_0} \hat{\boldsymbol{m}}(\hat{\boldsymbol{y}}_{i\Delta}, i\Delta) \right\| \leq \varepsilon(1-\gamma)t_{\text{alg}} + (\gamma + \delta)t_{\text{alg}} \leq c \cdot t_{\text{alg}}$$

for every $c \in (0, 1)$ by taking $\varepsilon, \gamma$ small enough, as $\delta = o_d(1)$. Suppose that from Condition iii.2), the event $\{\phi(\boldsymbol{a} + \boldsymbol{B}_t) = 0 \text{ for all } t \geq t_1, \boldsymbol{a} \in \mathcal{A}_d(c)\}$ holds. Then it is clear that $\hat{\boldsymbol{m}}(\hat{\boldsymbol{y}}_{t_1}, t_1) = \phi(\hat{\boldsymbol{y}}_{t_1}, t_1) = 0$ by definition of $\hat{\boldsymbol{m}}$. Suppose the inductive hypothesis that $\hat{\boldsymbol{m}}(\hat{\boldsymbol{y}}_{t_1+i\Delta}, t_1 + i\Delta) = 0$ for all $0 \leq i \leq k$. Then from Eq. (26), we get that $\hat{\boldsymbol{y}}_{t_1+(k+1)\Delta} - \boldsymbol{B}_{t_1+(k+1)\Delta} \in \mathcal{A}_d(c)$ and so $\hat{\boldsymbol{m}}(\hat{\boldsymbol{y}}_{t_1+(k+1)\Delta}, t_1 + (k+1)\Delta) = 0$. Consequently, $\hat{\boldsymbol{m}}(\hat{\boldsymbol{y}}_t, t) = 0$ for all $t = \ell\Delta, t \geq t_1$. By Condition $iii.2)$, the preceding event holds with high probability, so that for each $t = \ell\Delta, t \geq t_1$, $\hat{\boldsymbol{m}}(\hat{\boldsymbol{y}}_t, t) = 0$ with high probability. By boundedness of $\hat{\boldsymbol{m}}$ and definition of the Wasserstein-1 distance used on the function $f(\cdot) = \|\cdot\|$, we obtain that $W_1(\hat{\boldsymbol{m}}(\hat{\boldsymbol{y}}_t, t), \boldsymbol{x}) \geq \alpha - o(1)$. The proof ends here.

## H   PROOF OF COROLLARY 3.2

### H.1   AUXILIARY LEMMAS

We will use the following lemmas, whose proofs are deferred to Appendix L.

**Lemma H.1.** *Let $\boldsymbol{W} \sim \text{GOE}(n, 1/2)$, and $C > \sqrt{2}$ some positive constant. Then we have*

$$\mathbb{P}\left( \max_{\boldsymbol{v} \in \Omega_{n,k}} |\langle \boldsymbol{v}, \boldsymbol{W}\boldsymbol{v}\rangle| \geq C\sqrt{\log\binom{n}{k}} \right) \leq 2\binom{n}{k}^{-C^2/2+2}.$$

We state the following non-asymptotic result from Peng (2012).

**Lemma H.2** (Theorem 3.1, Peng (2012).). *Let $\boldsymbol{y} = \theta\boldsymbol{u}\boldsymbol{u}^\mathsf{T} + \text{GOE}(n, 1/n)$, and denote by $\lambda_1(\boldsymbol{y})$ the top eigenvalue of $\boldsymbol{y}$. Letting $\xi(\theta) := \theta + \theta^{-1}$, the following holds for every $x \geq 0$ and $\theta > 1$:*

$$\mathbb{P}\left( \lambda_1(\boldsymbol{y}) \leq \xi(\theta) - x - \frac{2}{n} \right) \leq \exp\left( -\frac{(n-1)(\theta-1)^4}{16\theta^2} \right) + 8\exp\left( -\frac{1}{4} \cdot \frac{(n-1)(\theta-1)^5 x^2}{(\theta+1)^3} \right).$$

In Appendix L, we will use the last lemma to prove the bound below.

**Lemma H.3** (Alignment bound). *Let $\boldsymbol{y} = \theta\boldsymbol{u}\boldsymbol{u}^\mathsf{T} + \text{GOE}(n, 1/n)$, where $\theta = \sqrt{1+\delta}$. Let $\boldsymbol{v}_1$ be the top eigenvector of $\boldsymbol{y}$. Then, there exist constants $C, c > 0$ such that the following holds for any $\delta = \delta_n$ with $n^{-c} \ll \delta \ll 1$ for some $c > 0$ small enough:*

$$\mathbb{P}\left( |\langle \boldsymbol{v}_1, \boldsymbol{u}\rangle| \leq c\delta^2 \right) \leq C\, e^{-cn^{1/3}}. \tag{27}$$

We also use the following lemma, which is implied in the proof of Lemma F.4.

**Lemma H.4.** *Let $\boldsymbol{g} \sim \text{N}(\boldsymbol{0}, \boldsymbol{I}_n)$ and define the set*

$$\mathcal{L}(\boldsymbol{g}; C) = \left\{ i : 1 \leq i \leq n, |g_i| \geq C\sqrt{\log(n/k)} \right\}.$$

*Assume $\sqrt{n} \ll k \ll n$. Then, for any $C$ large enough, there exists $C_*$ such that*

$$\mathbb{P}\left( |\mathcal{L}(\boldsymbol{g}; C)| \geq \left( \sqrt{k} \vee \frac{k^2}{n} \right) \right) \leq C_* e^{-n^{1/4}}.$$

## H.2 PROOF OF COROLLARY 3.2

Let $\delta = o_n(1)$ be a parameter to be chosen later and recall that $t_{\mathrm{alg}} = n/2$. We define $\hat{m}$ as follows:

$$
\hat{\boldsymbol{m}}(\boldsymbol{y},t) = \begin{cases}
\hat{\boldsymbol{m}}_0(\boldsymbol{y},t)\mathbf{1}_{\|\hat{\boldsymbol{m}}_0(\boldsymbol{y},t)\| \leq \varepsilon} & \text{if } t \leq (1-\gamma)t_{\mathrm{alg}}, \\
\hat{\boldsymbol{m}}_0(\boldsymbol{y},t) & \text{if } (1-\gamma)t_{\mathrm{alg}} < t < (1+\delta)t_{\mathrm{alg}}, \\
\hat{\boldsymbol{m}}_0(\boldsymbol{y},t)\phi_1(\boldsymbol{y},t) & \text{if } (1+\delta)t_{\mathrm{alg}} \leq t < n^4, \\
\hat{\boldsymbol{m}}_0(\boldsymbol{y},t)\phi_2(\boldsymbol{y},t) & \text{if } t \geq n^4,
\end{cases}
$$

where $\phi_1, \phi_2 : \mathbb{R}^{n \times n} \times \mathbb{R}_{\geq 0} \to \{0,1\}$ are defined below and $\gamma$ is given in Assumption 1. It will be clear from the constructions below that $\phi_1, \phi_2$ can be evaluated in polynomial time. To establish the relationship between Corollary 3.2 and Theorem 1, we make a few remarks:

- Assumption 1 is used in (29);
- By defining the hypothesis test $\phi$ such that

$$
\phi(\boldsymbol{y},t) = \begin{cases}
\phi_1(\boldsymbol{y},t) & \text{if } (1+\delta)t_{\mathrm{alg}} \leq t < n^4 \\
\phi_2(\boldsymbol{y},t) & \text{if } t \geq n^4
\end{cases}
$$

  we recover the desired properties of Assumption 2, with $\eta_2(n) = O(n^{-D})$ for any $D > 0$.

In order to prove claim M2, we need to bound $J_n(0, \infty)$, whereby, for $t_a \leq t_b$,

$$
J_n(t_a, t_b) := \int_{t_a}^{t_b} \mathbb{E}\left[\|\hat{\boldsymbol{m}}(\boldsymbol{y}_t,t) - \hat{\boldsymbol{m}}_0(\boldsymbol{y}_t,t)\|^2\right] \mathrm{d}t
$$

Setting $t_1 = (1+\delta)t_{\mathrm{alg}}$ and $t_2 = n^4$, we write

$$
J_n(0,\infty) = J_n(0,t_1) + J_n(t_1,t_2) + J_n(t_2,\infty), \tag{28}
$$

and will bound each of the three terms separately.

**Bounding $J_n(0,t_1)$.** By Assumption 1, we know that

$$
J_n(0,t_1) \leq \int_0^{(1-\gamma)t_0} \mathbb{P}(\|\hat{\boldsymbol{m}}_0(\boldsymbol{y}_t,t)\| > \varepsilon)\, \mathrm{d}t = O(n^{-D}), \tag{29}
$$

for every $D > 0$.

**Bounding $J_n(t_1,t_2)$.** For a matrix $\boldsymbol{y} \in \mathbb{R}^{n \times n}$ and time point $t$, we define $\phi_1(\boldsymbol{y},t)$ according to the following procedure:

1. Compute the symmetrized matrix $\boldsymbol{A} = (1/2)(\boldsymbol{y} + \boldsymbol{y}^\mathsf{T})$;
2. Compute its top eigenvector $\boldsymbol{v}$ (choose at random if this is not unique).
3. Let $S \subseteq [n]$ be the $k$ positions in $\boldsymbol{v}$ with the largest magnitude, and define $\hat{\boldsymbol{v}} \in \mathbb{R}^n$ with $\hat{v}_i = (1/\sqrt{k})\operatorname{sign}(v_i)\mathbf{1}_{i \in S}$;
4. Compute the test statistic $s := \langle \hat{\boldsymbol{v}}, \boldsymbol{A}\hat{\boldsymbol{v}} \rangle$, and return 1 if $s \geq \beta t$; 0 otherwise.

Here $\beta \in (0,1)$ is a fixed constant to be chosen later.

We will show that $\phi_1(\boldsymbol{y}_t, t) = 1$ with overwhelming probability for the true model $\boldsymbol{y}_t = t\boldsymbol{x} + \boldsymbol{B}_t$. Define

$$
\boldsymbol{A}_t = \frac{\boldsymbol{y}_t + \boldsymbol{y}_t^\mathsf{T}}{2} = t\boldsymbol{u}\boldsymbol{u}^\mathsf{T} + \boldsymbol{W}_t
$$

where we recall that $\boldsymbol{u} \sim \mathrm{Unif}(\Omega_{n,k})$ and $\boldsymbol{W}_t$ is a $\mathsf{GOE}(n)$ process. Let $\boldsymbol{v}_t, \hat{\boldsymbol{v}}_t$ be the top eigenvector and thresholded vector of $\boldsymbol{A}_t$, respectively.

We have

$$
s_t := \langle \hat{\boldsymbol{v}}_t, \boldsymbol{A}_t\hat{\boldsymbol{v}}_t \rangle = t \cdot \langle \boldsymbol{u}, \hat{\boldsymbol{v}}_t \rangle^2 + \langle \hat{\boldsymbol{v}}_t, \boldsymbol{W}_t\hat{\boldsymbol{v}}_t \rangle. \tag{30}
$$

Using Lemma H.1, we know that $|\langle \hat{\boldsymbol{v}}_t, \boldsymbol{W}\hat{\boldsymbol{v}}_t\rangle| \le 4\sqrt{k\log(n/k)} \cdot \sqrt{t}$ with probability at least $1 - \binom{n}{k}^{-6}$, say. Further

$$\boldsymbol{v}_t = \langle \boldsymbol{v}_t, \boldsymbol{u}\rangle\boldsymbol{u} + \sqrt{1 - \langle \boldsymbol{v}_t, \boldsymbol{u}\rangle^2}\, \boldsymbol{w}\,,$$

where $\boldsymbol{w}$ is a uniformly random unit vector orthogonal to $\boldsymbol{u}$, independent of $\langle \boldsymbol{v}_t, \boldsymbol{u}\rangle$. Alternatively, there exists $\boldsymbol{g} \sim \mathsf{N}(\boldsymbol{0}, \boldsymbol{I}_n)$, , independent of $\langle \boldsymbol{v}_t, \boldsymbol{u}\rangle$ so that:

$$\boldsymbol{w} = \frac{(\boldsymbol{I}_n - \boldsymbol{u}\boldsymbol{u}^\mathsf{T})\boldsymbol{g}}{\|(\boldsymbol{I}_n - \boldsymbol{u}\boldsymbol{u}^\mathsf{T})\boldsymbol{g}\|}\,.$$

For every $1 \le i \le n$, we have

$$|((\boldsymbol{I}_n - \boldsymbol{u}\boldsymbol{u}^\mathsf{T})\boldsymbol{g})_i| \le |g_i| + \frac{|\langle \boldsymbol{u}, \boldsymbol{g}\rangle|}{\sqrt{k}}\,.$$

Since by assumption $k\log(n/k) \ll n$, with probability at least $1 - C_1\exp(-k/2)$, $|\langle \boldsymbol{u}, \boldsymbol{g}\rangle| \le \sqrt{k\log(n/k)}$ and $\|(\boldsymbol{I}_n - \boldsymbol{u}\boldsymbol{u}^\mathsf{T})\boldsymbol{g}\| \ge \sqrt{n}/2$, so that

$$i \in \mathcal{L}(\boldsymbol{g}; C) \quad\Rightarrow\quad |w_i| \le (2C+2)\cdot\sqrt{\frac{1}{n}\log(n/k)}\,.$$

Therefore, by using Lemma H.4 and $k\log(n/k) \gg n^{1/2}$, we get that with probability at least $1 - C_*\exp(-n^{1/4})$, $|\mathcal{A}(\boldsymbol{w}; C)| \le \max\{\sqrt{k}, k^2/n\}$ for some constant $C > 0$, where

$$\mathcal{A}(\boldsymbol{w}; C) := \left\{ i : 1 \le i \le n, |w_i| \ge C\sqrt{\frac{1}{n}\log(n/k)} \right\}\,.$$

By Lemma H.3, we know that with probability at least $1 - C_*\exp(-cn^{1/3})$ for some $C_*, c > 0$, $|\langle \boldsymbol{v}_{t_0(1+\delta)}, \boldsymbol{u}\rangle| \ge c\delta^2$. Since $|\langle \boldsymbol{v}_t, \boldsymbol{u}\rangle|$ is stochastically increasing with $t$ (Fact O.2), we actually have that for any $t \ge (1+\delta)t_{\mathrm{alg}}$, with the same probability $|\langle \boldsymbol{v}_t, \boldsymbol{u}\rangle| \ge c\delta^2$.

On the event $\mathcal{E}_\delta := \{|\langle \boldsymbol{v}_t, \boldsymbol{u}\rangle| \ge c\delta^2\}$, further suppose without loss of generality that $\langle \boldsymbol{v}_t, \boldsymbol{u}\rangle > 0$. As a consequence of the result above, if $i \in \mathrm{supp}(\boldsymbol{u})$ and $i \notin \mathcal{A}(\boldsymbol{w}; C)$, then

$$u_i > 0, i \notin \mathcal{A}(\boldsymbol{w}; C) \quad\Rightarrow\quad v_{t,i} \ge \frac{c\delta^2}{\sqrt{k}} - C\sqrt{\frac{1}{n}\log(n/k)}\,,$$

$$u_i < 0, i \notin \mathcal{A}(\boldsymbol{w}; C) \quad\Rightarrow\quad v_{t,i} \le -\frac{c\delta^2}{\sqrt{k}} + C\sqrt{\frac{1}{n}\log(n/k)}\,.$$

Similarly, if $i \notin \mathrm{supp}(\boldsymbol{u})$, then

$$u_i = 0, \quad i \notin \mathcal{A}(\boldsymbol{w}; C) \quad\Rightarrow\quad |v_{t,i}| \le C\sqrt{\frac{1}{n}\log(n/k)}\,.$$

We next choose $\delta = \delta_n$ such that $\sqrt{(k/n)\log(n/k)} \ll \delta_n \ll 1$. Hence, we conclude that

$$i \in \mathrm{supp}(\boldsymbol{u}) \setminus \mathcal{A}(\boldsymbol{w}; C) \quad\Rightarrow\quad \hat{v}_{t_i} = \mathrm{sign}(\langle \boldsymbol{v}_t, \boldsymbol{u}\rangle)\cdot u_i\,, \tag{31}$$

whence

$$|\langle \boldsymbol{u}, \hat{\boldsymbol{v}}_t\rangle| \ge 1 - \frac{2}{k}|\mathcal{A}(\boldsymbol{w}; C)| = 1 - o_n(1)\,, \tag{32}$$

with probability at least $1 - C_*\exp(-n^{1/4}/3)$. On this event, by Eq. (30) we obtain that

$$s_t \ge t\cdot(1 - o_n(1))^2 - 4\cdot\sqrt{k\log(n/k)}\cdot\sqrt{t}$$

For $t \ge (1+\delta)t_{\mathrm{alg}} = (1+\delta)n/2$, we this implies that, for any fixed $\beta \in (0,1)$, with probability at least $1 - C_*\exp(-n^{1/4}/3)$ (possibly adjusting the constant $C_*$).

Recalling that $\hat{\boldsymbol{m}}(\boldsymbol{y}_t, t) = \hat{\boldsymbol{m}}_0(\boldsymbol{y}_t, t)\phi_1(\boldsymbol{y}_t, t)$ for $t \in [t_1, t_2]$, and $\|\hat{\boldsymbol{m}}_0(\boldsymbol{y}, t)\| \le 1$, we have, for $t_1 = (1+\delta)t_{\mathrm{alg}}$, $t_2 = n^4$ as defined above,

$$J_n(t_1, t_2) = \int_{t_1}^{t_2} \mathbb{E}[\|\hat{\boldsymbol{m}}(\boldsymbol{y}_t, t) - \hat{\boldsymbol{m}}_0(\boldsymbol{y}_t, t)\|^2]\mathrm{d}t \le \int_{t_1}^{t_2} \mathbb{P}\left(\phi_1(\boldsymbol{y}_t, t) = 0\right) \le C_* n^4\, e^{-cn^{1/3}}\,. \tag{33}$$

**Bounding** $J_n(t_2, \infty)$**.** When $t \geq t_2$, we use a simple eigenvalue test. For a matrix $\boldsymbol{y}$, and time point $t$, we define $\phi_2(\boldsymbol{y}, t)$ according to the following procedure:

1. Compute symmetrized matrix $\boldsymbol{A} = (1/2)(\boldsymbol{y} + \boldsymbol{y}^{\mathsf{T}})$.
2. Compute top eigenvalue $\lambda_1(\boldsymbol{A})$.
3. Return 1 if $\lambda_1(\boldsymbol{A}) \geq t/2$, and 0 otherwise.

Under the true model $\boldsymbol{y}_t = t\boldsymbol{x} - \boldsymbol{B}_t$, we have:

$$\lambda_1(\boldsymbol{A}_t) \geq t - \|\boldsymbol{W}_t\|_{\mathrm{op}} \geq t - t^{2/3}$$

with error probability given by

$$\mathbb{P}\left(\|\boldsymbol{W}_t\|_{\mathrm{op}} \geq t^{2/3}\right) \leq C \exp\left(-ct^{1/3}\right),$$

for constants $C, c > 0$. Thus, we get that

$$J_n(t_2, \infty) \leq 2 \int_{t_2}^{\infty} \mathbb{P}(\|\boldsymbol{W}_t\|_{\mathrm{op}} \geq t^{2/3}) \, \mathrm{d}t \leq C_* \int_{t_2}^{\infty} e^{-ct^{1/3}} \, \mathrm{d}t$$

$$\leq C_{**} e^{-ct_2^{1/3}} \leq C_{**} e^{-cn^{4/3}}. \tag{34}$$

Claim M2 of Theorem 3.2 follows from Eqs. (29), (33), (34).

We finally consider claim M3. Equation (4) yields, for every $\ell \geq 1$:

$$\hat{\boldsymbol{y}}_{\ell\Delta} = \Delta \sum_{i=1}^{\ell} \hat{\boldsymbol{m}}(\hat{\boldsymbol{y}}_{(i-1)\Delta}, (i-1)\Delta) + \sqrt{\Delta} \sum_{i=1}^{\ell} \boldsymbol{g}_{i\Delta}. \tag{35}$$

We define

$$\bar{\boldsymbol{m}}_{t_1} := \Delta \sum_{i=1}^{\ell_1} \hat{\boldsymbol{m}}(\hat{\boldsymbol{y}}_{(i-1)\Delta}, (i-1)\Delta), \,, \quad \ell_1 := \lfloor t_{\mathrm{alg}}(1+\delta)/\Delta \rfloor. \tag{36}$$

We further define the following auxiliary process for $t \geq \ell_1\Delta = \lfloor t_1/\Delta \rfloor \Delta$:

$$\tilde{\boldsymbol{y}}_t = \bar{\boldsymbol{m}}_{t_1} + \boldsymbol{B}_t, \tag{37}$$

where $(\boldsymbol{B}_t : t \geq 0)$ is a BM such that $\boldsymbol{B}_{j\Delta} = \sqrt{\Delta} \sum_{i=1}^{k} \boldsymbol{g}_{i\Delta}$. In particular, $\tilde{\boldsymbol{y}}_{\ell_1\Delta} = \hat{\boldsymbol{y}}_{\ell_1\Delta}$.

From triangle inequality, using Assumption 1, the condition $\|\hat{\boldsymbol{m}}_0(\boldsymbol{y}, t)\|_F \leq 1$, and that by construction $\|\hat{\boldsymbol{m}}(\boldsymbol{y}, t)\|_F \leq \varepsilon$ for $t \leq (1-\gamma)t_{\mathrm{alg}}$, we get

$$\|\bar{\boldsymbol{m}}\|_F \leq \varepsilon(1-\gamma)t_{\mathrm{alg}} + (\gamma+\delta)t_{\mathrm{alg}}.$$

We claim that (with high probability) $\phi_1(\tilde{\boldsymbol{y}}_t, t) = 0$ simultaneously for all $t \in [t_1, t_2]$. As a consequence, recalling the definition of $\hat{\boldsymbol{m}}$, we obtain that $\tilde{\boldsymbol{y}}_t = \hat{\boldsymbol{y}}_t$ for all $t \in [t_1, t_2]$. In order to prove the claim, define, for $\ell \geq 1$:

$$\mathcal{T}_n := \left\{ t_\ell^+ = t_1 + \frac{\ell-1}{n} : \ell \in \mathbb{N} \right\} \cap [t_1, t_2].$$

For every $t \in [t_1, t_2]$, consider the thresholded vector $\hat{\boldsymbol{v}}_t$ in the definition of the test $\phi_1$. We know that

$$\langle \hat{\boldsymbol{v}}_t, \tilde{\boldsymbol{y}}_t \hat{\boldsymbol{v}}_t \rangle = \langle \hat{\boldsymbol{v}}_t, \bar{\boldsymbol{m}} \hat{\boldsymbol{v}}_t \rangle + \langle \hat{\boldsymbol{v}}_t, \boldsymbol{B}_t \hat{\boldsymbol{v}}_t \rangle \leq \varepsilon(1-\gamma)t_{\mathrm{alg}} + (\gamma+\delta)t_{\mathrm{alg}} + \max_{\boldsymbol{v} \in \Omega_{n,k}} \langle \boldsymbol{v}, \boldsymbol{B}_t \boldsymbol{v} \rangle. \tag{38}$$

Using Lemma H.1, we get that

$$\mathbb{P}\left( \max_{\boldsymbol{v} \in \Omega_{n,k}} |\langle \boldsymbol{v}, \boldsymbol{B}_t \boldsymbol{v} \rangle| \geq C \sqrt{\log \binom{n}{k}} \sqrt{t} \right) \leq 2 \binom{n}{k}^{-C^2/2+2}.$$

Note that $|\mathcal{T}_n| \leq n^5$, whence

$$\mathbb{P}\left(\exists t \in \mathcal{T}_n : \max_{\boldsymbol{v} \in \Omega_{n,k}} |\langle \boldsymbol{v}, \boldsymbol{W}_{t_\ell} \boldsymbol{v}\rangle| \geq C\sqrt{\log\binom{n}{k}}\sqrt{t_\ell}\right) \leq 2\binom{n}{k}^{-C^2/2+2} n^5.$$

For $t \in [t_\ell, t_{\ell+1}]$, we have

$$\max_{t_\ell \leq t \leq t_{\ell+1}} \left\{ \max_{\boldsymbol{v} \in \Omega_{n,k}} |\langle \boldsymbol{v}, \boldsymbol{W}_t \boldsymbol{v}\rangle| - \max_{\boldsymbol{v} \in \Omega_{n,k}} |\langle \boldsymbol{v}, \boldsymbol{W}_{t_\ell} \boldsymbol{v}\rangle| \right\} \leq \max_{t_\ell \leq t \leq t_{\ell+1}} \|\boldsymbol{W}_t - \boldsymbol{W}_{t_\ell}\|_{\mathrm{op}}.$$

Using Lemma F.3, we get that $\max_{t_\ell \leq t \leq t_{\ell+1}} \|\boldsymbol{W}_t - \boldsymbol{W}_{t_\ell}\|_{\mathrm{op}} \leq 16\sqrt{(t_{\ell+1} - t_\ell)n} = 16$ with probability at least $1 - 2\exp(-32n)$. Taking a union bound over $\mathcal{T}_n$, we get that

$$\mathbb{P}\left(\exists t \in [t_1, t_2] : \max_{\boldsymbol{v} \in \Omega_{n,k}} |\langle \boldsymbol{v}, \boldsymbol{W}_t \boldsymbol{v}\rangle| \geq C\sqrt{\log\binom{n}{k}}\sqrt{t}\right) \leq 2\binom{n}{k}^{-C^2/2+2} n^5 + 2n^5 e^{-32n},$$

for possibly a different constant $C > 0$.

Using Eq. (38) and the last estimate, we obtain that

$$\mathbb{P}\left(\exists t \in [t_1, t_2] : \langle \hat{\boldsymbol{v}}_t, \tilde{\boldsymbol{y}}_t \hat{\boldsymbol{v}}_t\rangle \geq b_n t_{\mathrm{alg}} + C\sqrt{\log\binom{n}{k}}\sqrt{t}\right) \leq 2\binom{n}{k}^{-C^2/2+2} n^5 + 2n^5 \exp(-32n), \tag{39}$$

where $b_n := \varepsilon(1-\gamma) + (\gamma + \delta_n)$. Since $\delta_n = o_n(1)$, we have $b_n \to (1-\gamma)\varepsilon + \gamma$. Further, by choosing $\gamma, \varepsilon$ small enough, we get that $\limsup_n b_n < c/2$, say. Hence, we get that for $t \geq t_1$, and all $n$ large enough

$$b_n t_{\mathrm{alg}} + C\sqrt{\log\binom{n}{k}}\sqrt{t} < ct/2$$

for the constant $c$ of Assumption 2. Therefore Eq. (39) implies that $\phi_1(\tilde{\boldsymbol{y}}_t, t) = 0$ simultaneously for all $t \in [t_1, t_2]$, with high probability, by choosing $\beta > c$. We conclude that, with high probability $\hat{\boldsymbol{y}}_t = \tilde{\boldsymbol{y}}_t$ throughout $t \in [t_1, t_2]$.

Finally, we extend the analysis to $t \in [t_2, \infty)$ by proving that, with high probability, $\phi_2(\tilde{\boldsymbol{y}}_t, t) = 0$ and hence $\hat{\boldsymbol{y}}_t = \tilde{\boldsymbol{y}}_t$ for all $t \in [t_2, \infty)$. We use (for $\boldsymbol{A}_t = (\tilde{\boldsymbol{y}}_t + \tilde{\boldsymbol{y}}_t^{\mathsf{T}})/2$, $\boldsymbol{W}_t = (\boldsymbol{B}_t + \boldsymbol{B}_t^{\mathsf{T}})/2$)

$$\lambda_1(\boldsymbol{A}_t) \leq \varepsilon(1-\gamma)t_{\mathrm{alg}} + (\delta + \gamma)t_{\mathrm{alg}} + \lambda_1(\boldsymbol{W}_t) \tag{40}$$

Following exactly the argument as in the proof of Proposition E.1 (in particular, Subsection F.3.2), we get that $\lambda_1(\boldsymbol{W}_t) \leq t/3$ simultaneously for all $t \geq t_2$, with high probability. On this event, $\lambda_1(\boldsymbol{A}_t) < t/2$ with high probability (because $t/6 \gg (1-\gamma)t_{\mathrm{alg}}(1-\varepsilon) + (\gamma + \delta)t_{\mathrm{alg}}$). Hence, with high probability $\phi_2(\tilde{\boldsymbol{y}}_t, t) = 0$ and hence $\hat{\boldsymbol{y}}_t = \tilde{\boldsymbol{y}}_t$ for all $t \in [t_2, \infty)$.

We thus proved that, with high probability, $\hat{\boldsymbol{y}}_t = \tilde{\boldsymbol{y}}_t$ for all $t \geq t_1 = (1+\delta)t_{\mathrm{alg}}$, whence $\hat{\boldsymbol{m}}(\hat{\boldsymbol{y}}_t, t) = 0$ as well (because $\phi_1(\tilde{\boldsymbol{y}}_t, t) = 0$ for $t \in [t_1, t_2]$ and $\phi_2(\tilde{\boldsymbol{y}}_t, t) = 0$ for $t \in [t_2, \infty)$). Claim M3 thus follows.

# I  PROOF OF COROLLARY E.2

## I.1  PROPERTIES OF THE ESTIMATOR $\hat{\boldsymbol{m}}(\cdot)$

**Proposition I.1.** *Assume $(\log n)^2 \ll k \ll \sqrt{n}$ and let $\hat{\boldsymbol{m}}(\cdot)$ be the estimator defined in Algorithm 1. Recall that in this case, $t_{\mathrm{alg}} = k^2\log(n/k^2)$. Then for any $\delta > 0$ there exists $\varepsilon > 0$ such that, letting $s = \sqrt{(1+\varepsilon)\log(n/k^2)}$, we have*

$$\sup_{t \geq (1+\delta)t_{\mathrm{alg}}} \mathbb{P}(\hat{\boldsymbol{m}}(\boldsymbol{y}_t, t) \neq \boldsymbol{x}) = O(n^{-D}) \tag{41}$$

*for any fixed $D > 0$.*

The proof of this proposition is a modification of the one in Cai et al. (2017), and will be presented in Appendix K. Note that Proposition I.1 directly implies the first inequality in Condition M2v of Theorem E.2, as $\|\hat{\boldsymbol{m}}(\boldsymbol{y}_t, t) - \boldsymbol{x}\| \leq 2$.

By definition, when $t < t_{\mathrm{alg}}$, the algorithm will return $\hat{\boldsymbol{m}} = \boldsymbol{0}$, so we automatically have the following, which implies the second inequality in Condition M2v.

**Proposition I.2.** *For any fixed $\delta > 0$, and $t \leq (1 - \delta)t_{\mathrm{alg}}$, we have $\|\hat{\boldsymbol{m}}(\boldsymbol{y}_t, t) - \boldsymbol{x}\| = 1$.*

In the proof of Theorem E.2, we will also make use of the following estimates, whose proof is deferred to the Appendix N.

**Lemma I.3.** *Let $(\boldsymbol{w}_t : t \geq 0)$ be a process defined as*

$$\boldsymbol{w}_t = \frac{1}{2}\left\{(\boldsymbol{B}_t + \sqrt{\varepsilon}\boldsymbol{g}_t) + (\boldsymbol{B}_t + \sqrt{\varepsilon}\boldsymbol{g}_t)^{\mathsf{T}}\right\}$$

*where $\boldsymbol{B}, \boldsymbol{g}$ are independent $n^2$-dimensional BMs, and $0 \leq \varepsilon < 1$. Then, for any $0 \leq t_0 \leq t_1$, and $t \geq 0$, $s \geq 1$,*

$$\mathbb{P}\Big(\max_{t_0 \leq t \leq t_1} \|\boldsymbol{w}_t - \boldsymbol{w}_{t_0}\|_F \geq 4\sqrt{(t_1 - t_0)s} \cdot n\Big) \leq 2\,e^{-n^2 s/4}\,, \tag{42}$$

$$\mathbb{P}\big(\|\boldsymbol{w}_t\|_F \geq 4\sqrt{ts} \cdot n\big) \leq 2\,e^{-n^2 s/4}\,. \tag{43}$$

### I.2 Analysis of the diffusion process: Proof of Corollary E.2

We are left to prove that Condition M3v of Corollary E.2 holds.

For that purpose, we make the following choices about Algorithm 1:

(C1) We select the constants in the algorithm to be $\varepsilon_n = o_n(1)$ and $s_n = \sqrt{(1 + \varepsilon_n)\log(n/k^2)}$. We will use the shorthands $s = s_n$ and $\varepsilon = \varepsilon_n$, unless there is ambiguity.

(C2) The process $(\boldsymbol{g}_t)_{t \geq 0}$ used in Algorithm 1 follows a $n^2$-dimensional BM.

Note that Propositions I.1, I.2 hold under these choices, and in particular $\boldsymbol{g}_t \sim \mathsf{N}(\boldsymbol{0}, t\boldsymbol{I}_{n \times n})$ at all times. Also the sequence of random vectors $\boldsymbol{g}_{\ell\Delta}$, $\ell \in \mathbb{N}$ can be generated via $\boldsymbol{g}_{\ell\Delta} = \boldsymbol{g}_{(\ell-1)\Delta} + \sqrt{\Delta}\hat{\boldsymbol{g}}_\ell$, for some i.i.d. standard normal vectors $\{\hat{\boldsymbol{g}}_\ell\}_{\ell \geq 0}$.

Letting $(\boldsymbol{z}_t)_{t \geq 0}$ a standard BM in $\mathbb{R}^{n \times n}$, and $\hat{\boldsymbol{y}}_0 = \boldsymbol{0}$ we can rewrite the approximate diffusion (4) as follows (for $t \in \mathbb{N} \cdot \Delta$)

$$\hat{\boldsymbol{y}}_{t+\Delta} = \hat{\boldsymbol{y}}_t + \Delta \cdot \hat{\boldsymbol{m}}\left(\hat{\boldsymbol{y}}_t, t\right) + (\boldsymbol{z}_{t+\Delta} - \boldsymbol{z}_t)\,. \tag{44}$$

We further define

$$\boldsymbol{w}_t = \frac{1}{2}\left\{(\boldsymbol{z}_t + \sqrt{\varepsilon}\boldsymbol{g}_t) + (\boldsymbol{z}_t + \sqrt{\varepsilon}\boldsymbol{g}_t)^{\mathsf{T}}\right\}\,. \tag{45}$$

It is easy to see that $(c(\varepsilon)\boldsymbol{w}_t : t \geq 0)$ is a $\mathsf{GOE}(n)$ process for $c(\varepsilon) := ((1 + \varepsilon)/2)^{-1/2}$. The key technical estimate in the proof of Theorem E.2, Condition M3v is stated in the next proposition.

**Proposition I.4.** *Let $(\boldsymbol{w}_t : t \geq 0)$ be defined as per Eq. (45), and assume $\varepsilon_n = o_n(1)$ and $s_n = \sqrt{(1 + \varepsilon_n)\log(n/k^2)}$. Further, assume $k \geq C(\log n)^{5/2}$ for some sufficiently large absolute constant $C > 0$. Then*

$$\lim_{n \to \infty} \mathbb{P}\left\{\|\eta_s(\boldsymbol{w}_t/\sqrt{t})\|_{\mathrm{op}} \leq k + \sqrt{t}/s\ \forall t \geq 1\right\} = 1\,. \tag{46}$$

Before proving this proposition, let us show that it implies Condition M3v of Theorem E.2. Indeed we claim that, with high probability, for all $\ell \in \mathbb{N}$, $\hat{\boldsymbol{m}}(\hat{\boldsymbol{y}}_{\ell\Delta}, \ell\Delta) = \boldsymbol{0}$ and $\hat{\boldsymbol{y}}_{\ell\Delta} = \boldsymbol{z}_{\ell\Delta}$. This is proven by induction over $\ell$. Indeed, if it holds up to a certain $\ell - 1 \in \mathbb{N}$, then we have $\hat{\boldsymbol{y}}_{\ell\Delta} = \boldsymbol{z}_{\ell\Delta}$ by Eq. (44) whence it follows that $\boldsymbol{A}_{t,+} = \boldsymbol{w}_t/\sqrt{t}$, for $t = \ell\Delta$ (c.f. Algorithm 1, line 4) and therefore $\hat{\boldsymbol{m}}(\hat{\boldsymbol{y}}_t, t) = \boldsymbol{0}$ by Proposition I.4 (because the condition in Algorithm 1, line 6, is never passed).

We therefore have

$$\inf_{\ell \geq 0} W_1(\hat{\boldsymbol{m}}(\hat{\boldsymbol{y}}_{\ell\Delta}, \ell\Delta), \boldsymbol{x}) \geq \inf_{\ell \geq 0} \mathbb{P}(\hat{\boldsymbol{m}}(\hat{\boldsymbol{y}}_{\ell\Delta}, \ell\Delta) = \boldsymbol{0}) = 1 - o_n(1)\,. \tag{47}$$

This concludes the proof of Theorem E.2. We next turn to proving Proposition I.4.

*Proof of Proposition I.4.* We follow a strategy analogous to the proof of the Law of Iterated Logarithm. We choose a sparse sequence of time points $\{t_\ell\}_{\ell=1}^\infty$, and $(i)$ establish the statement jointly for these time points, and $(ii)$ control deviations in between. In particular, we consider

$$t_\ell = \left(1 + \frac{\ell - 1}{n^3}\right)^2$$

for all $\ell \geq 1$.

We first show that simultaneously for all $\ell \geq 1$, we have $\max_{i,j} \left|(\boldsymbol{w}_{t_\ell})_{ij}/\sqrt{t_\ell}\right| \leq 8t_\ell^{1/4}\sqrt{\log n}$. We have, by sub-gaussianity of $(\boldsymbol{w}_{t_\ell})_{ij}$ and a union bound (here we account also for the case where $i = j$, in which there is an inflated variance), along with $\varepsilon = o_n(1)$: using the bound $2xy \geq x + y$ when $x, y \geq 1$ for $x = t_\ell^{1/2}, y = \log n$. Taking a union bound once again over $\ell$, we have

$$\mathbb{P}\left(\exists \ell \geq 1, 1 \leq i, j \leq n : |(\boldsymbol{w}_{t_\ell})_{ij}| \geq 8t_\ell^{3/4}\sqrt{\log n}\right) \leq n^{-6} \cdot \sum_{\ell=0}^\infty \exp\left(-8 \cdot \left(1 + \frac{\ell}{n^3}\right)\right)$$

We have, as the summands form a decreasing function of $\ell$ integer:

$$\sum_{\ell=0}^\infty \exp\left(-8 \cdot \left(1 + \frac{\ell}{n^3}\right)\right) \leq C + \int_0^\infty \exp\left(-\frac{8x}{n^3}\right) \mathrm{d}x \leq Cn^3. \tag{48}$$

We thus obtain that

$$\mathbb{P}\left(\exists \ell \geq 1, 1 \leq i, j \leq n : |(\boldsymbol{w}_{t_\ell})_{ij}| \geq 8t_\ell^{3/4}\sqrt{\log n}\right) = O(n^{-3}). \tag{49}$$

The point of this calculation is that simultaneously for all $\ell \geq 1$, we can truncate the entries of $\eta_s(\boldsymbol{w}_{t_\ell}/\sqrt{t_\ell})$ by $8t_\ell^{1/4}\sqrt{\log n}$ without worry.

Namely, for each $\ell \geq 1$, $\vartheta_\ell = 8t_\ell^{1/4}\sqrt{\log n}$ we define $\tilde{\boldsymbol{w}}_{t_\ell} \in \mathbb{R}^{n \times n}$ by

$$(\tilde{\boldsymbol{w}}_{t_\ell})_{ij} := \begin{cases} \eta_s(\boldsymbol{w}_{t_\ell}/\sqrt{t_\ell}) & \text{if } |\eta_s(\boldsymbol{w}_{t_\ell}/\sqrt{t_\ell})| \leq \vartheta_\ell, \\ \vartheta_\ell & \text{if } \eta_s(\boldsymbol{w}_{t_\ell}/\sqrt{t_\ell}) > \vartheta_\ell, \\ -\vartheta_\ell & \text{if } \eta_s(\boldsymbol{w}_{t_\ell}/\sqrt{t_\ell}) < -\vartheta_\ell. \end{cases} \tag{50}$$

By Eq. (49), we have

$$\mathbb{P}\left(\exists \ell \geq 1 : \eta_s(\boldsymbol{w}_{t_\ell}/\sqrt{t_\ell}) \neq \tilde{\boldsymbol{w}}_{t_\ell}\right) = O(n^{-3}). \tag{51}$$

We have from Bandeira & van Handel (2016), for every $x \geq 0$:

$$\mathbb{P}\left(\|\tilde{\boldsymbol{w}}_{t_\ell}\|_{\mathrm{op}} \geq 4\sigma + x\right) \leq n\exp\left(-\frac{cx^2}{\sigma_\star^2}\right), \tag{52}$$

for some absolute constant $c > 0$, where

$$\sigma^2 := \max_{i \leq n} \sum_{j=1}^n \mathbb{E}\left[(\tilde{\boldsymbol{w}}_{t_\ell})_{ij}^2\right] \leq \sum_{j=1}^n \mathbb{E}\left[\eta_s\left(\frac{\boldsymbol{w}_{t_\ell}}{\sqrt{t_\ell}}\right)_{ij}^2\right], \tag{53}$$

$$\sigma_\star := \max_{i,j \leq n} |(\tilde{\boldsymbol{w}}_{t_\ell})_{ij}| \leq 8t_\ell^{1/4}\sqrt{\log n}. \tag{54}$$

It can be seen from an immediate Gaussian calculation that, for $i \neq j$ and $Z \sim \mathsf{N}(0, 1)$:

$$\begin{aligned}
\mathbb{E}\left[\eta_s\left(\frac{\boldsymbol{w}_{t_\ell}}{\sqrt{t_\ell}}\right)_{ij}^2\right] &= \int_0^\infty 4z \cdot \mathbb{P}\left(\sqrt{\frac{1+\varepsilon}{2}} Z \geq z + s\right) \mathrm{d}z \\
&\overset{(a)}{\leq} \int_0^\infty 4z \cdot \frac{1}{z+s} \cdot \exp\left(-\frac{(z+s)^2}{1+\varepsilon}\right) \mathrm{d}z \\
&\overset{(b)}{\ll} \frac{1}{s} \exp\left(-\frac{s^2}{1+\varepsilon}\right) \leq \exp\left(-\frac{s^2}{1+\varepsilon}\right)
\end{aligned}$$

Here in $(a)$ we employ the Mill's ratio bound, and $(b)$ follows from $z + s \geq s$ and $s \to \infty$.

Proceeding analogously for the diagonal entries of $\eta_s(\boldsymbol{w}_{t_\ell}/\sqrt{t_\ell})$, we obtain that $\sigma \ll \sqrt{n} \exp(-s^2/(2(1+\varepsilon))) = k$ by definition of $s$.

We set $x = k/3 + \sqrt{t_\ell}/(3s)$. Since $x \gg \sigma$, we have $4\sigma + x \leq (3/2)x$ if $n, k$ are sufficiently large. Using Eq. (52) we obtain that, for some universal constants $c, c', c'' > 0$:

$$
\mathbb{P}\left( \|\tilde{\boldsymbol{w}}_{t_\ell}\|_{\mathrm{op}} \geq \frac{k}{2} + \frac{\sqrt{t_\ell}}{2s} \right) \leq n \exp\left( -\frac{c\left(\frac{k}{3} + \frac{\sqrt{t_\ell}}{3s}\right)^2}{64 t_\ell^{1/2} \log n} \right) \overset{(a)}{\leq} n \exp\left( -\frac{c'k}{s \log n} - \frac{c'' t_\ell^{1/2}}{s^2 \log n} \right).
$$

In step $(a)$, we simply expand the squared term in the numerator and drop the quadratic term in $k$. Now, taking a union bound over $\ell \geq 1$, we get that (similar to Eq. (48))

$$
\mathbb{P}\left( \exists \ell \geq 1 : \|\tilde{\boldsymbol{w}}_{t_\ell}\|_{\mathrm{op}} \geq \frac{k}{2} + \frac{\sqrt{t_\ell}}{2s} \right) \leq n \exp\left( -\frac{c'k}{s \log n} \right) \sum_{\ell=1}^{\infty} \exp\left( -\frac{c'' t_\ell^{1/2}}{s^2 \log n} \right)
$$

$$
\leq n \exp\left( -\frac{c'k}{s \log n} \right) \left( O(1) + \int_0^{\infty} \exp\left( -\frac{c''x}{s^2 n^3 \log n} \right) \mathrm{d}x \right)
$$

$$
= O\left( \exp\left( -\frac{c'k}{s \log n} \right) \cdot s^2 n^4 \log n \right)
$$

$$
= o_n(1),
$$

where the last estimate holds if $k \geq C(\log n)^{5/2}$ for some sufficiently large $C > 0$. In conclusion, using the last display and Eq. (51) we have shown that

$$
\mathbb{P}\left( \exists \ell \geq 1 : \left\| \eta_s\left( \frac{\boldsymbol{w}_{t_\ell}}{\sqrt{t_\ell}} \right) \right\|_{\mathrm{op}} \geq \frac{k}{2} + \frac{\sqrt{t_\ell}}{2s} \right) = o_n(1). \tag{55}
$$

Now we control the in-between fluctuations. Noting that $\eta_s(\cdot)$ is a 1-Lipschitz function, we have the following crude bound:

$$
\max_{t_\ell \leq t \leq t_{\ell+1}} \left\| \eta_s\left( \frac{\boldsymbol{w}_t}{\sqrt{t}} \right) - \eta_s\left( \frac{\boldsymbol{w}_{t_\ell}}{\sqrt{t_\ell}} \right) \right\|_{\mathrm{op}} \leq \max_{t_\ell \leq t \leq t_{\ell+1}} \left\| \frac{\boldsymbol{w}_t}{\sqrt{t}} - \frac{\boldsymbol{w}_{t_\ell}}{\sqrt{t_\ell}} \right\|_F
$$

$$
\leq \frac{\max_{t_\ell \leq t \leq t_{\ell+1}} \|\boldsymbol{w}_t - \boldsymbol{w}_{t_\ell}\|_F}{\sqrt{t_\ell}} + \|\boldsymbol{w}_{t_\ell}\|_F \left( \frac{1}{\sqrt{t_\ell}} - \frac{1}{\sqrt{t_{\ell+1}}} \right).
$$

From Lemma I.3, we obtain that

$$
\mathbb{P}\left( \max_{t_\ell \leq t \leq t_{\ell+1}} \left\| \eta_s(\boldsymbol{w}_t/\sqrt{t}) - \eta_s(\boldsymbol{w}_{t_\ell}/\sqrt{t_\ell}) \right\|_{\mathrm{op}} \geq 4n \cdot \sqrt{t_{\ell+1} - t_\ell} + 4n \cdot \sqrt{t_\ell} \cdot \left( 1 - \sqrt{\frac{t_\ell}{t_{\ell+1}}} \right) \right) \leq 4 e^{-n^2 t_\ell/4}.
$$

By definition of $t_\ell$, simple algebra reveals that (we also use the fact that $n^{-1/2} \ll s^{-1}$):

$$
4n \cdot \sqrt{t_{\ell+1} - t_\ell} + 4n \cdot \sqrt{t_\ell} \cdot \left( 1 - \sqrt{\frac{t_\ell}{t_{\ell+1}}} \right) \leq \frac{\sqrt{t_\ell}}{2s}.
$$

By union bound over $\ell \geq 1$,

$$
\mathbb{P}\left( \exists \ell \geq 1 : \max_{t_\ell \leq t \leq t_{\ell+1}} \left\| \eta_s(\boldsymbol{w}_t/\sqrt{t}) - \eta_s(\boldsymbol{w}_{t_\ell}/\sqrt{t_\ell}) \right\|_{\mathrm{op}} \geq \frac{k}{2} + \frac{\sqrt{t_\ell}}{2s} \right)
$$

$$
\leq 4 \sum_{\ell=1}^{\infty} \exp\left( -\frac{n^2}{8} - \frac{t_\ell}{8} \right)
$$

$$
= 4 \exp(-n^2/8) \sum_{\ell=1}^{\infty} \exp\left( -\frac{1}{8}\left( 1 + \frac{\ell-1}{n^3} \right)^2 \right)
$$

$$\leq 4 \exp(-n^2/8) \left( O(1) + \int_0^\infty \exp\left(-\frac{x^2}{8n^6}\right) dx \right) = O(\exp(-n^2/8)n^3) = o(1) \,.$$

Using this estimate together with Eq. (55), we conclude that with high probability the following holds simultaneously for all $t \geq 1$. Letting $\ell$ be largest such that $t_\ell \leq t$:

$$\left\| \eta_s(\boldsymbol{w}_t/\sqrt{t}) \right\|_{\mathrm{op}} \leq \left\| \eta_s(\boldsymbol{w}_{t_\ell}/\sqrt{t_\ell}) \right\|_{\mathrm{op}} + \left\| \eta_s(\boldsymbol{w}_t/\sqrt{t}) - \eta_s(\boldsymbol{w}_{t_\ell}/\sqrt{t_\ell}) \right\|_{\mathrm{op}} \leq k + \frac{\sqrt{t_\ell}}{s} \leq k + \frac{\sqrt{t}}{s} \,,$$

and this finishes the proof. $\qquad\square$

We remark that Assumption (C2) in the proof above is technically not needed, meaning that the additional noise stream $\boldsymbol{g}_t$ can in fact be discarded entirely: an appropriate thresholding of $\boldsymbol{v}_t$, the top eigenvector of $\eta_s(\boldsymbol{A}_{t,+})$, as in Algorithm 2, will also suffice to satisfy all conditions of Theorem E.2, although $\boldsymbol{x}$ will not be recovered exactly; some $o(k)$ positions outside the support of $\boldsymbol{x}$ will also be chosen, at most. The reason for this is that the alignment $|\langle \boldsymbol{v}_t, \boldsymbol{u} \rangle| = 1 - o_n(1)$ already, from a close inspection of our proof of Proposition I.1. Regarding the proof of Proposition I.4 above, one can easily realize that even if $\varepsilon = 0$, it will go through without any modification. We choose to keep our formulation of Algorithm 1 as faithful to the original work of Cai et al. (2017) as possible to discuss a variety of approaches, and leave this to the interested reader.

## J    PROOF OF PROPOSITION 2.1

We take the first row of $\hat{\boldsymbol{z}}_1$, and let $A = \{z_{11}, \cdots, z_{1n}\}$. Let $r_j = \mathrm{rank}(z_{1j})$ denote the rank of $z_{1j}$ with respect to the elements of $A$. Then since $z_{1j} \sim \mathsf{N}(0,1)$ across $j$, the collection of the first $k$ ranks $A_k = \{r_1, \cdots, r_k\}$ constitutes a sample without replacement from $[n]$. Construct $\boldsymbol{v}$ a binary vector such that $v_i = 1$ if and only if $i \in A_k$, and let $\boldsymbol{u}$ be a randomized-sign vector version of $(1/\sqrt{k})\boldsymbol{v}$. Let

$$\hat{\boldsymbol{m}}(\boldsymbol{y}, t; \boldsymbol{g}_1) = \boldsymbol{u}\boldsymbol{u}^\mathsf{T} = \boldsymbol{x}' \tag{56}$$

then it is clear that $\hat{\boldsymbol{m}}(\boldsymbol{y}, t; \hat{\boldsymbol{z}}_1) \sim \boldsymbol{x}$ and is independent of $\boldsymbol{x}$ (as it is a function of only $\hat{\boldsymbol{z}}_1$). The identity from $(i)$ follows accordingly. To see that this error is clearly sub-optimal compared to polynomial time algorithms, observe that $\hat{\boldsymbol{m}} = \boldsymbol{0}$ is a polynomial time drift, which achieves error 1 at every $t$.

Point $(ii)$ also follows immediately. Indeed, for every $\ell \geq 0$,

$$W_1(\hat{\boldsymbol{m}}(\hat{\boldsymbol{y}}_{\ell\Delta}, \ell\Delta), \boldsymbol{x}) = W_1(\boldsymbol{x}', \boldsymbol{x}) = 0$$

Lastly, regarding point $(iii)$, note that since $\boldsymbol{x}'$ is not dependent on $t$, we have, for every $\ell \geq 1$,

$$\hat{\boldsymbol{y}}_{\ell\Delta} = \hat{\boldsymbol{y}}_{(\ell-1)\Delta} + \Delta\boldsymbol{x}' + \sqrt{\Delta}\hat{\boldsymbol{z}}_{\ell\Delta}$$

Simple induction gives

$$\hat{\boldsymbol{y}}_{\ell\Delta} = (\ell\Delta)\boldsymbol{x}' + \sqrt{\Delta}\sum_{j=1}^{\ell} \hat{\boldsymbol{z}}_{j\Delta}$$

We take the coupling of $(\hat{\boldsymbol{y}}_{\ell\Delta}/(\ell\Delta), \boldsymbol{x})$ such that $\boldsymbol{x}' = \boldsymbol{x}$. Then by definition of the Wasserstein-2 metric,

$$W_2(\hat{\boldsymbol{y}}_{\ell\Delta}/(\ell\Delta), \boldsymbol{x})^2 \leq \mathbb{E}\left[ \left\| \frac{\sqrt{\Delta}\sum_{j=1}^{\ell} \hat{\boldsymbol{z}}_{j\Delta}}{\ell\Delta} \right\|^2 \right] = \frac{n^2}{\ell\Delta}$$

It is clear that as $\ell \to \infty$, this quantity converges to 0. Hence we are done.

## K    PROOF OF PROPOSITION I.1

We conduct our analysis conditional on $\boldsymbol{u}$ (recall that $\boldsymbol{x} = \boldsymbol{u}\boldsymbol{u}^\mathsf{T}$), and $S$ be the support of $\boldsymbol{u}$. Let $\boldsymbol{A}_0 = \sqrt{t}\boldsymbol{u}\boldsymbol{u}^\mathsf{T}$ and notice that

$$\boldsymbol{A}_{t,+} = \boldsymbol{A}_0 + \sigma_+\boldsymbol{Z}\,, \qquad \boldsymbol{A}_{t,-} = \boldsymbol{A}_0 + \sigma_-\boldsymbol{W}\,, \tag{57}$$

where $\sigma_+^2 := (1+\varepsilon)/2$, $\sigma_-^2 := (1+\varepsilon)/(2\varepsilon)$, and $\boldsymbol{Z}, \boldsymbol{W} \sim \mathsf{GOE}(n)$ are independent random matrices.

We have

$$\eta_s(\boldsymbol{A}_{t,+}) = \boldsymbol{A}_0 + \eta_s(\sigma_+ \boldsymbol{Z}) + \mathbb{E}[\boldsymbol{B}] + \left(\boldsymbol{B} - \mathbb{E}[\boldsymbol{B}]\right), \tag{58}$$

where

$$B_{ij} = \eta_s\left(\sqrt{t} \cdot u_i u_j + \sigma_+ Z_{ij}\right) - \sqrt{t} \cdot u_i u_j - \eta_s(\sigma_+ Z_{ij}). \tag{59}$$

Our first order of business is to analyze $\mathbb{E}[\boldsymbol{B}]$. If $i \notin S$ or $j \notin S$, we have $\mathbb{E}[B_{ij}] = 0$. On the other hand, if $i, j \in S$, then

(i) **Case 1:** $u_i u_j = 1/k$

In this case, we have $\mathbb{E}[B_{ij}] = -b_0 + b_1 \mathbf{1}_{i=j}$ where (below $G \sim \mathsf{N}(0,1)$)

$$b_0 := -\mathbb{E}\left\{\eta_s\left(\frac{\sqrt{t}}{k} + \sigma_+ G\right) - \frac{\sqrt{t}}{k}\right\}, \quad b_1 := \mathbb{E}\left\{\eta_s\left(\frac{\sqrt{t}}{k} + \sqrt{2}\sigma_+ G\right) - \eta_s\left(\frac{\sqrt{t}}{k} + \sigma_+ G\right)\right\}. \tag{60}$$

Recalling that $\sigma_+$ is bounded and bounded away from 0 (without loss of generality we can assume $\varepsilon < 1/2$) and $s$, $\sqrt{t}/k - s$ grows with $n, k$, so that $\eta_s(\sqrt{t}/k + Z_{ij}) = \sqrt{t}/k + Z_{ij} - s$ with high probability; hence $|B_{ij} + s| = o_P(s)$ (as $Z_{ij} = o(s)$ with high probability). Noting that $|B_{ij}| \le 2s$, we get $b_0 = s(1 + o(1))$ and $b_1 = o(s)$ (distribution on diagonal is different).

(ii) **Case 2:** $u_i u_j = -1/k \Rightarrow i \ne j$

By a similar reasoning, we have $\mathbb{E}[B_{ij}] = b_0 = s(1 + o(1))$.

We can thus rewrite

$$\eta_s(\boldsymbol{A}_{t,+}) = (\sqrt{t} - kb_0) \cdot \boldsymbol{u}\boldsymbol{u}^\mathsf{T} + b_1 \cdot \boldsymbol{P}_S + \eta_s(\sigma_+ \boldsymbol{Z}) + \left(\boldsymbol{B} - \mathbb{E}[\boldsymbol{B}]\right), \tag{61}$$

where $(\boldsymbol{P}_S)_{ij} = 1$ if $i = j \in S$ and $= 0$ otherwise.

Next, we analyze the operator norm of $\eta_s(\sigma_+ \boldsymbol{Z})$. Let $\tilde{\boldsymbol{Z}} = (\tilde{Z}_{ij})_{i,j \le n}$ be defined as

$$\tilde{Z}_{ij} = \eta_s(\sigma_+ Z_{ij})\, \mathbf{1}(|\eta_s(\sigma_+ Z_{ij})| \le C \log n). \tag{62}$$

for some constant $C > 0$; we have $\max_{ij \le n} |Z_{ij}| \le C \log n$ with error probability at most $\exp(-c(\log n)^2) \ll n^{-D}$ for any fixed $D > 0$. We have $\tilde{\boldsymbol{Z}} = \eta_s(\boldsymbol{Z})$. By Bandeira & van Handel (2016), there exists an absolute constant $c > 0$ such that for every $u > 0$:

$$\mathbb{P}\left(\|\tilde{\boldsymbol{Z}}\|_{\mathrm{op}} \ge 4\sigma + u\right) \le n \exp\left(-\frac{cu^2}{L^2}\right), \tag{63}$$

where

$$\sigma^2 = \max_{i \le n} \sum_{j=1}^n \mathbb{E}[\eta_s(Z_{ij})^2], \tag{64}$$

$$L = \max_{i,j \le n} \|\tilde{Z}_{ij}\|_\infty \le C \log n. \tag{65}$$

An immediate Gaussian calculation yields, for $i \ne j$:

$$\mathbb{E}[\eta_s(\sigma_+ Z_{ij})^2] = \int_0^\infty 4z \cdot \mathbb{P}(\sigma_+ Z_{ij} \ge z + s)\mathrm{d}z \le C_1 e^{-s^2/(1+\varepsilon)}. \tag{66}$$

for some constant $C_1 > 0$.

Proceeding analogously for $\eta_s(\sigma_+ Z_{ii})$ and substituting in Eq. (63), we get $\sigma^2 \le 2C_1 n \exp\{-s^2/(1+\varepsilon)\}$. Applying Eq. (63) there exists an absolute constant $C, C' > 0$ such that, by taking $u = C'k$, with probability at least $1 - \exp(-ck^2/(\log n)^2)$,

$$\|\eta_s(\sigma_+ \boldsymbol{Z})\|_{\mathrm{op}} \le C\left(\sqrt{n}\exp\{-s^2/2(1+\varepsilon)\} \vee \sqrt{\log n}\right) \tag{67}$$

$$\leq C\left(k \vee \sqrt{\log n}\right) \leq Ck. \tag{68}$$

Note that the error probability is at most $\exp(-ck)$, because we already know that $k \gg (\log n)^2$.

Lastly, consider $\boldsymbol{B} - \mathbb{E}[\boldsymbol{B}]$. By Eq. (59) we know that the entries of this matrix are independent with mean $0$ and bounded by $2s$, hence subgaussian. Further only a $k \times k$ submatrix is nonzero, so that

$$\|\boldsymbol{B} - \mathbb{E}[\boldsymbol{B}]\|_{\mathrm{op}} \leq C_1 \sqrt{k}s, \tag{69}$$

with high probability (for instance, the operator norm tail bound above can be applied once more, which gives an error probability of at most $C \exp(-ck)$ for some absolute constant $C, c > 0$).

Summarizing, we proved that

$$\eta_s(\boldsymbol{A}_{t,+}) = (\sqrt{t} - kb_0) \cdot \boldsymbol{u}\boldsymbol{u}^{\mathsf{T}} + \boldsymbol{\Delta}, \tag{70}$$

$$\|\boldsymbol{\Delta}\|_{\mathrm{op}} \leq C(k + \sqrt{k}s) \leq C'k, \tag{71}$$

where in the last step we used $k \gg (\log n)^2$

Recall that $\boldsymbol{v}_t$ denotes the top eigenvector of $\eta_s(\boldsymbol{A}_{t,+})$. By Davis-Kahan,

$$\min_{a \in \{+1, -1\}} \|\boldsymbol{v}_t - a\boldsymbol{u}\| \leq \frac{Ck}{\sqrt{t} - kb_0} \tag{72}$$

$$\overset{(a)}{\leq} \frac{Ck}{\sqrt{t} - (1 + \varepsilon)ks} \tag{73}$$

$$\overset{(b)}{\leq} \varepsilon, \tag{74}$$

where in $(a)$ we used the fact that $b_0 = s + o(s)$ and in $(b)$ the fact that $t \geq (1 + \delta)k^2 \log(n/k^2)$, whereby we can assume $\delta \geq C\varepsilon$ for $C$ a sufficiently large absolute constant. Recalling the definition of the score $\hat{\boldsymbol{v}}_t$:

$$\hat{\boldsymbol{v}}_t = \boldsymbol{A}_{t,-}\boldsymbol{v}_t = \sqrt{t}\boldsymbol{u}\langle\boldsymbol{u}, \boldsymbol{v}_t\rangle + \sigma_-\boldsymbol{W}\boldsymbol{v}_t$$

where we know that $\boldsymbol{G} = \boldsymbol{W}\boldsymbol{v}_t \sim \mathsf{N}(0, \boldsymbol{I}_n)$ by independence of $\boldsymbol{W}$ and $\boldsymbol{v}_t$. Assuming to be definite that the sign of the eigenvector is chosen so that the last bound holds with $a = +1$, we get that $\langle\boldsymbol{u}, \boldsymbol{v}_t\rangle \geq 1 - \varepsilon^2$. We get that for every $j \in S$:

$$u_j > 0 \Rightarrow \hat{v}_{t,j} \geq (1 - \varepsilon^2)\sqrt{\frac{t}{k}} - \sigma_-|G_j| \geq (1 - \varepsilon^2)\sqrt{\frac{t}{k}} - \frac{C}{\varepsilon}\log n \tag{75}$$

$$u_j < 0 \Rightarrow \hat{v}_{t,j} \leq -(1 - \varepsilon^2)\sqrt{\frac{t}{k}} + \sigma_-|G_j| \leq -(1 - \varepsilon^2)\sqrt{\frac{t}{k}} + \frac{C}{\varepsilon}\log n \tag{76}$$

where we use a union bound to get $|G_j| \leq C\log n$ for all $j \leq n$, with probability at least $1 - \exp(-c(\log n)^2)$. Similarly, for all $i \notin S$,

$$|\hat{v}_{t,j}| \leq \sigma_-|G_j| \leq \frac{C}{\sqrt{\varepsilon}}\log n$$

These calculations reveal that: (i) the entries with the largest magnitudes are the elements of $S$, and (ii) if $u_i$ and $\hat{v}_{t,i}$ share the same sign for all $i \in S$. On this event, $\|\hat{\boldsymbol{m}}(\boldsymbol{y}_t, t) - \boldsymbol{x}\| = 0$.

Lastly, we claim that the top eigenvalue of $\eta_s(\boldsymbol{A}_{t,+})$ is larger than $k + \sqrt{t}/s$. From triangle inequality applied to Eq. (61), we have

$$\lambda_1(\eta_s(\boldsymbol{A}_{t,+})) \geq (\sqrt{t} - kb_0) - b_1 - \|\eta_s(\sigma_+\boldsymbol{Z})\|_{\mathrm{op}} - \|\boldsymbol{B} - \mathbb{E}[\boldsymbol{B}]\|_{\mathrm{op}} \tag{77}$$

$$\overset{(a)}{\geq} (\sqrt{t} - (1 + \varepsilon)ks) - Ck - C\sqrt{k}s \tag{78}$$

$$\overset{(b)}{\geq} \sqrt{t} - (1 + C_0\varepsilon)ks \tag{79}$$

$$\overset{(c)}{\geq} \varepsilon ks. \tag{80}$$

Here $(a)$ follows from Eqs. (68) and (69), $(b)$ because $k \gg 1$ and $s \gg 1$ and $(c)$ follows by taking $\delta \geq C\varepsilon$ for $C$ a sufficiently large absolute constant. The claim follows because $ks \gg k$ and $ks \gg \sqrt{t}$, and that $\exp(-c(\log n)^2)$ is a super-polynomially small rate.

# L  AUXILIARY LEMMAS FOR SECTION H

## L.1  PROOF OF LEMMA H.1

We let $\boldsymbol{B} \sim \mathsf{N}(\boldsymbol{0}, \boldsymbol{I}_{n^2})$ so that $\boldsymbol{W} = (\boldsymbol{B} + \boldsymbol{B}^\mathsf{T})/2$. For $\boldsymbol{v} \in \Omega_{n,k}$ we have $\langle \boldsymbol{v}, \boldsymbol{W}\boldsymbol{v} \rangle = \langle \boldsymbol{v}, \boldsymbol{B}\boldsymbol{v} \rangle \sim \mathsf{N}(0, 1)$. We thus have, by Gaussian tail bounds and a triangle inequality:

$$\mathbb{P}\left( |\langle \boldsymbol{v}, \boldsymbol{W}\boldsymbol{v} \rangle| \geq C\sqrt{\log\binom{n}{k}} \right) \leq 2\exp\left( -\frac{C^2}{2}\log\binom{n}{k} \right).$$

Taking the union bound over $\boldsymbol{v} \in \Omega_{n,k}$ gives the desired statement, since the cardinality of this set is $\binom{n}{k}2^k$.

## L.2  PROOF OF LEMMA H.3

Using Lemma H.2, we can take $x = n^{-1/4}$, say, and $\theta = \sqrt{1+\delta}$ for $\delta \geq n^{-c_0}$ for some small enough $c_0 > 0$, to get that

$$\mathbb{P}\left( \lambda_1(\boldsymbol{y}) \leq \theta + 1/\theta - n^{-1/4} - 2/n \right) \leq C\exp(-cn^{1/3}),$$

for some absolute constants $C, c > 0$.

We have the following identity, letting $\boldsymbol{W} \sim \mathsf{GOE}(n, 1/n)$:

$$\langle \boldsymbol{u}, \boldsymbol{v}_1 \rangle^2 = \frac{1}{\theta^2 \langle \boldsymbol{u}, (\lambda_1(\boldsymbol{y})\boldsymbol{I} - \boldsymbol{W})^{-2}\boldsymbol{u} \rangle} \geq \frac{1}{\theta^2 \cdot \|(\lambda_1(\boldsymbol{y})\boldsymbol{I} - \boldsymbol{W})^{-2}\|_{\mathrm{op}}}.$$

By standard Gaussian concentration, we know that, for any $\Delta > 0$

$$\mathbb{P}\left( \|\boldsymbol{W}\|_{\mathrm{op}} \geq 2 + \Delta \right) \leq C\exp(-cn\Delta^2).$$

In this inequality, we take

$$\Delta = \frac{1}{4}\left( \theta + 1/\theta - n^{-1/4} - 2/n - 2 \right).$$

Note that with $\theta = \sqrt{1+\delta}$ and $\delta = o_n(1)$, we know that $\theta + 1/\theta - 2 = \Theta(\delta^2)$, so that $\Delta = \Theta(\delta^2)$ if $\delta \geq n^{-c_0}$ with $c_0 \leq 1/8$. Hence, by a union bound on the two concentration inequalities,

$$\mathbb{P}\left( \lambda_{\min}(\lambda_1(\boldsymbol{y})\boldsymbol{I} - \boldsymbol{W}) \leq 2\Delta \right) \leq C\exp(-cn^{1/3})$$

and on the complement of this event, we know that

$$\langle \boldsymbol{v}_1, \boldsymbol{u} \rangle^2 \geq \frac{4\Delta^2}{\theta^2} = \Theta(\delta^4)$$

since $\theta = \Omega(1)$, and so $|\langle \boldsymbol{v}_1, \boldsymbol{u} \rangle| = \Omega(\delta^2)$.

# M  PROOF OF PROPOSITION F.1

In our proof, we will use the following elementary facts.

**Fact M.1.** *For any deterministic unit vector $\boldsymbol{u}$, a unit vector $\boldsymbol{v}$ is uniformly random on the orthogonal subspace to $\boldsymbol{u}$ if and only if $\langle \boldsymbol{v}, \boldsymbol{u} \rangle = 0$ and $\boldsymbol{v} \overset{\mathrm{d}}{=} \boldsymbol{Q}\boldsymbol{v}$ for every orthogonal matrix $\boldsymbol{Q}$ such that $\boldsymbol{Q}\boldsymbol{u} = \boldsymbol{u}$.*

**Fact M.2.** *Let $\boldsymbol{A}$ be a symmetric matrix, and $\boldsymbol{u}$ a unit vector. Denote $\boldsymbol{B}_\alpha = \alpha\boldsymbol{u}\boldsymbol{u}^\mathsf{T} + \boldsymbol{A}$, and let $\boldsymbol{v}(\alpha)$ be a top eigenvector of $\boldsymbol{B}_\alpha$. Then $f(\alpha) = |\langle \boldsymbol{v}(\alpha), \boldsymbol{u} \rangle|$ is an increasing function of $\alpha > 0$.*

Let $\boldsymbol{u} \sim \mathrm{Unif}(\Omega_{n,k})$. Recall that $\boldsymbol{A}_t = \sqrt{t}\boldsymbol{u}\boldsymbol{u}^\mathsf{T} + \sqrt{t}\boldsymbol{W}$ where $\boldsymbol{W} \sim \mathsf{GOE}(n, 1/2)$.

We conduct our analysis conditional on $\boldsymbol{u}$. Let $\boldsymbol{v}_t$ be a top eigenvector of $\boldsymbol{A}_t$. For $t = (1+\delta)n/2$, $|\langle \boldsymbol{v}_t, \boldsymbol{u} \rangle| \overset{a.s.}{\to} \sqrt{\delta/(1+\delta)}$, so that with high probability $|\langle \boldsymbol{v}_t, \boldsymbol{u} \rangle| \geq \sqrt{\delta}/(2\sqrt{1+\delta})$. If $t \geq (1+\delta)n/2$, we can use Fact O.2 to obtain the same result. By choosing $\varepsilon$ such that $2\varepsilon <$

$\sqrt{\delta/(1+\delta)}$, we know from standard concentration of the alignment (Lemma F.6) that $|\langle \boldsymbol{v}_t, \boldsymbol{u} \rangle| \geq 2\varepsilon$ with probability at least $1 - \exp(-cn)$ for some $c > 0$ possibly dependent on $(\varepsilon, \delta)$.

By rotational invariance of $\boldsymbol{W}$, $\boldsymbol{w}_t := \dfrac{\boldsymbol{v}_t - \langle \boldsymbol{v}_t, \boldsymbol{u} \rangle \boldsymbol{u}}{\|\boldsymbol{v}_t - \langle \boldsymbol{v}_t, \boldsymbol{u} \rangle \boldsymbol{u}\|}$ is uniformly random on the orthogonal subspace to $\boldsymbol{u}$. hence, there exists $\boldsymbol{g} \sim \mathsf{N}(\boldsymbol{0}, \boldsymbol{I}_n)$, such that

$$\boldsymbol{w}_t \sim \frac{(\boldsymbol{I}_n - \boldsymbol{u}\boldsymbol{u}^\top)\boldsymbol{g}}{\|(\boldsymbol{I}_n - \boldsymbol{u}\boldsymbol{u}^\top)\boldsymbol{g}\|}.$$

Since $\|(\boldsymbol{I}_n - \boldsymbol{u}\boldsymbol{u}^\top)\boldsymbol{g}\| \sim \|\boldsymbol{g}'\|$ for some $\boldsymbol{g}' \sim \mathsf{N}(\boldsymbol{0}, \boldsymbol{I}_{n-1})$, we have, for some constant $c > 0$,

$$\mathbb{P}\left(\|(\boldsymbol{I}_n - \boldsymbol{u}\boldsymbol{u}^\top)\boldsymbol{g}\| \leq \frac{\sqrt{n}}{2}\right) \leq \exp(-cn).$$

Further, for every $1 \leq i \leq n$, we know that

$$|((\boldsymbol{I}_n - \boldsymbol{u}\boldsymbol{u}^\top)\boldsymbol{g})_i| \leq |g_i| + \|\boldsymbol{u}\|_\infty \cdot |\langle \boldsymbol{u}, \boldsymbol{g} \rangle| \leq |g_i| + \frac{|\langle \boldsymbol{u}, \boldsymbol{g} \rangle|}{\sqrt{k}}.$$

We next show that, with the claimed probability, only a few entries of $\boldsymbol{w}_t$ can have large magnitude. As a result, less than $\ell$ entries of $\boldsymbol{u}$ can be estimated incorrectly (with $\ell = 1$ if $k \ll n/\log n$).

Define $\ell = \lceil n \exp(-a_n \cdot n/k) \rceil \geq 1$ (with $a_n$ a sequence to be chosen later) and $g_{(\ell)}^{\mathrm{abs}}$ as the $\ell$-th largest value among the $|g_i|$'s. We have

$$\mathbb{P}(|\langle \boldsymbol{u}, \boldsymbol{g} \rangle| \geq \sqrt{n a_n}) \leq \exp\left(-\frac{n a_n}{2}\right).$$

Furthermore, from a union bound, we get that

$$\begin{aligned}
\mathbb{P}\left(g_{(\ell)}^{\mathrm{abs}} \geq \frac{2\sqrt{n a_n}}{\sqrt{k}}\right) &\leq \binom{n}{\ell} \cdot \exp\left(-\frac{2n\ell \cdot a_n}{k}\right) \\
&\leq \left(\frac{en}{\ell}\right)^\ell \exp\left(-\frac{2n\ell \cdot a_n}{k}\right) \\
&= \exp\left(-\frac{2n\ell \cdot a_n}{k} + \ell \log \frac{n}{\ell} + \ell\right).
\end{aligned}$$

By definition, we know that $\ell \geq \max\{n \exp(-a_n \cdot n/k), 1\}$, so that

$$\frac{2n a_n}{k} - \log \frac{n}{\ell} - 1 \geq \frac{2n a_n}{k} - \min\{\log n, a_n \cdot n/k\} - 1 \geq \frac{n a_n}{2k}$$

as long as $n a_n \gg k$. This means that

$$\mathbb{P}\left(g_{(\ell)}^{\mathrm{abs}} \geq \frac{2\sqrt{n a_n}}{\sqrt{k}}\right) \leq \exp\left(-\frac{n a_n}{2k}\right).$$

Define the set

$$\mathcal{A}_n(t) := \left\{i \leq n : |w_{ti}| \geq 6\sqrt{\frac{a_n}{k}}\right\}.$$

By the bounds above we have

$$\mathbb{P}(|\mathcal{A}_n(t)| \leq \ell - 1) \geq 1 - e^{-cn} - e^{-n a_n/2k} - e^{-n a_n/2}.$$

Suppose that $|\langle \boldsymbol{v}_t, \boldsymbol{u} \rangle| \geq 2\varepsilon$ also holds, and suppose without loss of generality that $\langle \boldsymbol{v}_t, \boldsymbol{u} \rangle \geq 2\varepsilon$. Then, we have (as long as $6\sqrt{a_n} \leq (9/10)\varepsilon$)

$$i \in S, i \notin \mathcal{A}_n(t)\, u_i > 0 \Rightarrow v_{ti} \geq \frac{\langle \boldsymbol{v}_t, \boldsymbol{u} \rangle}{\sqrt{k}} - \frac{6\sqrt{a_n}}{\sqrt{k}} > \frac{\varepsilon}{\sqrt{k}} \Rightarrow i \in \hat{S}, \mathrm{sign}(v_{ti}) > 0,,$$

$$i \in S, i \notin \mathcal{A}_n(t), u_i < 0 \Rightarrow v_{ti} \leq -\frac{\langle \boldsymbol{v}_t, \boldsymbol{u} \rangle}{\sqrt{k}} + \frac{6\sqrt{a_n}}{\sqrt{k}} < -\frac{\varepsilon}{\sqrt{k}} \Rightarrow i \in \hat{S}, \mathrm{sign}(v_{ti}) < 0.$$

Analogously, for $i \notin S, i \notin \mathcal{A}_n(t)$, we have

$$|v_{ti}| \leq \frac{6\sqrt{a_n}}{\sqrt{k}} < \frac{\varepsilon}{\sqrt{k}} .$$

and we obtain that at most $\ell - 1$ positions could be mis-identified.

Next, we show that the termination condition (Line 5, Algorithm 2) does not trigger for each $t \geq n^2$ (with high probability). We write

$$\boldsymbol{A}_t = \frac{\boldsymbol{y}_t + \boldsymbol{y}_t^\mathsf{T}}{2\sqrt{t}} = \sqrt{t}\boldsymbol{u}\boldsymbol{u}^\mathsf{T} + \left( \frac{\boldsymbol{B}_t + \boldsymbol{B}_t^\mathsf{T}}{2\sqrt{t}} \right)$$

From Weyl's inequality:

$$\lambda_1(\boldsymbol{A}_t) \geq \sqrt{t} - \left\| \frac{\boldsymbol{B}_t + \boldsymbol{B}_t^\mathsf{T}}{2\sqrt{t}} \right\|_{\mathrm{op}}$$

From standard operator norm results for GOE matrices (as $(\boldsymbol{B}_t + \boldsymbol{B}_t^\mathsf{T})/\sqrt{2t} \sim \mathsf{GOE}(n)$), we know that $\|(\boldsymbol{B}_t + \boldsymbol{B}_t^\mathsf{T})/(2\sqrt{t})\|_{\mathrm{op}} \leq 2\sqrt{n}$ with probability at least $1 - \exp(-cn)$, for some $c > 0$. Hence $\lambda_1(\boldsymbol{A}_t) \geq \sqrt{t} - 2\sqrt{n} > \sqrt{t}/2$ as $t \geq n^2 \gg n$.

We obtain that

$$\mathbb{E}\left[ \|\hat{\boldsymbol{m}}(\boldsymbol{y}_t, t) - \boldsymbol{x}\|^2 \right] = O\left( \frac{\ell - 1}{k} + \exp\left( -\frac{na_n}{k} \right) \right) = O\left( \exp(-n\varepsilon^2/64k) \right)$$

where we picked $a_n > \varepsilon^2/64$ satisfying the bounds outlined above, namely $(i)$ $6\sqrt{a_n} \leq 0.9\varepsilon$, and $(ii)$ $na_n \gg k$. Notice that $na_n/k \gg \log(na_n/k)$ if $na_n \gg k$, and $\ell - 1 \leq n \cdot \exp(-a_n \cdot n/k)$.

## N    PROOF OF LEMMA I.3

*Proof.* By the Markov Property, we know that $\max_{t_\ell \leq t \leq t_{\ell+1}} \|\boldsymbol{w}_t - \boldsymbol{w}_{t_\ell}\|_F \overset{d}{=} \max_{0 \leq t \leq t_{\ell+1} - t_\ell} \|\boldsymbol{w}_t\|_F$. By Gaussian concentration, we have

$$\mathbb{P}\left( \max_{0 \leq t \leq t_{\ell+1} - t_\ell} \|\boldsymbol{w}_t\|_F - \mathbb{E}\left[ \max_{0 \leq t \leq t_{\ell+1} - t_\ell} \|\boldsymbol{w}_t\|_F \right] \geq x \right) \leq 2\exp\left( -\frac{x^2}{4(t_{\ell+1} - t_\ell)} \right)$$

This can be proven, e.g. by discretizing the interval $[0, t_{\ell+1} - t_\ell]$ into $r$ equal-length intervals and employing standard Gaussian concentration on vectors (then pushing $r \to \infty$). As the argument is standard, we omit the proof for brevity.

Now we bound $\mathbb{E}\left[ \max_{0 \leq t \leq t_{\ell+1} - t_\ell} \|\boldsymbol{w}_t\|_F \right]$. We know that $\|\boldsymbol{w}_t\|_F$ is a non-negative submartingale, so that from Doob's inequality:

$$\mathbb{E}\left[ \max_{0 \leq t \leq t_{\ell+1} - t_\ell} \|\boldsymbol{w}_t\|_F^2 \right] \leq 4\mathbb{E}[\|\boldsymbol{w}_{t_{\ell+1} - t_\ell}\|_F^2] \leq 9(t_{\ell+1} - t_\ell)n^2$$

so that from Cauchy-Schwarz, $\mathbb{E}\left[ \max_{0 \leq t \leq t_{\ell+1} - t_\ell} \|\boldsymbol{w}_t\|_F \right] \leq 3\sqrt{t_{\ell+1} - t_\ell}n$. Hence as $t_\ell \geq 1$,

$$\mathbb{P}\left( \max_{t_\ell \leq t \leq t_{\ell+1}} \|\boldsymbol{w}_t - \boldsymbol{w}_{t_\ell}\|_F \geq 4\sqrt{(t_{\ell+1} - t_\ell)t_\ell} \cdot n \right) \leq 2\exp\left( -\frac{n^2 t_\ell}{4} \right)$$

The second tail bound follows immediately (at least, the proof would be analogous to the preceding display). $\square$

## O    PROOF OF PROPOSITION F.1

In our proof, we will use the following facts; since they are elementary, we omit the proof.

**Fact O.1.** *For any deterministic unit vector $\boldsymbol{u}$, a unit vector $\boldsymbol{v}$ is uniformly random on the orthogonal subspace to $\boldsymbol{u}$ if and only if $\langle \boldsymbol{v}, \boldsymbol{u} \rangle = 0$ and $\boldsymbol{v} \sim \boldsymbol{Q}\boldsymbol{v}$ for every orthogonal matrix $\boldsymbol{Q}$ such that $\boldsymbol{Q}\boldsymbol{u} = \boldsymbol{u}$.*

**Fact O.2.** *Let $\boldsymbol{A}$ be a symmetric matrix, and $\boldsymbol{u}$ a unit vector. Denote $\boldsymbol{B}_\alpha = \alpha \boldsymbol{u}\boldsymbol{u}^\mathsf{T} + \boldsymbol{A}$, and let $\boldsymbol{v}(\alpha)$ be a top eigenvector of $\boldsymbol{B}_\alpha$. Then $f(\alpha) = |\langle \boldsymbol{v}(\alpha), \boldsymbol{u}\rangle|$ is an increasing function of $\alpha > 0$.*

We suppose that $t \geq (1+\delta)^2 n/2$ instead of $(1+\delta)n/2$, for notational convenience. Let $\boldsymbol{u}$ be a unit vector of random signs generated from a uniformly random set $S \subset [n]$ of size $k$. Since scaling does not change the eigenvectors, we instead consider the matrix

$$\tilde{\boldsymbol{Y}}_t = \frac{\boldsymbol{A}_t}{\sqrt{n}} = \sqrt{\frac{t}{n}} \boldsymbol{u}\boldsymbol{u}^\mathsf{T} + \left( \frac{\boldsymbol{B}_t + \boldsymbol{B}_t^\mathsf{T}}{2\sqrt{tn}} \right) = \sqrt{\frac{t}{n}} \boldsymbol{u}\boldsymbol{u}^\mathsf{T} + \boldsymbol{W}_n$$

where it is clear that $\sqrt{2}\boldsymbol{W}_n \sim \mathsf{GOE}(n)$.

We conduct our analysis conditional on $\boldsymbol{u}$. Let $\boldsymbol{v}_t$ be a top eigenvector of $\tilde{\boldsymbol{Y}}_t$. We know that when $t = (1+\delta)n/2$, $|\langle \boldsymbol{v}_t, \boldsymbol{u}\rangle| \overset{a.s.}{\to} \sqrt{\frac{\delta}{1+\delta}}$, so that with high probability $|\langle \boldsymbol{v}_t, \boldsymbol{u}\rangle| \geq \frac{\sqrt{\delta}}{2\sqrt{1+\delta}}$. If $t \geq (1+\delta)n/2$, we can use Fact O.2 to obtain the same result. By choosing $\varepsilon$ such that $2\varepsilon < \sqrt{\delta/(1+\delta)}$, we know from standard concentration of the alignment (Lemma F.6) that $|\langle \boldsymbol{v}_t, \boldsymbol{u}\rangle| \geq 2\varepsilon$ with probability at least $1 - \exp(-cn)$ for some $c > 0$ possibly dependent on $(\varepsilon, \delta)$.

By rotational invariance of $\boldsymbol{W}_n$, we know that $\boldsymbol{v}_t \sim \boldsymbol{Q}\boldsymbol{v}_t$ for any orthogonal matrix $\boldsymbol{Q}$ such that $\boldsymbol{Q}\boldsymbol{u} = \boldsymbol{u}$. We obtain that $\boldsymbol{v}_t - \langle \boldsymbol{v}_t, \boldsymbol{u}\rangle \boldsymbol{u}$ also has this property, so by Fact O.1, we get that $\boldsymbol{w}_t = \frac{\boldsymbol{v}_t - \langle \boldsymbol{v}_t, \boldsymbol{u}\rangle \boldsymbol{u}}{\|\boldsymbol{v}_t - \langle \boldsymbol{v}_t, \boldsymbol{u}\rangle \boldsymbol{u}\|}$ is uniformly random on the orthogonal subspace to $\boldsymbol{u}$. We can write, with $\boldsymbol{g} \sim \mathsf{N}(\boldsymbol{0}, \boldsymbol{I}_n)$:

$$\boldsymbol{w}_t \sim \frac{(\boldsymbol{I}_n - \boldsymbol{u}\boldsymbol{u}^\mathsf{T})\boldsymbol{g}}{\|(\boldsymbol{I}_n - \boldsymbol{u}\boldsymbol{u}^\mathsf{T})\boldsymbol{g}\|}$$

We first deal with the denominator. From triangle inequality, we know that $\|(\boldsymbol{I}_n - \boldsymbol{u}\boldsymbol{u}^\mathsf{T})\boldsymbol{g}\| \geq \|\boldsymbol{g}\| - \|\boldsymbol{u}\boldsymbol{u}^\mathsf{T}\boldsymbol{g}\| = \|\boldsymbol{g}\| - |\langle \boldsymbol{u}, \boldsymbol{g}\rangle|$. Since $\langle \boldsymbol{u}, \boldsymbol{g}\rangle \sim \mathsf{N}(0, 1)$, we have from standard sub-exponential concentration on $\|\boldsymbol{g}\|$ that

$$\mathbb{P}\left( \|(\boldsymbol{I}_n - \boldsymbol{u}\boldsymbol{u}^\mathsf{T})\boldsymbol{g}\| \leq \frac{\sqrt{n}}{2} \right) \leq \exp(-cn)$$

for some constant $c > 0$. For every $1 \leq i \leq n$, we know that

$$|((\boldsymbol{I}_n - \boldsymbol{u}\boldsymbol{u}^\mathsf{T})\boldsymbol{g})_i| \leq |g_i| + \|\boldsymbol{u}\|_\infty \cdot |\langle \boldsymbol{u}, \boldsymbol{g}\rangle| \leq |g_i| + \frac{|\langle \boldsymbol{u}, \boldsymbol{g}\rangle|}{\sqrt{k}}$$

Define $\ell = \lceil n \exp(-a_n \cdot n/k)\rceil \geq 1$ and $g_{(\ell)}^{\text{abs}}$ as the $\ell$-th largest value among the $|g_i|$'s. We have

$$\mathbb{P}(|\langle \boldsymbol{u}, \boldsymbol{g}\rangle| \geq \sqrt{na_n}) \leq \exp\left( -\frac{na_n}{2} \right)$$

Furthermore, from a union bound, we get that

$$\mathbb{P}\left( g_{(\ell)}^{\text{abs}} \geq \frac{2\sqrt{na_n}}{\sqrt{k}} \right) \leq \binom{n}{\ell} \cdot \exp\left( -\frac{2n\ell \cdot a_n}{k} \right) \leq \left( \frac{en}{\ell} \right)^\ell \exp\left( -\frac{2n\ell \cdot a_n}{k} \right) = \exp\left( -\frac{2n\ell \cdot a_n}{k} + \ell \log \frac{n}{\ell} + \ell \right)$$

By definition, we know that $\ell \geq \max\{n \exp(-a_n \cdot n/k), 1\}$, so that

$$\frac{2na_n}{k} - \log \frac{n}{\ell} - 1 \geq \frac{2na_n}{k} - \min\{\log n, a_n \cdot n/k\} - 1 \geq \frac{na_n}{k}$$

as long as $na_n \gg k$. With probability at least $1 - \exp(-cn) - \exp(-na_n/k) - \exp(-na_n/2)$, we have that at most $\ell - 1$ positions $i$ in a "bad" set $A$ have $|w_{ti}| \geq 6\sqrt{a_n/k}$. Suppose that $|\langle \boldsymbol{v}_t, \boldsymbol{u}\rangle| \geq 2\varepsilon$ also holds, and suppose without loss of generality that $\langle \boldsymbol{v}_t, \boldsymbol{u}\rangle \geq 2\varepsilon$. Then, we have

$$i \in S, i \notin A, u_i > 0 \Rightarrow v_{ti} \geq \frac{\langle \boldsymbol{v}_t, \boldsymbol{u}\rangle}{\sqrt{k}} - \frac{6\sqrt{a_n}}{\sqrt{k}} > \frac{\varepsilon}{\sqrt{k}} \Rightarrow i \in \hat{S}, \text{sign}(v_{ti}) > 0$$

$$i \in S, i \notin A, u_i < 0 \Rightarrow v_{ti} \leq -\frac{\langle \boldsymbol{v}_t, \boldsymbol{u}\rangle}{\sqrt{k}} + \frac{6\sqrt{a_n}}{\sqrt{k}} < -\frac{\varepsilon}{\sqrt{k}} \Rightarrow i \in \hat{S}, \text{sign}(v_{ti}) < 0$$

as long as $6\sqrt{a_n} < 3\varepsilon/4$, say. Analogously, for $i \notin S, i \notin A$, we have

$$|v_{ti}| \leq \frac{6\sqrt{a_n}}{\sqrt{k}} < \frac{\varepsilon}{\sqrt{k}}$$

and we obtain that at most $\ell - 1$ positions could be mis-identified.

For completeness, we show that the termination condition (Line 5, Algorithm 2) does not trigger for each $t \geq n^2$ (with high probability). We write

$$\boldsymbol{A}_t = \frac{\boldsymbol{y}_t + \boldsymbol{y}_t^\mathsf{T}}{2\sqrt{t}} = \sqrt{t}\boldsymbol{u}\boldsymbol{u}^\mathsf{T} + \left( \frac{\boldsymbol{B}_t + \boldsymbol{B}_t^\mathsf{T}}{2\sqrt{t}} \right)$$

From Weyl's inequality:

$$\lambda_1(\boldsymbol{A}_t) \geq \sqrt{t} - \left\| \frac{\boldsymbol{B}_t + \boldsymbol{B}_t^\mathsf{T}}{2\sqrt{t}} \right\|_{\mathrm{op}}$$

From standard operator norm results for GOE matrices (as $(\boldsymbol{B}_t + \boldsymbol{B}_t^\mathsf{T})/\sqrt{2t} \sim \mathsf{GOE}(n)$), we know that $\|(\boldsymbol{B}_t + \boldsymbol{B}_t^\mathsf{T})/(2\sqrt{t})\|_{\mathrm{op}} \leq 2\sqrt{n}$ with probability at least $1 - \exp(-cn)$, for some $c > 0$. Hence $\lambda_1(\boldsymbol{A}_t) \geq \sqrt{t} - 2\sqrt{n} > \sqrt{t}/2$ as $t \geq n^2 \gg n$.

We obtain that

$$\mathbb{E}\left[ \|\hat{\boldsymbol{m}}(\boldsymbol{y}_t, t) - \boldsymbol{x}\|^2 \right] = O\left( \frac{\ell - 1}{k} + \exp\left( -\frac{na_n}{k} \right) \right) = O\left( \exp(-a_n \cdot n/k) \right)$$

where we notice that $na_n/k \gg \log(na_n/k)$ if $na_n \gg k$.

## P    PROOF OF LEMMA F.3

By Gaussian concentration, we have

$$\mathbb{P}\left( \max_{0 \leq t \leq t_{\ell+1} - t_\ell} \|\boldsymbol{W}_t\|_{\mathrm{op}} - \mathbb{E}\left[ \max_{0 \leq t \leq t_{\ell+1} - t_\ell} \|\boldsymbol{W}_t\|_{\mathrm{op}} \right] \geq x \right) \leq 2\exp\left( -\frac{x^2}{2(t_{\ell+1} - t_\ell)} \right)$$

This can be proven, e.g. by discretizing the interval $[0, t_{\ell+1} - t_\ell]$ into $r$ equal-length intervals and employing Gaussian concentration on vectors (then pushing $r \to \infty$). As the argument is standard, we omit the proof for brevity.

To evaluate $\mathbb{E}[\max_{0 \leq t \leq t_{\ell+1} - t_\ell} \|\boldsymbol{W}_t\|_{\mathrm{op}}]$, we recognize that $\|\boldsymbol{W}_t\|_{\mathrm{op}}$ is a submartingale, so that from Doob's inequality:

$$\mathbb{E}\left[ \max_{0 \leq t \leq t_{\ell+1} - t_\ell} \|\boldsymbol{W}_t\|_{\mathrm{op}}^2 \right] \leq 4\mathbb{E}\left[ \|\boldsymbol{W}_{t_{\ell+1} - t_\ell}\|_{\mathrm{op}}^2 \right]$$

Once again from Gaussian concentration,

$$\mathbb{P}\left( |\|\boldsymbol{W}_1\|_{\mathrm{op}} - \mathbb{E}[\|\boldsymbol{W}_1\|_{\mathrm{op}}]| \geq x \right) \leq 2\exp\left( \frac{-x^2}{2} \right)$$

so that $\mathbb{P}\left( |\|\boldsymbol{W}_{t_{\ell+1} - t_\ell}\|_{\mathrm{op}} - \mathbb{E}[\|\boldsymbol{W}_{t_{\ell+1} - t_\ell}\|_{\mathrm{op}}]| \geq x \right) \leq 2\exp\left( -x^2/(2(t_{\ell+1} - t_\ell)) \right)$. Hence $\|\boldsymbol{W}_{t_{\ell+1} - t_\ell}\|_{\mathrm{op}}$ is $(t_{\ell+1} - t_\ell)$-subgaussian, implying that $\mathrm{Var}(\|\boldsymbol{W}_{t_{\ell+1} - t_\ell}\|_{\mathrm{op}}) \leq 6(t_{\ell+1} - t_\ell)$. As $\mathbb{E}[\|\boldsymbol{W}_{t_{\ell+1} - t_\ell}\|_{\mathrm{op}}]^2 \sim 4(t_{\ell+1} - t_\ell)n$ (one can obtain this from the Bai-Yin Theorem along with sub-gaussianity, for instance), we get that $\mathbb{E}\left[ \|\boldsymbol{W}_{t_{\ell+1} - t_\ell}\|^2 \right] \leq 16(t_{\ell+1} - t_\ell)n$ eventually as $n$ gets large.

From Cauchy-Schwarz inequality, we get that

$$\mathbb{E}\left[ \max_{0 \leq t \leq t_{\ell+1} - t_\ell} \|\boldsymbol{W}_t\|_{\mathrm{op}} \right] \leq 8\sqrt{t_{\ell+1} - t_\ell}\sqrt{n}$$

We conclude that

$$\mathbb{P}\left( \max_{0 \leq t \leq t_{\ell+1} - t_\ell} \|\boldsymbol{W}_t\|_{\mathrm{op}} \geq 16\sqrt{(t_{\ell+1} - t_\ell)n} \right) \leq 2\exp(-32n)$$

## Q  PROOF OF LEMMA F.4

First, by orthogonal invariance of $\boldsymbol{W}_t$, we know that $\boldsymbol{v}_t$ is uniformly random over the unit sphere $\mathbb{S}^{n-1}$. We can write, using $\boldsymbol{g} \sim \mathsf{N}(0, \boldsymbol{I}_n)$, the following representation

$$\boldsymbol{v}_t \sim \frac{\boldsymbol{g}}{\|\boldsymbol{g}\|}$$

As in the statement of the Lemma, we define the following set, for $\boldsymbol{v} \in \mathbb{R}^n$ and $C > 0$:

$$A(\boldsymbol{v}; C) = \left\{ i : 1 \leq i \leq n, |v_i| \geq \frac{C\sqrt{\log(n/k)}}{\sqrt{n}} \right\}$$

As with the proof of Proposition F.1, we first deal with the denominator $\|\boldsymbol{g}\|$: indeed, sub-exponential concentration gives us

$$\mathbb{P}\left( \sum_{j=1}^n g_j^2 \leq \frac{n}{2} \right) \leq 2\exp(-n/8) \tag{81}$$

This leads us to define another set

$$B(\boldsymbol{g}; C) = \left\{ i : 1 \leq i \leq n, |g_i| \geq C\sqrt{\log(n/k)} \right\}$$

Let $p_n = \mathbb{P}(|g_1| \geq C\sqrt{\log(n/k)})$, then we have $|B(\boldsymbol{g}; C)| \sim \mathrm{Bin}(n, p_n)$. From Gaussian tail bounds, we know that $p_n \leq (n/k)^{-C^2/2}$. We now use a Chernoff bound of the following form: for every $x \geq 4\mathbb{E}[X]$, where $X \sim \mathrm{Bin}(n, p)$, then

$$\mathbb{P}(X \geq x) \leq \exp(-x/3)$$

It is clear that $np_n \ll k^2/n \leq \max\{k^2/n, \sqrt{k}\}$ when $C > 2$, so that we have

$$\mathbb{P}\left( |B(\boldsymbol{g}; C)| \geq \max\{\sqrt{k}, k^2/n\} \right) \leq \exp\left( -\frac{1}{3}\max\{\sqrt{k}, k^2/n\} \right) \leq \exp\left( -\frac{1}{3}n^{1/4} \right)$$

Therefore, with each fixed $t$, by union bound with probability at least $1 - O(\exp(-\sqrt{n}))$, we have, for a possibly different $C > 0$, $|A(\boldsymbol{v}_t; C)| \leq \max\{\sqrt{k}, k^2/n\}$. Our proof ends here, as $\max\{\sqrt{k}, k^2/n\} \ll k/2$ for $\sqrt{n} \ll k \ll n$.

## R  PROOF OF LEMMA F.5

We know that

$$\begin{aligned}
\boldsymbol{v}_t^\mathsf{T} \boldsymbol{W}_{t_\ell} \boldsymbol{v}_t &= \boldsymbol{v}_t^\mathsf{T} \boldsymbol{W}_t \boldsymbol{v}_t - \boldsymbol{v}_t^\mathsf{T}(\boldsymbol{W}_t - \boldsymbol{W}_{t_\ell})\boldsymbol{v}_t \\
&= \lambda_1(\boldsymbol{W}_t) - \boldsymbol{v}_t^\mathsf{T}(\boldsymbol{W}_t - \boldsymbol{W}_{t_\ell})\boldsymbol{v}_t \\
&= \lambda_1(\boldsymbol{W}_{t_\ell}) - \boldsymbol{v}_t^\mathsf{T}(\boldsymbol{W}_t - \boldsymbol{W}_{t_\ell})\boldsymbol{v}_t + (\lambda_1(\boldsymbol{W}_t) - \lambda_1(\boldsymbol{W}_{t_\ell}))
\end{aligned}$$

from which we obtain from Weyl's inequality that

$$\sup_{t_\ell \leq t \leq t_{\ell+1}} \left| \boldsymbol{v}_t^\mathsf{T} \boldsymbol{W}_{t_\ell} \boldsymbol{v}_t - \lambda_1(\boldsymbol{W}_{t_\ell}) \right| \leq 2 \sup_{t_\ell \leq t \leq t_{\ell+1}} \|\boldsymbol{W}_t - \boldsymbol{W}_{t_\ell}\|_{\mathrm{op}} \leq 32\sqrt{(t_{\ell+1} - t_\ell)n}$$

with probability at least $1 - 2\exp(-32n)$.

## S  PROOF OF LEMMA F.6

By Weyl's inequality, $\boldsymbol{W} \mapsto \lambda_1(\boldsymbol{Y})$ (with $\boldsymbol{Y} = \theta\boldsymbol{v}\boldsymbol{v}^\mathsf{T} + \boldsymbol{W}$) is a 1-Lipschitz function and therefore, by Borell inequality (and Baik et al. (2005)), letting $\lambda_*(\theta) := \theta + 1/\theta$, for any $\varepsilon > 0$,

$$\mathbb{P}\left( |\lambda_1(\boldsymbol{Y}) - \lambda_*(\theta)| \geq \varepsilon \right) \leq 2e^{-n\varepsilon^2/4}. \tag{82}$$

To prove concentration of $\langle \boldsymbol{v}_1(\boldsymbol{Y}), \boldsymbol{v}\rangle^2$, note that simple linear algebra yields

$$\frac{1}{\langle \boldsymbol{v}_1(\boldsymbol{Y}), \boldsymbol{v}\rangle^2} = \langle \boldsymbol{v}, (\lambda_1(\boldsymbol{Y})\boldsymbol{I} - \boldsymbol{W})^{-2}\boldsymbol{v}\rangle =: F(\boldsymbol{W}). \tag{83}$$

It is therefore sufficient to prove that $F(\boldsymbol{W})$ concentrates around a value that is bounded away from 0. Fix $\varepsilon_0 > 0$ such that $2 + 3\varepsilon_0 < \lambda_*(\theta)$ and define the event

$$\mathcal{E} := \left\{ \boldsymbol{W} : \|\boldsymbol{W}\|_{\mathrm{op}} \leq 2 + \varepsilon_0 \,, \ |\lambda_1(\boldsymbol{Y}) - \lambda_*| \leq \varepsilon_0 \right\}. \tag{84}$$

By the Bai-Yin law and Gaussian concentration (plus the above concentration of $\lambda_1$), $\mathbb{P}(\mathcal{E}) \geq 1 - 2e^{-c(\varepsilon_0)n}$ for some $c(\varepsilon_0) > 0$. Further, it is easy to check that $F(\boldsymbol{W})$ is Lipschitz on $\mathcal{E}$, whence the concentration of $\langle \boldsymbol{u}, \boldsymbol{v}_1(\boldsymbol{W})\rangle^2$ follows by another application of Borell inequality.

# T    PROOFS OF REDUCTION RESULTS

## T.1    PROOF OF THEOREM 2

We state and prove a more detailed version of Theorem 2.

**Theorem 4.** *Assume that $\hat{\boldsymbol{m}}(\,\cdot\,, \cdot\,)$ has complexity $\chi$ and that for any $T \leq \theta d$, $D_{\mathrm{KL}}(\overline{\mathrm{P}}_{\hat{\boldsymbol{y}}}^{T,\Delta} \| \mathrm{P}_{\boldsymbol{y}}^T) \leq \varepsilon$ (where $\overline{\mathrm{P}}_{\hat{\boldsymbol{y}}}^{T,\Delta}$ is the continuous time process obtained by Brownian-linear interpolation of Eq. (4)).*

*Then for any $\sigma > 0$ there exists an algorithm with complexity $O(\chi \cdot T/\Delta)$, that takes as input $\boldsymbol{y} = \boldsymbol{x} + \sigma\boldsymbol{g}$, $(\boldsymbol{x}, \boldsymbol{g}) \sim \mu \otimes \mathsf{N}(0, \boldsymbol{I})$, and outputs $\hat{\boldsymbol{x}}$, such that*

$$\mathbb{E}\|\mathrm{P}_{\boldsymbol{x}|\boldsymbol{y}} - \mathrm{P}_{\hat{\boldsymbol{x}}|\boldsymbol{y}}\|_{\mathrm{TV}} \leq \sqrt{2\varepsilon} + \varepsilon_0(\theta) =: \overline{\varepsilon}, \tag{85}$$

*where $\varepsilon_0(\theta) := \mathbb{E}\|\mathrm{P}_{\boldsymbol{x}|\boldsymbol{y}} - \mathsf{N}(0, (\theta d)^{-1}\boldsymbol{I}_d) * \mathrm{P}_{\boldsymbol{x}|\boldsymbol{y}}\|_{\mathrm{TV}}$ is the expected TV distance between $\mathrm{P}_{\boldsymbol{x}|\boldsymbol{y}}$ and the convolution of $\mathrm{P}_{\boldsymbol{x}|\boldsymbol{y}}$ with a Gaussian with variance $1/(\theta d)$. As a consequence, there exists a randomized algorithm $\hat{\boldsymbol{m}}_+$ with complexity $(N\chi \cdot T/\Delta)$ that approximates the posterior expectation:*

$$\mathbb{E}\big\{ \|\hat{\boldsymbol{m}}_+(\boldsymbol{y}) - \boldsymbol{m}(\boldsymbol{y})\|^2 \big\} \leq 2\overline{\varepsilon} + 2N^{-1}. \tag{86}$$

*Proof.* The algorithm consists in running the discretized diffusion (4) with initialization $\hat{\boldsymbol{y}}_{t_0} = \boldsymbol{y}/\sigma^2$ at $t = t_0 := 1/\sigma^2$. To avoid notational burden, we will assume $(T - t_0)/\Delta$ to be an integer. Let $\hat{\boldsymbol{y}}_{t_0}^*$ be generated by the discretized diffusion with initialization at $\hat{\boldsymbol{y}}_0$ at $t = 0$. Note that the distribution of $\hat{\boldsymbol{y}}_{t_0}$ is the same as the one of $t_0\boldsymbol{x} + \sqrt{t_0}\boldsymbol{g}$ and hence by Assumption $(b)$, and Pinsker's inequality

$$\|\mathrm{P}_{\hat{\boldsymbol{y}}_{t_0}} - \mathrm{P}_{\hat{\boldsymbol{y}}_{t_0}^*}\|_{\mathrm{TV}} \leq \sqrt{\frac{1}{2}D_{\mathrm{KL}}(\mathrm{P}_{\hat{\boldsymbol{y}}_{t_0}^*} \| \mathrm{P}_{\hat{\boldsymbol{y}}_{t_0}})} \leq \sqrt{\frac{1}{2}D_{\mathrm{KL}}(\overline{\mathrm{P}}_{\hat{\boldsymbol{y}}}^{T,\Delta} \| \mathrm{P}_{\boldsymbol{y}}^T)} \leq \sqrt{\frac{\varepsilon}{2}}. \tag{87}$$

Hence $\hat{\boldsymbol{y}}_{t_0}, \hat{\boldsymbol{y}}_{t_0}^*$ can be coupled so that $\mathbb{P}(\hat{\boldsymbol{y}}_{t_0} \neq \hat{\boldsymbol{y}}_{t_0}^*) \leq \sqrt{\varepsilon/2}$.

We extend this to a coupling of $(\hat{\boldsymbol{y}}_t^*)_{t_0 \leq t \leq T}$ and $(\hat{\boldsymbol{y}}_t)_{t_0 \leq t \leq T}$ in the obvious way: we generate the two trajectories according to the discretized diffusion (4) with the same randomness $\hat{\boldsymbol{z}}_t$. Therefore $\mathbb{P}(\hat{\boldsymbol{y}}_T \neq \hat{\boldsymbol{y}}_T^*) \leq \sqrt{\varepsilon/2}$. Another application of the assumption $D_{\mathrm{KL}}(\overline{\mathrm{P}}_{\hat{\boldsymbol{y}}}^{T,\Delta} \| \mathrm{P}_{\boldsymbol{y}}^T) \leq \varepsilon$ and Pinsker's inequality yields $\mathbb{P}(\boldsymbol{y}_T \neq \hat{\boldsymbol{y}}_T^*) \leq \sqrt{\varepsilon/2}$, for $\boldsymbol{y}_T \stackrel{\mathrm{d}}{=} T\boldsymbol{x} + \sqrt{T}\boldsymbol{g}'$ with $(\boldsymbol{x}, \boldsymbol{g}') \sim \mu \otimes \mathsf{N}(0, \boldsymbol{I})$. We conclude by triangle inequality $\mathbb{P}(\boldsymbol{y}_T \neq \hat{\boldsymbol{y}}_T) \leq 2\sqrt{\varepsilon/2}$, which coincides with the claim (85).

Finally, Eq. (86) follows by generating $N$ i.i.d. copies $\hat{\boldsymbol{x}}_1, \ldots, \hat{\boldsymbol{x}}_N$ using the above procedure, and letting $\hat{\boldsymbol{m}}(\boldsymbol{y})$ be their empirical average. $\qquad\square$

## T.2    PROOF OF THEOREM 5

The next statement makes a weaker assumption on the accuracy of the diffusion sampler (transportation instead of KL distance), but in exchnage assumes the approximate drift $\hat{\boldsymbol{m}}$ to be Lipschitz. We note that $\mathrm{Lip}(\boldsymbol{m}(\,\cdot\,, t)) = \sup_{\boldsymbol{y}} \|\mathrm{Cov}(\boldsymbol{x}|\boldsymbol{y}_t = \boldsymbol{y})\|_{\mathrm{op}}$, and the latter is of $O(1/d)$ (for instance) if the coordinates of $\boldsymbol{x}$ are weakly dependent under the posterior.

**Theorem 5.** *Assume that $\hat{m}(\,\cdot\,,\,\cdot\,)$ has computational complexity $\chi$ and satisfies the following: (a) For every $t \geq 1/\sigma^2$, $\boldsymbol{y} \mapsto \hat{m}(\boldsymbol{y},t)$ is $L/d$-Lipschitz. (b) There is a stepsize $\Delta$ such that $W_1(\mathrm{P}_{\hat{\boldsymbol{y}}}^{T,\Delta}, \mathrm{P}_{\boldsymbol{y}}^T) \leq \varepsilon$ for any $T \leq \theta d$.*

*Then for any $\sigma > 0$ there exists an algorithm with complexity $O(\chi \cdot T/\Delta)$, that takes as input $\boldsymbol{y} = \boldsymbol{x} + \sigma \boldsymbol{g}$, $(\boldsymbol{x}, \boldsymbol{g}) \sim \mu \otimes \mathsf{N}(0, \boldsymbol{I})$, and outputs $\hat{\boldsymbol{x}}$, such that*

$$\mathbb{E}_{\boldsymbol{y}} W_1(\mathrm{P}_{\boldsymbol{x}|\boldsymbol{y}}, \mathrm{P}_{\hat{\boldsymbol{x}}|\boldsymbol{y}}) \leq 2e^{\theta L}\varepsilon + \frac{1}{\sqrt{\theta}} =: \bar{\varepsilon}. \tag{88}$$

*As a consequence, Eq. (13) holds also in this case with the new definition of $\bar{\varepsilon}$.*

The algorithm consists in running the discretized diffusion (4) with initialization $\hat{\boldsymbol{y}}_{t_0} = \boldsymbol{y}/\sigma^2$ at $t = t_0 := 1/\sigma^2$. To avoid notational burden, we will assume $(T - t_0)/\Delta$ to be an integer. Let $\hat{\boldsymbol{y}}_{t_0}^*$ be generated by the discretized diffusion with initialization at $\hat{\boldsymbol{y}}_0$ at $t = 0$. Note that the distribution of $\hat{\boldsymbol{y}}_{t_0}$ is the same as the one of $t_0 \boldsymbol{x} + \sqrt{t}\boldsymbol{g}$ and hence by Assumption $(b)$,

$$W_1(\mathrm{P}_{\hat{\boldsymbol{y}}_{t_0}}, \mathrm{P}_{\hat{\boldsymbol{y}}_{t_0}^*}) \leq W_1(\mathrm{P}_{T,\Delta}^{\hat{\boldsymbol{y}}}, \mathrm{P}_T^{\boldsymbol{y}}) \leq \varepsilon. \tag{89}$$

In other words there exists a coupling of $\hat{\boldsymbol{y}}_{t_0}^*$ and $\hat{\boldsymbol{y}}_{t_0}$ such that $\mathbb{E}\|\hat{\boldsymbol{y}}_{t_0}^* - \hat{\boldsymbol{y}}_{t_0}\|_2 \leq \varepsilon$.

We extend this to a coupling of $(\hat{\boldsymbol{y}}_t^*)_{t_0 \leq t \leq T}$ and $(\hat{\boldsymbol{y}}_t)_{t_0 \leq t \leq T}$ in the obvious way: we generate the two trajectories according to the discretized diffusion (4) with the same randomness $\hat{\boldsymbol{z}}_t$. A simple recursive argument (using the Lipschitz property of $\hat{m}$, in Assumption $(a)$) then yields

$$\mathbb{E}\|\hat{\boldsymbol{y}}_T^* - \hat{\boldsymbol{y}}_T\|_2 \leq \left(1 + L\Delta/d\right)^{T/\Delta} \varepsilon \leq e^{LT/d}\varepsilon. \tag{90}$$

(See for instance Montanari & Wu (2023) or Alaoui et al. (2023) for examples of this calculation.) Let now $\boldsymbol{y}_T \stackrel{\mathrm{d}}{=} T\boldsymbol{x} + \sqrt{T}\boldsymbol{g}'$ for $(\boldsymbol{x}, \boldsymbol{g}') \sim \mu \otimes \mathsf{N}(\boldsymbol{0}, \boldsymbol{I})$. Another application of Assumption $(a)$ implies that this can be coupled to $\hat{\boldsymbol{y}}_T^*$ so that $\mathbb{E}\|\boldsymbol{y}_T - \hat{\boldsymbol{y}}_T^*\| \leq \varepsilon$, and therefore

$$\mathbb{E}\|\hat{\boldsymbol{y}}_T - \boldsymbol{y}_T\|_2 \leq 2\, e^{LT/d}\varepsilon. \tag{91}$$

As output, we return $\hat{\boldsymbol{x}} = \hat{\boldsymbol{y}}_T/T$. Using $\mathbb{E}\|\boldsymbol{y}_T - \boldsymbol{x}\| = \mathbb{E}\|\boldsymbol{g}\|/\sqrt{T}$ and $T = \theta d$,

$$\mathbb{E}\|\boldsymbol{x} - \hat{\boldsymbol{x}}\| \leq 2e^{\theta L}\varepsilon + \frac{1}{\sqrt{\theta}}. \tag{92}$$

Since the coupling has been constructed conditionally on $\boldsymbol{y}$, the claim (88) follows.

Finally, Eq. (13) follows by generating $N$ i.i.d. copies $\hat{\boldsymbol{x}}_1, \ldots, \hat{\boldsymbol{x}}_N$ using the above procedure, and letting $\hat{m}(\boldsymbol{y})$ be their empirical average.

## U    PROOF OF LEMMA H.1

We let $\boldsymbol{B} \sim \mathsf{N}(0, \boldsymbol{I}_{n^2})$ so that $\boldsymbol{W} \sim (\boldsymbol{B} + \boldsymbol{B}^{\intercal})/2$. We consider a non-random vector $\boldsymbol{v}$ fitting the description, and note that

$$|\langle \boldsymbol{v}, \boldsymbol{W}\boldsymbol{v}\rangle| \leq \frac{1}{2}\{|\langle \boldsymbol{v}, \boldsymbol{B}\boldsymbol{v}\rangle| + |\langle \boldsymbol{v}, \boldsymbol{B}^{\intercal}\boldsymbol{v}\rangle|\}$$

We know that $\langle \boldsymbol{v}, \boldsymbol{B}\boldsymbol{v}\rangle, \langle \boldsymbol{v}, \boldsymbol{B}^{\intercal}\boldsymbol{v}\rangle \sim \mathsf{N}(0, 1)$. We thus have, by Gaussian tail bounds and a triangle inequality:

$$\mathbb{P}\left(|\langle \boldsymbol{v}, \boldsymbol{W}\boldsymbol{v}\rangle| \geq C\sqrt{\log\binom{n}{k}}\right) \leq 2\exp\left(-\frac{C^2}{2}\log\binom{n}{k}\right)$$

Union bounding over the set of all such vectors gives us the desired statement, as the cardinality of this set is $\binom{n}{k}2^k$.

## V  PROOF OF LEMMA H.3

*Proof of Lemma H.3*:  Using Lemma H.2, we can take $x = n^{-1/4}$, say, and $\theta = \sqrt{1+\delta}$ for $\delta = o_n(1)$, to get that

$$\mathbb{P}\left(\lambda_1(\boldsymbol{y}) \leq \theta + 1/\theta - n^{-1/4} - 2/n\right) \leq C \exp(-cn^{1/2})$$

for some absolute constants $C, c > 0$.

We have the following identity, letting $\boldsymbol{W} \sim \mathsf{GOE}(n, 1/n)$:

$$\langle \boldsymbol{u}, \boldsymbol{v}_1 \rangle^2 = \frac{1}{\theta^2 \langle \boldsymbol{u}, (\lambda_1(\boldsymbol{y})\boldsymbol{I} - \boldsymbol{W})^{-2}\boldsymbol{u}\rangle} \geq \frac{1}{\theta^2 \cdot \|(\lambda_1(\boldsymbol{y})\boldsymbol{I} - \boldsymbol{W})^{-2}\|_{\mathrm{op}}}$$

By standard Gaussian concentration, we know that

$$\mathbb{P}\left(\|\boldsymbol{W}\|_{\mathrm{op}} \geq 2 + x\right) \leq C \exp(-cnx^2)$$

We take

$$x = \frac{\theta + 1/\theta - n^{-1/4} - 2/n - 2}{4}$$

Note that with $\theta = \sqrt{1+\delta}$ and $\delta = o_n(1)$, we know that $\theta + 1/\theta - 2 = \Theta(\delta^2)$, so that $x = \Theta(\delta^2)$ if $\delta \gg n^{-1/8}$. Furthermore we have, by Theorem 1 above,

$$\mathbb{P}\left(\lambda_{\min}(\lambda_1(\boldsymbol{y})\boldsymbol{I} - \boldsymbol{W}) \leq 2x\right) \leq C \exp(-cn^{1/2})$$

and on the complement of this event, we know that

$$\langle \boldsymbol{v}_1, \boldsymbol{u} \rangle^2 \geq \frac{4x^2}{\theta^2} = \Theta(\delta^4)$$

since $\theta = \Omega(1)$, and so $|\langle \boldsymbol{v}_1, \boldsymbol{u} \rangle| = \Omega(\delta^2)$. Hence we are done.

## W  PROOF OF THEOREM 3

The optimality of $\hat{\boldsymbol{m}}$ with respect to scalings $c\hat{\boldsymbol{m}}$ implies, by Pythagoras' theorem:

$$\mathbb{E}\{\|\hat{\boldsymbol{m}}(\boldsymbol{y}_t, t) - \boldsymbol{x}\|^2\} = \mathbb{E}\{\|\boldsymbol{x}\|^2\} - \mathbb{E}\{\|\hat{\boldsymbol{m}}(\boldsymbol{y}_t, t)\|^2\},$$

whence, using assumption (16), we obtain that

$$\sup_{t \leq (1-\gamma)t_{\mathrm{alg}}} \mathbb{E}[\|\hat{\boldsymbol{m}}(\boldsymbol{y}_t, t)\|^2] = o(t_{\mathrm{alg}}^{-1}). \tag{93}$$

Recall that $(\hat{\boldsymbol{y}}_t)$ is the generated diffusion, defined in Eq.(4). From Girsanov's formula on $[0, (1-\gamma)t_{\mathrm{alg}}]$, we get that:

$$\mathsf{KL}\left((\boldsymbol{y}_t)_{t\in\mathbb{N}\Delta\cap[0,(1-\gamma)t_{\mathrm{alg}}]} \| (\hat{\boldsymbol{y}}_t)_{t\in\mathbb{N}\Delta\cap[0,(1-\gamma)t_{\mathrm{alg}}]}\right) = \frac{\Delta}{2} \sum_{t\in\mathbb{N}\Delta\cap[0,(1-\gamma)t_{\mathrm{alg}}]} \mathbb{E}[\|\hat{\boldsymbol{m}}(\boldsymbol{y}_t, t)\|^2] = o(1)$$

From Eq. (98), we get from Markov's inequality that with high probability,

$$\frac{\Delta}{2} \sum_{t\in\mathbb{N}\Delta\cap[0,(1-\gamma)t_{\mathrm{alg}}]} \|\hat{\boldsymbol{m}}(\boldsymbol{y}_t, t)\|^2 = o(1) \overset{(a)}{\Rightarrow} \frac{\Delta}{2} \sum_{t\in\mathbb{N}\Delta\cap[0,(1-\gamma)t_{\mathrm{alg}}]} \|\hat{\boldsymbol{m}}(\boldsymbol{y}_t, t)\| = o(\sqrt{t_{\mathrm{alg}}}),$$

where $(a)$ follows by Cauchy-Schwarz. By Pinsker's inequality on Eq. (99), we obtain that the same event holds for $(\hat{\boldsymbol{y}}_t)$ with high probability:

$$\frac{\Delta}{2} \sum_{t\in\mathbb{N}\Delta\cap[0,(1-\gamma)t_{\mathrm{alg}}]} \|\hat{\boldsymbol{m}}(\hat{\boldsymbol{y}}_t, t)\| = o(\sqrt{t_{\mathrm{alg}}}).$$

Fix a constant $\varepsilon_0 > 0$ to be chosen later. By taking the constant $\gamma$ to be close enough to 1, we get that for $t_b := \min\{\ell\Delta : \ell\Delta \geq (1+\delta)t_{\mathrm{alg}}\}$:

$$\hat{\boldsymbol{y}}_{t_b} = \boldsymbol{B}_{t_b} + \Delta \sum_{t\in\mathbb{N}\Delta\cap[0,t_b]} \hat{\boldsymbol{m}}_n(\hat{\boldsymbol{y}}_t, t) := \boldsymbol{m}_0 + \boldsymbol{B}_{t_b}$$

with $\mathbb{P}(\|\boldsymbol{m}_0\| \geq \varepsilon_0 t_{\mathrm{alg}}) = o(1)$. Next we couple $(\hat{\boldsymbol{y}}_t : t \geq t_b)$ to $(\hat{\boldsymbol{y}}_t^0 : t \geq t_b)$ defined by letting $\hat{\boldsymbol{y}}_{t_b}^0 = \boldsymbol{B}_{t_b}$ and, for $t \in \mathbb{N}\Delta \cap [t_b, \infty)$,

$$\hat{\boldsymbol{y}}_{t+\Delta}^0 = \hat{\boldsymbol{y}}_t^0 + \hat{\boldsymbol{m}}(\hat{\boldsymbol{y}}_t^0, t)\Delta + \boldsymbol{B}_{t+\Delta} - \boldsymbol{B}_t\,.$$

By the assumed Lipschitz property of $\hat{\boldsymbol{m}}$ and Gromwall's lemma:

$$\|\hat{\boldsymbol{y}}_t - \hat{\boldsymbol{y}}_t^0\| \leq \prod_{t' \in \mathbb{N}\Delta \cap [t_b, t]} \left(1 + \frac{C\Delta}{t'}\right) \cdot \|\boldsymbol{m}_0\|$$

$$\leq \left(\frac{t}{t_{\mathrm{alg}}}\right)^{C'} \|\boldsymbol{m}_0\| \leq C' \varepsilon_0 t_{\mathrm{alg}}\,, \tag{94}$$

where the last inequality holds for some absolute constant $C'$ and all $t \leq C t_{\mathrm{alg}}$, on the high probability event $\|\boldsymbol{m}_0\| \leq \varepsilon_0 t_{\mathrm{alg}}$.

We are now in position to finish the proof of the theorem. We couple the process $(\hat{\boldsymbol{y}}_t^0 : t \geq t_b)$ defined above with $(\boldsymbol{B}_t : t \geq t_b)$ to get

$$\mathsf{KL}(\boldsymbol{B}_{t+\Delta}\|\hat{\boldsymbol{y}}_{t+\Delta}^0) \leq \mathsf{KL}(\boldsymbol{B}_t\|\boldsymbol{y}_t^0) + C\,\mathbb{E}\{\|\hat{\boldsymbol{m}}(\boldsymbol{B}_t, t)\|^2\} \cdot \Delta$$

Using $\mathsf{KL}(\boldsymbol{B}_{t_b}\|\hat{\boldsymbol{y}}_{t_b}^0) = 0$, summing the last inequality over $t \geq t_b$, and applying Pinsker's inequality we obtain, with $C'$ a suitably large constant

$$\sup_{t \in \mathbb{N}\Delta \cap [t_{\mathrm{alg}}(1+\delta), \infty)} \mathsf{TV}(\hat{\boldsymbol{y}}_t^0, \boldsymbol{B}_t) = o(1)\,. \tag{95}$$

as long as

$$\Delta \cdot \sum_{t \in \mathbb{N}\cdot\Delta \cap [t_{\mathrm{alg}}(1+\delta), \infty]} \mathbb{E}[\|\hat{\boldsymbol{m}}(\boldsymbol{B}_t, t)\|^2] = o(1)$$

Consequently, by using the function $\hat{\boldsymbol{m}}$, along with Eq. (15), we get that for some $c_n = o_n(1)$,

$$\sup_{t \in \mathbb{N}\Delta \cap [t_{\mathrm{alg}}(1+\delta), \infty)} \mathbb{P}(\|\hat{\boldsymbol{m}}(\hat{\boldsymbol{y}}_t^0, t)\| \geq c_n) = o(1)$$

Putting together this bound and Eq. (94) (which holds with high probability) we obtain from the Lipschitz property of $\hat{\boldsymbol{m}}$,

$$\max_{t \in [t_{\mathrm{alg}}(1+\delta), C t_{\mathrm{alg}}]} \mathbb{P}(\|\hat{\boldsymbol{m}}(\hat{\boldsymbol{y}}_t, t)\| \geq \varepsilon_0) = o(1)\,, \tag{96}$$

(we absorb $C'\varepsilon_0$ from Eq. (94) into $\varepsilon_0/2$ as it is arbitrary). Hence

$$\inf_{t \in [t_{\mathrm{alg}}(1+\delta), C t_{\mathrm{alg}}]} W_1(\hat{\boldsymbol{m}}(\hat{\boldsymbol{y}}_t, t), \boldsymbol{x}) \geq \alpha - \varepsilon_0 + o(1)$$

By taking $\varepsilon_0 \downarrow 0$, we obtain the claim of the theorem.

## X    PROOF OF COROLLARY 5.1

In order to simplify some of the formulas below we center $\mu_{n,k}$. Namely, we redefine $\mu_{n,k}$ to be the distribution of $\boldsymbol{x} = \boldsymbol{u}\boldsymbol{u}^\mathsf{T} - \mathbb{E}[\boldsymbol{u}\boldsymbol{u}^\mathsf{T}]$ when $\boldsymbol{u} \sim \mathrm{Unif}(B_{n,k})$.

Throughout this proof, $C$ denotes a generic constant which depends on the constants in the assumptions, and is allowed to change from line to line. We will write $\mathbb{E}_{n,k}$ for expectation under $\mu_{n,k}$ and $\mathbb{E}$ for expectation under $\overline{\mu}_{n,k} = \frac{1}{2}\mu_{n,k} + \frac{1}{2}\delta_{\mathbf{0}}$. Further $\boldsymbol{y}_t = t\boldsymbol{x} + \boldsymbol{W}_t$, where the distribution of $\boldsymbol{x}$ is either $\mu_{n,k}$ or $\overline{\mu}_{n,k}$ as indicated.

The optimality of $\hat{\boldsymbol{m}}_n$ with respect to scalings $c\hat{\boldsymbol{m}}_n$ implies, by Pythagoras' theorem:

$$\mathbb{E}\{\|\hat{\boldsymbol{m}}_n(\boldsymbol{y}_t, t) - \boldsymbol{x}\|^2\} = \mathbb{E}\{\|\boldsymbol{x}\|^2\} - \mathbb{E}\{\|\hat{\boldsymbol{m}}_n(\boldsymbol{y}_t, t)\|^2\}\,,$$

whence, using assumption (16), we obtain that

$$\sup_{t \leq (1-\gamma)t_{\mathrm{alg}}} \mathbb{E}[\|\hat{\boldsymbol{m}}_n(\boldsymbol{y}_t, t)\|^2] = o_n(n^{-1})\,. \tag{97}$$

By Law of Total Probability, we have

$$\mathbb{E}[\|\hat{\boldsymbol{m}}_n(\boldsymbol{y}_t, t)\|^2] = \frac{1}{2}\mathbb{E}_{\boldsymbol{x} \sim \mu_{n,k}}[\|\hat{\boldsymbol{m}}_n(\boldsymbol{y}_t, t)\|^2] + \frac{1}{2}\mathbb{E}[\|\hat{\boldsymbol{m}}_n(\boldsymbol{W}_t, t)\|^2],$$

from which we get

$$\sup_{t \leq (1-\gamma)t_{\text{alg}}} \mathbb{E}[\|\hat{\boldsymbol{m}}_n(\boldsymbol{W}_t, t)\|^2] = o_n(n^{-1}) \tag{98}$$

From Girsanov's formula on $[0, (1-\gamma)t_{\text{alg}}]$, we get that

$$\mathsf{KL}\left((\boldsymbol{W}_t)_{t \in \mathbb{N}\Delta \cap [0,(1-\gamma)t_{\text{alg}}]} \| (\hat{\boldsymbol{y}}_t)_{t \in \mathbb{N}\Delta \cap [0,(1-\gamma)t_{\text{alg}}]}\right) = \frac{\Delta}{2} \sum_{t \in \mathbb{N}\Delta \cap [0,(1-\gamma)t_{\text{alg}}]} \mathbb{E}[\|\hat{\boldsymbol{m}}_n(\boldsymbol{W}_t, t)\|^2] = o_n(1) \tag{99}$$

due to the fact that $t_{\text{alg}} = n/2$. From Eq. (98), we get from Markov's inequality that with high probability,

$$\frac{\Delta}{2} \sum_{t \in \mathbb{N}\Delta \cap [0,(1-\gamma)t_{\text{alg}}]} \|\hat{\boldsymbol{m}}_n(\boldsymbol{W}_t, t)\|^2 = o_n(1) \overset{(a)}{\Rightarrow} \frac{\Delta}{2} \sum_{t \in \mathbb{N}\Delta \cap [0,(1-\gamma)t_{\text{alg}}]} \|\hat{\boldsymbol{m}}_n(\boldsymbol{W}_t, t)\| = o_n(\sqrt{n}),$$

where $(a)$ follows by Cauchy-Schwarz. By Pinsker's inequality on Eq. (99), we obtain that the same event holds for $(\hat{\boldsymbol{y}}_t)$ with high probability:

$$\frac{\Delta}{2} \sum_{t \in \mathbb{N}\Delta \cap [0,(1-\gamma)t_{\text{alg}}]} \|\hat{\boldsymbol{m}}_n(\hat{\boldsymbol{y}}_t, t)\| = o_n(\sqrt{n}).$$

Fix a constant $\varepsilon_0 > 0$ to be chosen later. By taking the constant $\gamma$ to be close enough to 1, we get that for $t_b := \min\{\ell\Delta : \ell\Delta \geq (1+\delta)t_{\text{alg}}\}$:

$$\hat{\boldsymbol{y}}_{t_b} = \boldsymbol{B}_{t_b} + \Delta \sum_{t \in \mathbb{N}\Delta \cap [0, t_b]} \hat{\boldsymbol{m}}_n(\hat{\boldsymbol{y}}_t, t) := \boldsymbol{m}_0 + \boldsymbol{B}_{t_b}$$

with $\mathbb{P}(\|\boldsymbol{m}_0\| \geq \varepsilon_0 n) = o_n(1)$, and $(\hat{\boldsymbol{y}}_t)$ is the generated diffusion, defined in Eq.(4). Next we couple $(\hat{\boldsymbol{y}}_t : t \geq t_b)$ to $(\hat{\boldsymbol{y}}_t^0 : t \geq t_b)$ defined by letting $\hat{\boldsymbol{y}}_{t_b}^0 = \boldsymbol{B}_{t_b}$ and, for $t \in \mathbb{N}\Delta \cap [t_b, \infty)$,

$$\hat{\boldsymbol{y}}_{t+\Delta}^0 = \hat{\boldsymbol{y}}_t^0 + \hat{\boldsymbol{m}}_n(\hat{\boldsymbol{y}}_t^0, t)\Delta + \boldsymbol{B}_{t+\Delta} - \boldsymbol{B}_t \,.$$

By the assumed Lipschitz property of $\hat{\boldsymbol{m}}$ and Gronwall's lemma:

$$\|\hat{\boldsymbol{y}}_t - \hat{\boldsymbol{y}}_t^0\| \leq \prod_{t' \in \mathbb{N}\Delta \cap [t_b, t]} \left(1 + \frac{C\Delta}{t'}\right) \cdot \|\boldsymbol{m}_0\|$$

$$\leq \left(\frac{t}{t_{\text{alg}}}\right)^{C'} \|\boldsymbol{m}_0\| \leq C'\varepsilon_0 n \,, \tag{100}$$

where the last inequality holds for some absolute constant $C'$ and all $t \leq Cn = (2C)t_{\text{alg}}$, on the high probability event $\|\boldsymbol{m}_0\| \leq \varepsilon_0 n$.

In order to finish the proof, we state and prove a useful lemma. In a nutshell, $\hat{\boldsymbol{m}}_n$ resists improvements from eigenvalue hypothesis tests:

**Lemma X.1.** *Under the assumptions of Theorem 5.1, assume that $\delta_n$ vanishes slowly enough. Then, for $t \geq (1+\delta)t_{\text{alg}}$,*

$$\mathbb{E}\{\|\hat{\boldsymbol{m}}_n(\boldsymbol{B}_t, t)\|^2\} \leq Ce^{-(\sqrt{t}-\sqrt{t_{\text{alg}}})^4/Cn} \,. \tag{101}$$

*Proof.* Let $\lambda_1(\boldsymbol{y}_t)$ be the maximum eigenvalue of $(\boldsymbol{y}_t + \boldsymbol{y}_T^{\mathsf{T}})/\sqrt{2}$, $\lambda_*(t) := \sqrt{2}(\sqrt{t} + \sqrt{t_{\text{alg}}})^2$ and $\phi(\boldsymbol{y}_t) := \mathbf{1}(\lambda_1(\boldsymbol{y}_t) > \lambda_*(t))$. Concentration results about spiked GOE matrices imply, for all $t \geq (1+\delta)t_{\text{alg}}$,

$$\mathbb{P}_{n,k}(\phi(\boldsymbol{y}_t) = 0) \leq Ce^{-n(\sqrt{t}-\sqrt{t_{\text{alg}}})^4/C}, \quad \mathbb{P}(\phi(\boldsymbol{B}_t) = 0) \leq C \exp\left\{-\frac{1}{Cn}(\sqrt{t} - \sqrt{t_{\text{alg}}})^4\right\}. \tag{102}$$

(To simplify notations, we omit the dependence of $\phi$ on $t$.)

By assumption, the MSE of $\hat{\boldsymbol{m}}_n(\boldsymbol{y}_t, t)$ is not larger than the one of $\hat{\boldsymbol{m}}_n(\boldsymbol{y}_t, t)\phi(\boldsymbol{y}_t)$. Letting $\overline{\phi}(\boldsymbol{y}_t) := 1 - \phi(\boldsymbol{y}_t)$:

$$
\begin{aligned}
\mathbb{E}\{\|\hat{\boldsymbol{m}}(\boldsymbol{y}_t, t) - \boldsymbol{x}\|^2\} &\leq \mathbb{E}\{\|\hat{\boldsymbol{m}}(\boldsymbol{y}_t, t)\phi(\boldsymbol{y}_t) - \boldsymbol{x}\|^2\} \\
&= \mathbb{E}\{\|\hat{\boldsymbol{m}}(\boldsymbol{y}_t, t) - \boldsymbol{x}\|^2\phi(\boldsymbol{y}_t)\} + \mathbb{E}\{\|\boldsymbol{x}\|^2\overline{\phi}(\boldsymbol{y}_t)\} \\
&= \mathbb{E}\{\|\hat{\boldsymbol{m}}(\boldsymbol{y}_t, t) - \boldsymbol{x}\|^2\phi(\boldsymbol{y}_t)\} + \mathbb{P}_{n,k}\big(\phi(\boldsymbol{y}_t) = 0\big), \quad (103)
\end{aligned}
$$

whence

$$
\mathbb{E}\{\|\hat{\boldsymbol{m}}(\boldsymbol{y}_t, t) - \boldsymbol{x}\|^2\overline{\phi}(\boldsymbol{y}_t)\} \leq \mathbb{P}_{n,k}\big(\phi(\boldsymbol{y}_t) = 0\big). \quad (104)
$$

On the other hand

$$
\begin{aligned}
\mathbb{E}\{\|\hat{\boldsymbol{m}}(\boldsymbol{y}_t, t) - \boldsymbol{x}\|^2\overline{\phi}(\boldsymbol{y}_t)\} &\geq \mathbb{E}\{\|\hat{\boldsymbol{m}}(\boldsymbol{y}_t, t)\|^2 \mathbf{1}_{\boldsymbol{x}=0}\overline{\phi}(\boldsymbol{y}_t)\} \\
&= \frac{1}{2}\mathbb{E}\{\|\hat{\boldsymbol{m}}(\boldsymbol{B}_t, t)\|^2\overline{\phi}(\boldsymbol{W}_t)\} \\
&\geq \frac{1}{2}\mathbb{E}\{\|\hat{\boldsymbol{m}}(\boldsymbol{B}_t, t)\|^2\} - \frac{1}{2}\mathbb{P}\big(\phi(\boldsymbol{W}_t) = 1\big). \quad (105)
\end{aligned}
$$

Putting together Eqs. (102), (104), (105), we obtain (eventually adjusting the constant $C$)

$$
\begin{aligned}
\mathbb{E}\{\|\hat{\boldsymbol{m}}(\boldsymbol{B}_t, t)\|^2\} &\leq \mathbb{P}\big(\phi(\boldsymbol{B}_t) = 1\big) + 2\,\mathbb{P}_{n,k}\big(\phi(\boldsymbol{y}_t) = 0\big) \\
&\leq C\exp\Big\{-\frac{1}{Cn}(\sqrt{t} - \sqrt{t_{\mathrm{alg}}})^4\Big\}.
\end{aligned}
$$

$\square$

From Lemma X.1, Condition 2 of Theorem 3 is satisfied. Hence we are done.

## Y  DETAILS OF NUMERICAL SIMULATIONS

Our GNN architecture uses node embeddings that are generated by 3 iterations of the power method and 10 message passing layers. Each message passing layer comprises of the 'message' and 'node-update' multi-layer perceptrons (MLPs), both of which are 2-layer neural networks with LeakyReLU nonlinearity. We simply use the complete graph with self-loops for node embedding updates. We find that 'seeding' the node embeddings with iterations of power method is crucial for effective training.

During training of the denoiser, we sample time points $t$ as follows: choose a time threshold $t_\star$, and sample so that times $t > t_\star$ are picked with total probability 0.95 (and times $t \leq t_\star$ are picked with total probability 0.05). Within each interval $(0, t_\star]$ and $(t_\star, T)$, times are chosen at random. This allows the neural network to initially prioritize learning in a low-noise regime. Several fine-tuning steps are taken, for which $t_\star$ is gradually decreased to refine the network on lower SNR.

Empirically, training directly with 10 layers is difficult, due to its depth. We find that training initially with 7 layers, then subsequently introducing the later layers results in more stable training.

We train such a network using $N = 30000$ samples $\boldsymbol{x}_i$ from the distribution $\overline{\mu}_{n,k}$, and evaluate their MSE on $N_{\mathrm{test}} = 15000$ samples.

