# OpenReview forum: "Computational Bottlenecks for Denoising Diffusions"
_ICLR.cc/2026/Conference — ICLR 2026 Poster_

### Official Review · Reviewer_6LQL · 2025-10-31

**Soundness:** 4
**Presentation:** 3
**Contribution:** 4
**Rating:** 8
**Confidence:** 4

**Summary:**

This paper addresses a very interesting question: Can denoising diffusions efficiently sample from any distribution $\mu$ for which sampling is otherwise tractable?

The paper presents a rigorous negative answer by showing that diffusion samplers inherit the computational hardness of the underlying denoising task. The core mechanism of diffusion relies on learning the drift term $m(y,t)$, i.e. the Bayes-optimal denoiser. If this denoising problem exhibits an Information-Computation Gap (ICG), a regime where the statistically optimal error is low, but no polynomial-time algorithm can achieve it, then the authors show that diffusion sampling based on score-matching must fail.
The paper includes several key theoretical results.
Theorems 2 \& 5): Efficient diffusion sampling implies efficient near-Bayes-optimal estimation. The contrapositive confirms the central thesis: computational hardness in denoising implies diffusion sampling fails.
Theorem 1: The paper proves the existence of polynomial-time drifts that are super-polynomially close to the optimum (among poly-time algorithms) in the score-matching objective, yet produce samples maximally far from the target $\mu$ (in Wasserstein distance). So the standard training objective is insufficient and potentially misleading when an ICG exists.
Theorem 3: The paper shows that any poly-time, near-optimal drift satisfying a mild any \emph{Lipschitz} near-optimizer fails to sample correctly. Furthermore, \emph{Corollary~5.1} provides a concrete sufficient condition: a $C/t$-Lipschitz drift fails when $t \ge (1+\delta)\,t_{\mathrm{alg}}$.

These results are illustrated using the sparse PCA model where sampling is trivial but denoising is conjectured to be hard below a threshold $t_{\mathrm{alg}}$. Empirical results show that the diffusion process produces degenerate samples.

This work is very significant. It goes beyond analyzing statistical generalization or discretization errors, identifying fundamental limits of denoising diffusions.

**Strengths:**

Originality. The paper goes beyond the previous analysis of diffusion samplers devoted to KL/Wasserstein bounds or generalization arguments. It provides an interesting computational perspective. In particular, it identifies the information-computation gap as the key problem for correct diffusion sampling; it shows that even near-optimal score-matching drifts (among poly-time methods) can sample the wrong distribution and it establishes reductions from diffusion sampling to denoising.

Quality: Theorems 1--3 are stated with clear hypotheses, and Appendix proofs are thorough. The proofs are relatively easy to follow given the technicality of arguments.

Clarity: The paper is well-written. Assumptions specifying the ICG gap are stated clearly. The paper separates neatly the main conceptual arguments in the main text from the proofs. The figures nicely illustrate the theoretical claims.

Significance: This is a very interesting paper explaining when diffusion samplers fail for computational reasons—even when sampling from $\mu$ is itself easy. They also specify when positive results (which rely on accurate scores or favorable distribution classes) do not apply.

**Weaknesses:**

Dependence on assumptions: The main results depend on Assumptions 1 and 2, which do not hold unless an ICG is present. This aligns with predictions for models like Sparse PCA, the underlying hardness remains conjectural in the general case.
It would help to map Assumptions 1 and 2 to specific hardness conjectures in the literature.

Robustness of Theorem 1: Th 1 shows that sampling can fail when the drift is $O(n^{-D})$ close to the optimum among poly-time algorithms. This is a fairly stringent definition of near-optimality. How robust are the results to this assumption? even an heuristic argument would be good.

Construction of the Drift (Theorem 1) The drift $\hat{m}$ constructed to prove Theorem 1 is somewhat artificial. While this demonstrates the insufficiency of the score matching objective, it does not demonstrate that such deceptive drifts arise naturally during standard training procedures (e.g., gradient descent on neural networks).
        \item \emph{Actionable Insight:} Discuss whether standard optimization dynamics are likely to find these specific deceptive local minima, or if this is primarily an existential proof. Are there architectural choices or regularization techniques that might steer optimization away from these solutions?
    \end{itemize}

Lipschitz assumption. Th 3 establishes a negative result for Lipschitz drifts, specifically requiring a $C/t$-Lipschitz condition for $t \ge (1+\delta)t_{\mathrm{alg}}$. However, non-Lipschitz drifts might succeed.
 It would be good to clarify whether this Lipschitz condition is key to the failure of just a technical requirement. Are the typical architectures used in practice (e.g. your GNN in section 3) satisfy this specific $C/t$-Lipschitz property?

Applicability of the results. The analysis focuses on distributions where the ICG is well understood, e.g. Sparse PCA. How are these findings translate to standard distributions meet in physical systems; e.g. Lennard-Jones or \phi^4?
It would be beneficial if the authors could discuss even informally the characteristics of  distributions that might lead to a ICG.

Minor point: The authors follow a non-standard presentation of diffusion models (the one originating from stochastic localization). I know this is equivalent but i think it hurts the readability of the paper.

**Questions:**

Alternative Objectives: Proposition 2.1 suggests that score matching is the root cause of the failure when ICGs are present. Does this motivate the development of alternative training objectives for diffusion models that do not strictly enforce denoising optimality, particularly at low SNR (small $t$)?

Optimization: Theorem 1 implies that the score matching loss landscape possessing near-optimal solutions (in the poly-time class) that are bad for sampling.  Do we know anything about whether standard optimization methods are likely to find these bad near-optimizers rather than potential good ones?

---

> ### Author Response · Authors · 2025-11-20
>
> We thank the reviewer for their review and support of our work.
>
> **Weaknesses**
>
> - *Dependence on assumptions.* At the end of Section 3 we provide several other examples
>     for which we believe that the Theorem assumptions can be verified using results in the literature.
>     We can add further details to these examples.
>  - *Robustness of Theorem 1.* Theorem 1 does not make specific assumptions on the rate of suboptimality,
>     and applies generally. In Corollary 3.2 did our best to make this sub-optimality term in point M2 of order $O(n^{-D})$
>     as small as possible because it implies a stronger result: even drifts very close to optimal fail (note that Condition M2 is a result, not an assumption). Vice versa, the same proof implies that there exist drifts with weaker sub-optimality error and still failing to sample correctly.
>
> Regarding the robustness of Assumptions 1 and 2, Assumption 2 can be checked directly to have this rate of $O(n^{-D})$; moreover, for Assumption 1, it is well-believed that the best (asymptotic) polynomial-time denoiser below the computational threshold $t_{\text{alg}}$ is the (constant) marginal expectation $\mathbb{E}[\mathbf{x}]$. Consequently, the rate of $O(n^{-D})$ makes sense for both Assumptions in the setting of Corollary 3.2.
> - *Construction of the Drift (Theorem 1).* We can add a discussion of this point. In particular:
> 1. Empirically (Section 6) we observe that the denoiser produced by stochastic gradient descent with a standard GNN architecture fails at sampling from this problem. Further the failure mode is comparable to the prediction of our theory (the generated distribution concentrates close to $\boldsymbol{0}$).
> 2. Theorem 3 confirms that *all* Lipschitz continuous neural networks fail.
> 3. As discussed below, we believe that a more promising approach would be to identify and perform diffusions in latent space.
>
> - *Lipschitz assumption.* This assumption is mainly technical. The mathematical challenge in this problem is the following. Near-optimality with respect to score matching objective tells us what the denoiser will do (approximately) for typical $\mathbf{y}_t$ that are distributed according to the "correct distribution" (i.e. ${\mathbf{y}_t} \sim\mathcal{N}(t \mathbf{x},t \mathbf{I})$). However it is possible that the denoiser will perform in a very strange way outside this typical inputs, in such a way that:
> 1. At generation time $\mathbf{y}_t$ is atypical for all $t$;
> 2. The final distribution is correct.
>
> Of course this is not expected to happen for an architecture/optimization algorithm that is oblivious to the problem structure. However it cannot be ruled out unless we either assume some form of regularity.
>
> *Applicability of the results.*
>     Examples of Gibbs measures presenting this failure of diffusion sampling have been given in [1, 2, 3, 4] (although without a formal proof). From a physics viewpoint, these are mean field systems close to a glass transition. From a statistical viewpoint, these are problems for which the minimum mean square error undergoes a discontinuous phase transition. It is an interesting direction to understand what are the implications beyond mean field (we believe there are). We already reference these works in other paper: we will make more explicit their relevance.
>
> *Minor point.* We have written a new appendix section explaining the correspondence between ours and the papers suggested in the review. We hope this improves the readability of our paper.
>
> We also point out that many calculations/arguments
> that requires ad-hoc book-keeping in the standard formulation become straightforward in the stochastic localization formulation. For instance,
> the best time discretization bounds for diffusions
> were proven very elegantly within this formulation in [5].
>
> **Questions**
>
> *Alternative Objectives:* The construction of alternative objectives is an interesting
> research direction, but we do not think the construction in Proposition 2.1 optimizes any objective that might
> be applicable beyond this example.
>
> We believe that a more promising approach is outlined at the beginning of Section 2. Namely, identify
> latent low-dimensional structures in the distribution $\mu$, and use diffusions to sample these latent
> structure. For instance, in the present case $\mathbf{x}=\mathbf{u} \mathbf{u}^{\top}$, and denoising diffusions can be
> used to sample the sparse vector $\mathbf{u}$.
>
> Studying this approach is for future work.
>
> *Optimization:* A precise answer to this question would require to specify the network architecture and the optimization algorithm, while our theorems do not rely on any specific choice of these. On the other hand, Theorem 3 establishes that (modulo technical conditions all nearly optimal denoisers (neural networks) that are also Lipschitz continuous necessarily fail. So, changing the optimization algorithm cannot solve the issue.

---

> > ### Comment · Reviewer_6LQL · 2025-11-20
> > **--**
> >
> > Thanks for clarifying some of my misunderstandings. I think this paper is a very interesting contribution.

---

> ### Author Response · Authors · 2025-11-20
>
> (References)
>
> [1] Andrea Montanari, Federico Ricci-Tersenghi, and Guilhem Semerjian. Solving constraint satisfaction
> problems through belief propagation-guided decimation. arXiv:0709.1667, 2007.
>
> [2] Federico Ricci-Tersenghi and Guilhem Semerjian. On the cavity method for decimated random con-
> straint satisfaction problems and the analysis of belief propagation guided decimation algorithms.
> Journal of Statistical Mechanics: Theory and Experiment, 2009(09):P09001, 2009.
>
> [3] Davide Ghio, Yatin Dandi, Florent Krzakala, and Lenka Zdeborová. Sampling with flows, diffusion,
> and autoregressive neural networks from a spin-glass perspective. Proceedings of the National
> Academy of Sciences, 121(27):e2311810121, 2024.
>
> [4] Brice Huang, Andrea Montanari, and Huy Tuan Pham. Sampling from spherical spin glasses in
> total variation via algorithmic stochastic localization. arXiv:2404.15651, 2024.
>
> [5] Galen Reeves and Henry D Pfister. Information-theoretic proofs for diffusion sampling. arXiv
> preprint arXiv:2502.02305, 2025.

---

### Official Review · Reviewer_uMuZ · 2025-11-01

**Soundness:** 4
**Presentation:** 3
**Contribution:** 4
**Rating:** 8
**Confidence:** 2

**Summary:**

In this paper, the authors study the feasibility of sampling with diffusion models. More precisely, they are interested in answering the following question: "Are there distributions such that, there exist almost optimal denoisers that can be computed in polynomial time and for which diffusion sampling fail?" They answer this question by the affirmative. The paper proceeds as follows:

* In  Theorem 1, they show that for any arbitrary polynomial time denoiser which satisfies two generic assumptions, there exists a modified denoiser for which the diffusion sampling fails. In Corollary 3.1, they apply Theorem 1 to the case of sparse random matrix measures. Other sparse examples are considered

*  While Theorem 1, shows that "good denoiser does not imply good sampling", in Theorem 2 they show that "good sampling implies good denoiser".

* In Theorem 3, they show that under an additional Lipschitz condition, the results can be strengthen to show that *any* polynomial time algorithm which is nearly optimal for the denoising task will lead to poor sampling quality.

The authors conclude the papers with toy numerical experiments.

**Strengths:**

* The paper is theoretically strong. I have to command the quality of the writing (apart from the introduction of diffusion models and some notation choice discussed in the "Weaknesses" section). Even though I am not a domain expert on stochastic localization and random matrix theory, the paper was pleasant to read and the ideas clearly exposed.

* The results presented in the papers will be of great interest for the theoretical diffusion model community. Those results indeed fill a gap in the literature regarding what can be achieved with diffusion models.

**Weaknesses:**

* One of my main pain point with the paper is the choice of following the notation and convention of [1]. While I understand that it might be easier for the authors who seem to familiar with the techniques developed in [1] to build their theory on top of this framework it severely limits the adoption of the work, as it has to be translated back into "classical" diffusion model framework (see [2, 3, 4] for a (non-exhaustive) list of papers which adopt the classical diffusion model convention). The burden of the explanation and communication should be on the authors and not on the readers.

* In my opinion, the strongest result is Theorem 3 but there is little discussion on how realistic is the Lipschitz assumption the authors consider. I found Corollary 5.1. to be very hard to read. To come back to the Lipschitz justification I am not sure I understand the following sentence "We assume the Lipschitz constant to be C/t, because the input of the denoiser is yt = tx + Wt, and hence the two t-dependent factors cancel.)".

* Something that is unclear to me is the generality of the results presented in the paper. While Theorem 1 is very general it is not quite clear to me how to apply them beyond the sparse random matrix measure studied by the authors. In particular, I am wondering how specific those results are. To further specify my question: is (one of) the contribution(s) of the paper 1) the identification of a set of measures under which the assumptions stated in the general Theorems are valid thereby showing some limitations of diffusion models 2) the statement that most data distributions (even the ones considered in practice) fall under the category of the general Theorems thereby showing the much more stronger statement that diffusion models are flawed. It would be good to clarify  the scope of the paper in that respect.

[1] Montanari (2023) -- Sampling, diffusions, and stochastic localization

[2] Conforti et al. (2024) -- KL Convergence Guarantees for Score diffusion models under minimal data assumptions

[3] Li et al. (2024) -- O(d/T) Convergence Theory for Diffusion Probabilistic Models under Minimal Assumptions

[4] Chen et al. (2022) -- Sampling is as easy as learning the score: theory for diffusion models with minimal data assumptions

**Questions:**

* "l. However if such drifts exist, our results suggest that minimizing the score matching objective is not the right approach to find them (since the difference in value with bad drifts will be superpolynomially small)." (l.186) This is an interesting remark. There is some evidence that diffusion models fail at approximating the score. First, they are trained on the empirical measure hence the target distribution is the "optimal" target gives a fully memorizing denoiser.  Second, due to neural network approximations they often fail to correctly approximate the true denoiser. I would be keen to understand how the authors deal with that intrisic limitation of diffusion models and how this would affect their analysis.

* In Proposition 2.1., I do not understand why we can have equality to 0 in (ii).

* Could you provide more details regarding the construction of the drift in Proposition 2.1? (the construction of a drift which is a bad approximation to the true denoiser but gives good diffusion sampling results).

* Before Theorem 1 I think it would be worth providing just a few sentences regarding the validity of Assumption 2 and Assumption 1. At this stage it is very much unclear for the reader if those assumptions are easy or not to satisfy.

**Details Of Ethics Concerns:**

None.

---

> ### Author Response · Authors · 2025-11-20
>
> We thank the reviewer for their review and support of our work.
>
> **Weaknesses**
>
> 1. We agree that our formulation of diffusion models is
> not the most standard.
> - We have written a new appendix section explaining the correspondence between ours and the papers suggested in the review. We hope this improves the readability of our paper.
>
> We also point out that:
> - The formulation we adopt is standard within stochastic localization;
> - Many calculations/arguments that requires ad-hoc book-keeping in the standard formulation become straightforward in the stochastic localization formulation. For instance, the best time discretization bounds for diffusions
> were proven very elegantly within this formulation in [1].
>
> 2. To clarify this point note that we can always write the denoiser as
> $\hat{\mathbf{m}}(\mathbf{y}_t, t)=\mathbf{F}(\mathbf{y}_t/t , t)$, and hence our condition requires $\tilde{\mathbf{y}}_t\mapsto \mathbf{F}(\tilde{\mathbf{y}}_t,t)$
> to be $C$-Lipschitz for some constant independent of $t$.  Now note that this input has
> distribution $\tilde{\mathbf{y}}_t\sim\mathcal{N}(\mathbf{x},\mathbf{I}/t)$, or in other words $\tilde{\mathbf{y}}_t =\mathbf{x}+\mathbf{g}/\sqrt{t}$ with $\mathbf{g}$ standard Gaussian.
> Considering for simplicity large $t$, note that $\mathbf{F}$ is supposed to approximate $\mathbf{x}$ from $\tilde{\mathbf{y}}=\mathbf{x}+O(1/\sqrt{t})$, it is reasonable to believe that it is a Lipschitz map.
>
> Also note that, as an example, an MLP is Lipschitz with Lipschitz constant dependent on the operator norm of weights, and the number of layers (as activation functions are Lipschitz).
>
> Regarding the readability of Corollary 5.1., we have re-phrased the statement. We hope to have improved its presentation.
>
>
> 3. Our contribution is as in your point (1).
>
> Specifically, we show that diffusion sampling fails for distributions $\mu$ such that the
> denoising problem presents an information-computation gap.
> On the other hand, we expect to generally succeed when an information-computation gap is not present.
>
> A wealth of results are available in the literature on information computation gaps,
> and our general results could be applied to other distributions (beyond sparse low-rank matrices) by building on this literature.
> We give several examples at the end of Section 3.
>
> That said, there are also many distributions for which the bottlenecks established in our paper do not arise, because (approximately) Bayes optimal denoising is tractable at all noise levels.
>
> **Questions**
> 1. Our analysis assumes access to the true distribution $\mu$
> (infinite sample size). Hence our negative results cannot be overcome by increasing the sample size
> or by suitable regularizaton (unlike `memorization').
>
> Our results do not rely on the approximation error of specific neural network architectures
> that implement the denoiser. They apply to any architecture as long as inference can be performed in polynomial time.
>
> Theorem 1 shows that even if the sample size is infinite, even the denoiser is nearly optimal
> (among polytime ones) with respect to score-matching objective, and even if the step-size at generation time is arbitrarily small, denoising diffusions will fail.
>
> Additional complications such as finite sample size and limited network capacity can only make matters worse.
>
> 2. It is because we already know the distribution $\mu$ (infinite sample size), and $\mu$ admits an efficient sampler (albeit working in the latent space). Therefore we
> construct a denoiser $\hat{\mathbf{m}}$ that uses only the noise $\hat{\mathbf{z}}\sim \mathcal{N}(\boldsymbol{0}, \mathbf{I})$ to return an output with distribution $\mu$.
> (Note that here we are not claiming $\hat{\mathbf{m}}(\hat{\mathbf{y}}_{\ell\Delta},\ell\Delta)=\mathbf{x}$, we are only claiming that the two distributions coincide.)
>
> 3. The construction is given in Appendix I. We repeat it here.
>
> We take the first row of $\hat{\mathbf{z}}\sim \mathcal{N}(\boldsymbol{0}, \mathbf{I})$, and rank the entries by magnitude. Since the entries are exchangeable, the indices with the top $k$ ranks are a sample without replacement from $[n]$. For each of these indices, we multiply by an independent random sign $\pm 1$, then normalize by $\sqrt{k}$ to make a unit vector. The outer product of this unit vector must follow distribution $\mu$, which is of a rank-1 matrix formed from a $k$-sparse vector with entries in $\{0, \pm1/\sqrt{k}\}$.
>
> [1] Galen Reeves and Henry D Pfister. Information-theoretic proofs for diffusion sampling. arXiv
> preprint arXiv:2502.02305, 2025

---

> ### Author Response · Authors · 2025-11-20
>
> (continued)
>
> 4. Here are our justifications:
> - Assumption 1 can be checked directly from a given drift $\hat{\mathbf{m}}_{0}$, by evaluating the integral for every $\epsilon$ in Assumption 1. We expect this to hold for any reasonable drift in distributions with an information-computation gap (as discussed in Subsection 3.2.1 and Proposition B.1 of our paper).
> - Assumption 2 concerns the existence of a certain efficient hypothesis test $\phi$. Per our answer above (Question 3, Section "Weaknesses"), this assumption may only be satisfied by distributions with an information-computation gap. Regarding verifiability, firstly, checking these conditions for any given $\phi$ is straightforward.
>
> Secondly, we can leverage the literature on information-computation gaps to determine the (conjectured) optimal efficient algorithm $\hat{\mathbf{m}}(\mathbf{y}, t)$, and construct $\phi$ to test for large values of $\langle \hat{\mathbf{m}}(\mathbf{y}, t), \mathbf{y}\rangle$. Specifically, $\phi(\mathbf{y}, t)=\boldsymbol{1}(\langle \hat{\mathbf{m}}(\mathbf{y}, t), \mathbf{y}\rangle\geq c_{\star}t)$, for some $c_{\star}\in (0, 1)$. The rationale is as follows: the maximum likelihood problem for the model $\mathbf{y}_t=t \mathbf{x}+\mathbf{B}_t$ is to find $\hat{\mathbf{x}}$ to maximize $\langle \hat{\mathbf{x}}, \mathbf{y}\rangle$. For $t$ above the algorithmic threshold, efficient estimators $\hat{\mathbf{m}}(\mathbf{y}_t, t)$
>
> can approximate the (inefficient) MLE very well for the alternative model ${\mathbf{y}_{t}}$, leading to large values of $\langle \hat{\mathbf{m}}(\mathbf{y}, t), \mathbf{y}\rangle$ (in fact, it is $t - o(t)$). Note that since $t$ is far above the information-theoretic threshold, the MLE is in turn very close to the true signal $\mathbf{x}$.
>
> The worst-case Type I error guarantees can be checked directly with this $\phi$, and Condition 3 holds for the examples we listed in Subsection 3.2.2. In our paper, we employ a simple observation to show this: for the null model
> $\mathbf{a}+ {\mathbf{B}_t}$,
>
> we have
>
> $$\langle \hat{\mathbf{m}}(\mathbf{a}+\mathbf{B}_t, t), \mathbf{a} + {\mathbf{B}_t}\rangle \leq {\sup _ {\mathbf{x'} \in \text{supp}(\mu)}} \langle \mathbf{x'},  \mathbf{a} + {\mathbf{B}_t}\rangle \leq {||\mathbf{a}|| _ {\text{op}}} + {\sup _ {\mathbf{x'} \in \text{supp}(\mu)}} \langle \mathbf{x}',  {\mathbf{B} _ t}\rangle$$
>
>
> where $\text{supp}(\mu)$ is the support of $\mu$. For distributions with an information-computation gap, $\mathbf{x'}$ is often a "structured" vector (c.f. sparsity and examples in points (i) and (ii) of Subsection 3.2.2); as pure noise ${\mathbf{B} _ t}$ does not favor any structure, we have
>
> $${\sup _ {\mathbf{x'} \in \text{supp}(\mu)}} \langle \mathbf{x'}, {\mathbf{B}_t}\rangle \ll t \Rightarrow \langle \hat{\mathbf{m}}(\mathbf{a}+\mathbf{B}_t, t), \mathbf{a} + {\mathbf{B}_t}\rangle \leq c {t _ {\text{alg}}} + o(t)$$
>
> for $c$ of Condition 3. Choosing $c_{\star}>c$, we have succeeded in constructing $\phi$.

---

### Official Review · Reviewer_Bgxf · 2025-11-06

**Soundness:** 4
**Presentation:** 3
**Contribution:** 3
**Rating:** 6
**Confidence:** 3

**Summary:**

The paper's primary contribution is to rigorously prove that a computational bottleneck, not just a statistical one, limits the power of denoising diffusion models. The authors prove that not all easily-sampled probability distributions can be efficiently learned by these models. They construct a specific counterexample based on the sparse submatrix estimation problem, which is conjectured to be hard. This shows that even a nearly-perfect, efficiently-computed drift function will fail to generate accurate samples, thus identifying a fundamental gap between what is statistically possible and what is computationally feasible for this class of generative models.

**Strengths:**

1. This paper's core strength is its novel perspective, shifting the analysis of diffusion models from standard statistical limits to the overlooked statistical-computational gap.
2. The authors construct an excellent counterexample using exactly the sparse submatrix estimation to provide a distribution that is easy to sample from directly but computationally intractable for a diffusion model to learn.
3. Supportive simulations clearly validate the theory, showing that a computationally-bounded diffusion model fails to generate accurate samples from this "hard" distribution, just as the authors predict.
4. The paper's argument is mathematically rigorous, employing formal, well-established proofs.

**Weaknesses:**

1. The paper's organization is not straightforward; the structure is relatively loose, which makes the argument hard to follow.
2. The planted clique counterexample feels specific, which makes it difficult to connect this theoretical computational bottleneck to the broad success of diffusion models on real-world data.

**Questions:**

1. Would considering the 'average complexity' via diffusion over a class of easily-sampled distributions be more meaningful than focusing on 'bad' extreme counterexamples like the sparse submatrix prior?
2. Do your have conclusions about the training procedure in this paper, or does it imply that for the provided counterexample, the training of diffusion models will always fail due to its inherent computational hardness?
3. Is it possible this specific computational bottleneck is just an artifact of this particular formulation of diffusion sampling, or does it represent a fundamental hardness that alternative generative models like flow-matching models could not bypass?

---

> ### Author Response · Authors · 2025-11-20
>
> We thank the reviewer for their review and support of our work.
>
> **Weaknesses**
>
> 1. Regarding the paper's organization:
> - At the end of page 3, we wrote a `Roadmap' that explains the organization of the paper.
> We plan to expand this roadmap, so that the reader can better grasp the paper contents since the beginning.
> - We have written a new appendix section explaining the correspondence between ours and the papers suggested in the review. We hope this improves the readability of our paper.
>
> 2. We do not agree on this point: our results are not specific to sparse low-rank matrices.
>
> Our main theorems (Theorems 1, 2, 3) apply to general probability distributions under certain
> assumptions on information-computation gaps in the corresponding denoising
> problem. We then use a uniform distribution over sparse low-rank matrices as an example, in which it is relatively convenient to check the assumptions.
>
> At the end of Section 3 we give several other examples in which the assumptions could be checked as well
> modulo some technical work.
> Denoising can present an information-computation gap whenever the
> prior (distribution $\mu$) has a low-dimensional structure that is not linear or factorizable. This situation is pretty
> common, and the sparse low-rank case is merely a convenient example.
>
> **Questions**
> 1. There are obvious examples of sampling problems for which our arguments imply average-case hardness.
> The simplest such example is the one of sampling from the Bayesian posterior in a low-rank plus noise model. More precisely the problem of sampling from the posterior distribution of $\mathbf{x} \in \mathbb{R}^{n\times n}$ given observations $\mathbf{z} = \mathbf{x}+\sigma \mathbf{g}$, where the prior distribution of $\mathbf{x}$ is supported on
> sparse low-rank matrices, and $\mathbf{g}$ is Gaussian noise.
>
> The posterior is a random probability distribution (because of the conditioning) and our arguments can be used to show average case failure of diffusion sampling. We will add a reference to this example, although we defer a thorough analysis to future work.
>
> 2. The bottleneck we demonstrate is independent of the training procedure and even of
> the denoiser (neural network) architecture. It is uniquely related to the diffusion process:
> as long as we follow the standard construction of denoising diffusions, our impossibility results apply.
> (As per the discussion in Section 2, this bottleneck can be circumvented if we let the diffusion operate in a suitable latent space.)
>
> 3. Our diffusion sampling failure stems from the intractability of the true drift $\mathbf{m}(\mathbf{y}, t)=\mathbb{E}[\mathbf{x}|t\mathbf{x}+\sqrt{t}\mathcal{N}(0, \mathbf{I})=\mathbf{y}]$, and we believe it persists for the following formulation of flow-matching models.
>
> Suppose that we want to start from a Gaussian reference $\mathbf{g}\sim\mathcal{N}(\boldsymbol{0}, \mathbf{I})$ and want to sample from a target distribution $\mu$. The interpolated observation is
>
> $$\mathbf{y}_{s}=\alpha(s) \mathbf{x}+\beta(s)\mathbf{g}$$
>
> with $\alpha(0)=0, \alpha(1)=1$, and $\beta(0)=1, \beta(1)=0$. We want to learn the true velocity field $v_{\star}(s, \mathbf{y})$, and
>
> $$v_{\star}(s, \mathbf{y}) = \alpha'(s)\mathbb{E}[\mathbf{x} \mid \mathbf{y}_{s} = \mathbf{y}] + \beta'(s) \mathbb{E}[\mathbf{g}\mid    {\mathbf{y}_s} = \mathbf{y}] = \beta'(s)\mathbf{y}+(\alpha'(s)-\beta'(s)\alpha(s)/\beta(s)) \mathbb{E}[\mathbf{x} \mid {\mathbf{y}_s} = \mathbf{y}]$$
>
> This true velocity field is also a function of the conditional mean $\mathbb{E}[\mathbf{x} \mid {\mathbf{y}_s} = \mathbf{y}]$, and thus is hard to learn when $s$ is small. The inference/sampling process, which involves solving ODEs $(\text{d} / \text{d} t)\hat{\mathbf{y}_t}=\hat{\mathbf{v}}(t, \hat{\mathbf{y}_t})$ with starting Gaussian sample $\hat{\mathbf{y}_0}$, will not be guaranteed to be accurate.

---

> > ### Comment · Reviewer_Bgxf · 2025-11-20
> >
> > Thank you for the explanation; I have no further questions, and I would like to raise my score. Very interesting work.

---

### Author Response · Authors · 2025-12-03
**Summary of comments**

Dear Area Chair and Reviewers,

We thank the reviewers for their support of our work, and would like to summarize the main questions raised during the discussion process.

1. We have made Appendix B explaining the equivalence between our non-standard diffusion models with the time-reversal formulation in literature.
2. Our results do not apply only to sparse low-rank matrices; we use this case to illustrate our more general results (Theorems 1, 3). At the end of Section 3 we offer more examples of distributions with an information-computation gap, for which the conditions of our Theorems can also be verified.
3. *Conditions for Theorem 1*: Assumption 1 for Theorem 1 can be checked for any input poly-time algorithm $\hat{\mathbf{m}}_ {0}$, and we gave justifications in Subsection 3.2.1 and Proposition C.1 for why this condition should hold for reasonable poly-time denoisers. Assumption 2 can be checked directly for any hypothesis test $\phi$, and we gave a specific formula for such $\phi$, per our rebuttal to Reviewer uMuZ. We would also be willing to incorporate this into our final version.
4. The Lipschitz condition in Theorem 3 is mainly technical; we gave details of why our analysis needed this property, per our rebuttal to Reviewer 6LQL.
5. The information-computation gap in our paper could be circumvented for our toy example by operating in latent space ("Setting" paragraph in Discussion section), per our answer to Reviewer 6LQL (point "Alternative Objectives"). Finding the "right" latent space for diffusion sampling is an interesting angle, and we leave this for future work.

In light of the recent policy changes to reviewing, we hope that our summary would provide the AC with sufficient details to evaluate our work. Once again, we thank the AC and the Reviewers for their time and effort.

---

### Meta-Review · Area_Chair_9Zq9 · 2025-12-16

**Summary:**

The paper addresses fundamental problems with using standard diffusion model machinery to perform distribution learning, stemming from statistical-computational gaps. More precisely, by the interpretation of the score as an optimal denoiser, the authors provide impossibility results for distribution learning, assuming some (standard & widely believed) conjectures on statistically-optimal denoising using computationally efficient algorithms. The reviewers were all overall positive, and I am in agreement with them.

The paper is by and large clearly written and easy to read. Two reviewers (uMuZ and 6LQL) mentioned that the exposition and notation follows the stochastic localization framework, rather than the standard diffusion model framework, and this may harm readability. I tend to agree with them --- in large part because the audience at ICLR is much more likely to be familiar with the latter. The authors added an appendix delineating the mapping.

Reviewer 6LQL also pointed out that it isn't clear what "actionable" improvements to the loss or training procedure can be extracted from the lower bounds --- I tend to agree with this, and this is a nice direction for future work.

Finally, the discussion with the reviewers has resulted in various clarification, and added discussions in the paper, which improved it's quality.

**Reviewer Concerns:**

Nothing is unaddressed. The terminology being borrowed from stochastic localization rather than the more standard diffusion model framework was addressed through an appendix. Some discussion on the generality of the results and the restrictiveness of Lipschitzness was added. Some discussion was also added on how how much empirically the training process "tracks" the theoretical construction.

**Reviewer Scores:**

Reviewer Bgxf I think would have raised their score (they say so in their response). The other reviewers would have likely kept their score (which was already positive).

---

### Decision · Program_Chairs · 2026-01-26

Accept (Poster)